# Automatic identification of alternating morphological units in river channel using wavelet analysis and ridge extraction

Mounir Mahdade [1], Nicolas Le Moine [1], Roger Moussa [2], Oldrich Navratil [3], and Pierre Ribstein [1]

[1]Sorbonne Université, CNRS, EPHE, Milieux environnementaux, transferts et interaction dans les hydrosystèmes et les sols, METIS, F-75005 Paris, France
[2]INRA, UMR LISAH, 2 Place Pierre Viala, 34060 Montpellier, France
[3]University of Lyon, Lumière Lyon 2, Department of Geography, CNRS 5600 EVS, France
**Correspondence:** Mounir Mahdade (mounir.mahdade@upmc.fr)

**Abstract.** The accuracy of hydraulic models depends on the quality of the bathymetric data they are based on, whatever the scale at which they are applied. The along-stream (longitudinal) and cross-sectional geometry of *natural* rivers is known to vary at the scale of the hydrographic network (e.g., generally decreasing slope, increasing width), allowing parameterizations of main cross-sectional parameters with large-scale proxy such as drainage area or bankfull discharge (an approach coined downstream hydraulic geometry, DHG). However, higher-frequency morphological variability (i.e., at river reach scale) is known to occur for many stream types, associated with varying flow conditions along a given reach, as for instance, the alternate bars or the pool-riffle sequences and meanders. To consider this high-frequency variability of the geometry in the hydraulic models, a first step is to design robust methods to characterize the scales at which it occurs. In this paper, we introduce new wavelet analysis tools in the field of geomorphic analysis (namely, Wavelet Ridge Extraction), in order to identify the pseudo-periodicity of alternating morphological units from a general point of view (focusing on pool-riffle sequences) for six small French rivers. This analysis can be performed on a single variable (univariate case) but also on a set of multiple variables (multivariate case). In this study we chose a set of four variables describing the flow degrees of freedom : velocity, hydraulic radius, bed shear stress, and a planform descriptor which quantifies the local deviation of the channel from its mean direction. Finally, this method is compared with the Bedform Differencing Technique (BDT), by computing the mean, median, and standard deviation of their longitudinal spacings. The two methods show agreement in the estimation of the wavelength in all reaches except one. The aim of the method is to extract a pseudo-periodicity of the alternating bedforms that allow objectively identifying morphological units in a continuous approach with the respect of correlations between variables (i.e., At Many Station Hydraulic Geometry, AMHG) without the need to define a prior threshold for each variable to characterize the transition from one unit to another.

## 1 Introduction

Hydraulic modeling is based on the description of river morphology (cross-sectional geometry), and this is their essential input despite its scarcity and cost of acquisition. In fact, the most important aspect to know is the river bathymetric data at the local scale, detailed and specific to the site and local conditions (Alfieri et al., 2016). This component is important for an

accurate modeling of river hydraulics such as flood modeling (e.g., Neal et al. (2015); Trigg et al. (2009)), river restoration (e.g., Wheaton et al. (2004)), ecohydraulics (e.g., Pasternack and Brown (2013)), environmental modeling and fluvial process (e.g., Rodríguez et al. (2013)).

Many researchers are working on determining the best simplified representation of channel geometry (Saleh et al., 2013; Grimaldi et al., 2018), based on the variability of cross sections but without the knowledge of the bed elevation variability or the river sinuosity at smaller scale. Other studies focused on the generating of river channels with taking into account the sub-reach scale variability using geostatistics and variogram tools (Legleiter, 2014a, b) or a geometric framework modeling with geomorphic covariance structures (Brown et al., 2014). Longitudinal variability in river geometry may have greater impact on the simulation of the water level than the cross-sectional shapes (Saleh et al., 2013) and it must be taken into account in the hydraulic models. This topographic variability is related to the channel morphology types. In this study, we focus mainly on alternating alluvial channels especially pool-riffle sequences, even though the method presented here could be used to analyze any morphology characterized by alternating topographic forms (morphological units, MUs). The objective is to provide a continuous description of geometric and flow patterns along a reach, a description that could be subsequently used to create a synthetic river as in the RiverBuilder (Brown et al., 2014). To do that we calculate the dimensionless reach wavelength $\lambda^*$, which is the distance between pools (or riffles) divided by average channel width (e.g., Richards (1976a); Keller and Melhorn (1978); Carling and Orr (2000)) or bankfull width (e.g., Leopold et al. (1964)). However, it is necessary to find a method that can extract information concerning these morphologies (position, length, etc.). For this reason, it is interesting to list the works that quantitatively assess this morphological variability.

## 1.1 State of art methods for a quantitative assessment of morphological variability within a reach

Morphological units are topographic forms that shape the river corridor (Wadeson, 1994; Wyrick and Pasternack, 2014). They form alternating and rhythmic undulations continuously varying along the river (Thompson, 2001). This continuity is difficult to represent, for this reason most of the methods that model these patterns divide the topography into discrete units to analyze them (Kondolf, 1995; Wyrick et al., 2014).

Among the most frequently observed alternating MUs, pools and riffles have been recognized as fundamental geomorphological elements of meandering streams (Krueger and Frothingham, 2007). In fact, pools are located in the outer edge of each meander loop and defined as topographic lows along a longitudinal stream profile with high depth and low velocity (Fig. 1 (A), (B) and (D)) and research has shown that they generally have an asymmetrical cross section shape. Conversely, riffles are topographic highs with shallow depths and moderate to high velocities located in the straight parts of the reach between adjacent loops (Fig. 1 (A), (C) and (D)) and have symmetrical cross section shapes (O'Neill and Abrahams, 1984; Knighton, 1981).

For many years, many researchers have been trying to develop techniques to identify MUs and especially pools and riffles using hydraulic variables or topographic ones, or both (Table 1). In the one dimensional identification, some studies used bed topography only to determine the characteristics of MUs. Richards (1976a) proposed the zero-crossing method which fits a regression line to the longitudinal profile of the bed elevation and defines pools as points that have negative residuals and

riffles as points with positive residuals. O'Neill and Abrahams (1984) developed the Bedform Differencing Technique (BDT) as a refinement of Richards' methodology. This one uses bed elevations measured at a fixed interval along the channel to calculate the bed elevation difference series between local extrema (maximum and minimum) of the bed profile. The BDT introduces a tolerance value ($T$), which is the minimum absolute value of the cumulative elevation change required for the identification of a pool or riffle. The value of $T$ is based on the standard deviation of the bed elevation difference series and eliminates the erroneous classification of small undulations in the bed profile. Another method proposed by Knighton (1981) as the Areal Difference Asymmetry Index which is defined as the ratio of the difference between the area of the right and the left of channel centerline on the total cross-sectional area to identify the location of pools and riffles by their symmetrical or asymmetrical areas. On the other hand, some studies focused only on hydraulic parameters to identify MUs. For example, Yang (1971) proposed an identification of pools and riffles using the energy gradient and affirmed that the fundamental difference between riffles and pools is the difference in energy gradients. Also, Jowett (1993) proposed a classification criterion with Froude number and velocity/depth ratio to distinguish between pools, runs, and riffles.

All these methods handle topographic or hydraulic parameters separately. Recently, however, several researchers have improved MUs identification through the use of the covariance of several parameters in a multidimensional approach. Schweizer et al. (2007) used a joint depth and velocity distribution to predict pools, runs, and riffles without the knowledge of the river bathymetry. Hauer et al. (2009) used a functional linkage between depth-averaged velocity, water depth and bottom shear stress to describe and quantify six different hydro-morphological units (riffle, fast run, run, pool, backwater and shallow water) using a conceptual Mesohabitat Evaluation Model (MEM) under various flow conditions. These methods use digital elevation models (DEMs) to extract more information about MUs. In this purpose, Milne and Sear (1997) began with depth to define pool-riffle sequences using ArcGis tools and DEMs to model the geometry of river channels based on field surveyed cross-sections on a three-dimensional basis. But, by choosing depth alone, the difference between two bedforms with the same depth becomes difficult to know. In contrary, it is easy with different bed slopes and bed roughness that yield different velocities and shear stresses (Wyrick et al., 2014). So to overcome this and take into account the lateral variation of rivers, Wyrick et al. (2014) proposed a new method for the objective identification and mapping of landforms at the morphological unit scale. They used spatial grids of depth and velocity at low flow estimated using 2D hydrodynamic model and an expert classification scheme that determine the number and the nomenclature of MUs and range of base flow depth and velocity of each type.

Brown and Pasternack (2017) chose two variables : the minimum bed elevation and the channel top width across several flow discharges. They calculated the geomorphic covariance structure (GCS) which is a bivariate spatial relationship amongst or between standardized and possibly detrended variables along a river corridor. They found that there is a positive correlation between these two variables. Also, they used an autocorrelation function and power spectral density to prove a quasi-periodic pattern of wide and shallow or narrow and deep cross sections along the river. This pioneer work and other studies (e.g., Richards (1976b); Carling and Orr (2000)) proved that a single longitudinal cycle may contain a pool with a narrow and deep cross section, a riffle with a wide and shallow cross section, in addition to transitional forms. The work that we present in this paper aim to present a spectral method that extract this pseudo-periodicity from a river in order to characterize the alternating MUs and especially pool-riffle sequences, and to identify the key parameter (the wavelength) that characterizes the scale of

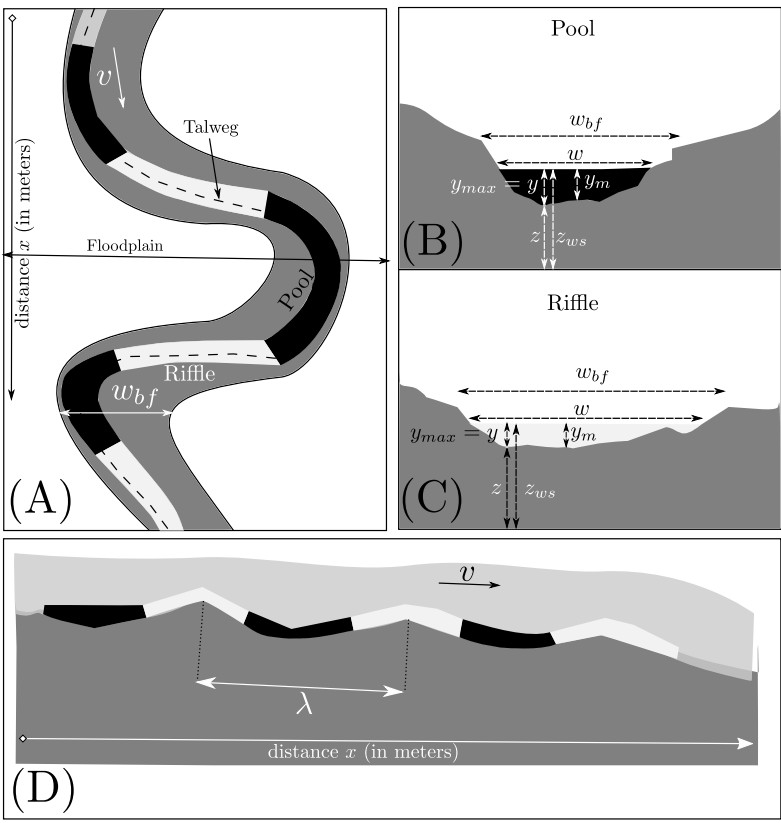

**Figure 1.** Different views of pool-riffle sequences. (A) Plan view pattern that includes bankfull width $w_{bf}$, floodplain extent, talweg line, velocity $v$, pools and riffles, and channel direction (planform); (B) cross-sectional view of a pool with a section width $w$ and a steeper water depth $y$ calculated from the talweg elevation, which is the deepest part of the bottom, and $y = y_{max} = z_{ws} - z$ with $z$ is the bed elevation and $z_{ws}$ the water surface elevation, and $y_m$ the mean water depth; (C) cross-sectional view of a riffle with a shallower water depth $y$, higher bed elevation $z$ and high bankfull width $w_{bf}$; (D) longitudinal profile that makes it possible to see the water surface, the bed slope, the pools and riffles, and the wavelength $\lambda$ calculated between two successive riffles or pools

variability of the river topography. This information can be further used to build a synthetic river such as the RiverBuilder (Pasternack and Zhang, 2020) or the channel builder for simulating river morphology of Legleiter (2014a).

Some of the methods presented in the literature have shown limits in calculating the wavelengths of pool-riffle sequences, others have given results that are often difficult to interpret in terms of bedform amplitude. This amplitude, which varies according to each bedform, involves the use of the pseudo-period. In fact, few methods are developed to extract this pseudo-period from alternating MUs rivers. We therefore choose to work with wavelet analysis that estimates the local variability strength of a signal and extract the signal amplitude and wavelength. In this study we apply continuous wavelet transform

(CWT) to calculate the wavelength $\lambda$ and the dimensionless wavelength spacing $\lambda^*$ (longitudinal spacing) which is

$$\lambda^* = \frac{\lambda}{w_{bf}} \tag{1}$$

with $w_{bf}$ is bankfull width.

**Table 1.** Review of some methods of morphological units' identification (variable used and MUs types)

| Methods | Variables | MUs | References |
|---|---|---|---|
| Control-point method | Energy gradient | Pools and riffles | Yang (1971) |
| Zero-crossing method | Bed topography | Pools and riffles | Richards (1976a); Milne (1982) |
| Areal difference asymmetry index | Cross-section area | Pools and riffles | Knighton (1981) |
| Power spectral analysis | Bed topography | Pools and riffles | Nordin (1971); Box and Jenkins (1976) |
| Bedform Differencing Technique (BDT) | Bed topography | Pools and riffles | O'Neill and Abrahams (1984) |
| Hydraulic characteristics classification | Froude number | Pools, runs, and riffles | Jowett (1993) |
| 3D identification | Water depth | Pools and riffles | Milne and Sear (1997) |
| Schweizer's method | Water depth and velocities | Pools, runs, and riffles | Schweizer et al. (2007) |
| MEM Model | Water depth, velocity, and bottom shear stress | Pool, riffle, run, fast run, shallow water, and backwater | Hauer et al. (2009, 2011) |
| Wyrick's method | Water depth and velocity | Pools, riffles, runs, and glides | Wyrick and Pasternack (2014); Wyrick et al. (2014) |
| Brown and Pasternack method | Minimum bed elevation and channel top width | Pools and riffles | Brown and Pasternack (2017) |

In reality, longitudinal spacing $\lambda^*$ has several definitions. Some authors have defined the wavelength $\lambda$ as the distance
5    between riffle crests (e.g., Harvey (1975); Hogan et al. (1986)), or the distance from the bottom of successive pools (e.g., Keller and Melhorn (1973, 1978)). Other authors have chosen channel width $w$ (e.g., Richards (1976a, b); Dury (1983)) instead of bankfull channel width $w_{bf}$ (e.g., Leopold et al. (1964)). These differences raise questions about the selection of these ratios and their dependence on geometric or hydraulic parameters. Moreover, the majority of researchers uses the average channel width instead of the bankfull width because both give a similar pool-riffle spacing interval. Here, we are working with $w_{bf}$ and with
10    a new automatic wavelength calculation method that uses the whole covariance structure of a set of hydraulically-independent variables without the need of ad hoc thresholding of these variables.

Some researchers have investigated the variability of longitudinal spacing in relation to geometric or hydraulic parameters. Rosgen (2001) developed an empirical relationship between the ratio of pool-to-pool spacing/bankfull width and the channel slope expressed as a percentage based on a negative power function of slope $S$ :

$$\lambda^* = 8.2513 S^{-0.9799} \tag{2}$$

In addition, Montgomery et al. (1995) showed that there is an influence of large woody debris (LWD) on channel morphology that leads to a relation between LWD and longitudinal spacing in a pool-riffle sequence, and found that 82% of pools were formed by LWD or other obstructions, and increased numbers of obstructions led to a decrease in pool–riffle spacing. Moreover, research has linked variation in spacing to channel characteristics including gradient (Gregory et al., 1994). Also, Harvey (1975) showed that pool–riffle spacing correlated strongly with discharges between the mean-annual flood and a 5 year recurrence

interval (Thompson, 2001). Recently, Wyrick and Pasternack (2014) measured spacing of six different morphological units using a tool in ArcGIS. Therefore, the definition of the characteristics and the measurement methods allowed us to expect some variation from one study to another in the estimated relationship between longitudinal spacing and bankfull width (Richards, 1976a; O'Neill and Abrahams, 1984; Gregory et al., 1994; Knighton, 1998). Aside from the interval $[5w_{bf}, 7w_{bf}]$ defined by Leopold et al. (1964) and the interval $[2w_{bf}, 4w_{bf}]$ defined by Montgomery et al. (1995) in forested streams, other values of the

longitudinal spacing exist, such as the Carling and Orr (2000) interval, which is $[3w_{bf}, 7.5w_{bf}]$ and decreases to $[3w_{bf}, 6w_{bf}]$ as sinuosity increases (Clifford, 1993; Carling and Orr, 2000).

## 1.2   Study objectives

The studies that used wavelet analysis in the geomorphological field consist in extracting components of a given spatial series (e.g., $w(x)$, $v(x)$), but they are not specifically designed to identify pseudo-periodic components in a univariate, let alone in a

multivariate case. For this reason, we introduce an automatic procedure called Wavelet Ridge Extraction defined by Lilly and Olhede (2011) and used in this study to extract the longitudinal spacing of the alternating MUs.

The objective is to extract some quantitative properties of these alternating morphological units such as the mean and the median of their longitudinal spacing, with a continuous vision of the topography instead of a discrete classification. This will be done by focusing on two numerical criteria computed at reach scale : the distribution of spacings between morphological

units (mean, median, etc.) and the evaluation of correlations between all geometrical and flow variables. This work will be done on four classical variables (velocity, hydraulic radius, bottom shear stress, and the local channel direction angle) because they respond directly to morphodynamic processes (flow convergence routing or meander migration) and they are independent hydraulic degrees of freedom.

In this study, we first present the dataset of six river reaches in France used for this analysis (section 2). In section 3, we

present the Wavelet Ridge Extraction method to identify pool-riffle sequences in the univariate and multivariate cases with the four variables. Section 4 presents results and compare them with the bedform differencing technique (BDT) developed by O'Neill and Abrahams (1984) to determine if they yield the same results in terms of spacing. We choose this method instead

of threshold methods because the latter require ad hoc thresholding / parameter range definition from independent calibration data, which was not possible in our case.

## 2 Data set and study reaches

Six reaches of small French rivers are used in this study (Navratil, 2005; Navratil et al., 2006) : the Graulade at St Sylvain Montaigut (1), the Semme at Droux (2), the Olivet at Beaumont Village (3), the Ozanne at Tirzay lès Bonneval (4), the Avenelles at Boissy-le-Châtel Les Avenelles (5), and the Orgeval at Boissy-le-Châtel Le Theil (6) (Fig. 2). These reaches contain mainly pool-riffle sequences, they have slopes between 0.002 and 0.013 m.m$^{-1}$ (estimated from the talweg elevation which is the lowest point in the section), mobile gravel beds, stable banks, and well-defined floodplains along at least one side of the channel (Navratil et al., 2006). These reaches are located in the Loire River Basin (four reaches) and the Seine River Basin (two reaches), and their length ranges from 155 to 495 m (Table 2). All reaches are located at or near the stream gauging stations of the French national hydrometric network. Long-term (about 20 years) hydrological records are available for most reaches. The bankfull widths vary from 4 to 12 m, with an average value of about 9 m.

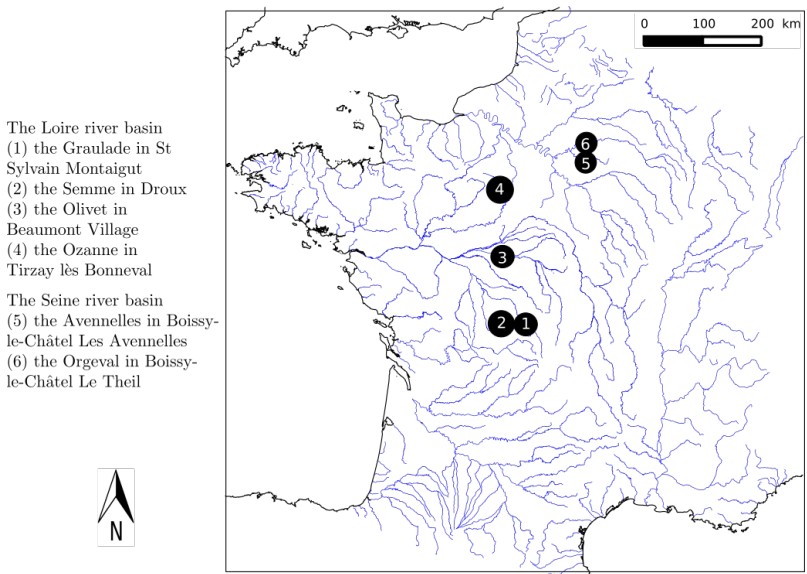

**Figure 2.** Location of the study reaches in France.

Cross-sections were surveyed along the river reaches at the level of hydraulic controls and morphological breaks in order to describe the major variations in terms of width, height, and slope in the main channel and the floodplain and at the level of pool-riffle sequences. Cross-sections and water surface profile measurements were surveyed in 2002 – 2004 covering the main channel and floodplain and using an electronic, digital, total-station theodolite. Water surface profiles were measured at different flow discharges (Navratil et al., 2006). Using this dataset, we solely rely on measurements at the lowest surveyed

discharge in the development of the method because it is the discharge through which we can visualize the variability of the bathymetry (alternating morphological units). We select four spatial series :

1. velocity $v(x)$;

2. hydraulic radius $R_h(x) = \frac{A(x)}{P(x)} \approx \frac{A(x)}{w(x)}$ with $A(x)$ is the cross-sectional area et $P(x)$ is the wetted perimeter;

3. bed shear stress $\tau_b(x) = (\rho g) n^2 v(x)^2 R_h(x)^{-1/3}$ with $\rho$ : water density ($1000 \text{ kg.m}^{-3}$) and $n$ is the Manning's roughness coefficient;

4. local channel direction angle (planform) $\theta(x)$.

All descriptors are derived from in-situ observations taken from Navratil et al. (2006), except the calibrated estimates of Manning's roughness coefficient $n$. These values were estimated by Navratil et al. (2006) using a one-dimensional open channel

steady and step backwater model FLUVIA (Baume and Poirson, 1984). However, we will use these values in order to compute the bed shear stress $\tau_b(x)$, along the reach : even if partly relies on calibration, it is a more robust way of computing $\tau_b$ here than through the finite differentiation of the total head function $\frac{v(x)^2}{2g} + z(x)$ between adjacent cross-sections to get the energy slope $J$, given the typical number and spacing of surveyed cross-sections for each reach in the dataset.

**Table 2.** Characteristics of the six river reaches and their catchment. The bankfull width $w_{bf}$ is taken from the study of Navratil et al. (2006), and the average width $w_m$, the standard deviation $\sigma(w)$ are calculated for minimum discharge $Q_{min}$

| Reach | 1: Graulade | 2: Semme | 3: Olivet | 4: Ozanne | 5: Avenelles | 6: Orgeval |
|---|---|---|---|---|---|---|
| Reach length $L$ (m) | 160 | 177 | 495 | 319 | 155 | 318 |
| Number of cross sections | 14 | 32 | 66 | 26 | 25 | 36 |
| Reach gradient $S$ (m.m$^{-1}$) | 0.0125 | 0.0044 | 0.0018 | 0.0024 | 0.0060 | 0.0047 |
| Bankfull width $w_{bf}$ (m) | 4 | 12 | 6 | 12 | 9 | 10 |
| Average width $w_m$ (m) | 2.8 | 9.3 | 4.7 | 7.0 | 3.3 | 6.1 |
| Standard deviation $\sigma(w)$ (m) | 0.4 | 1.9 | 0.9 | 1.1 | 0.9 | 1.0 |
| Surveyed flow discharges (m$^3$.s$^{-1}$) | 0.22 and 1.26 | 1.85 and 2.41 | 0.18; 1.13; 1.72 and 1.99 | 0.19; 0.33; 0.8 and 11.5 | 0.15 | 0.21 |
| Min discharge $Q_{min}$ (m$^3$.s$^{-1}$) | 0.22 | 1.85 | 0.18 | 0.19 | 0.15 | 0.21 |

The fourth variable chosen is related to the channel planform : we define $\theta(x)$ as the local angular deviation of the channel

direction from a lower-frequency curve. There are many possible definitions of this low-frequency behavior, such as parametric splines or Bezier curves; in order to avoid over-parameterization, we define this low-frequency planform as a constant curvature curve, i.e., the best-fitting arc-circle (Fig. 3), a choice suitable for all six reaches studied. Since $\theta$ is signed, it is expected to have a pseudo-periodicity which is approximately twice slower as other 1D variables : indeed, a large positive value of $\theta$ indicates a counterclockwise deviation from the low-frequency direction, while a large negative value of same amplitude indicates

a clockwise deviation. From a hydraulic perspective, both deviations have the same effect since they are symmetrical with respect to the low-frequency direction. For this reason, which chose to analyze the variable $\cos(\theta(x))$.

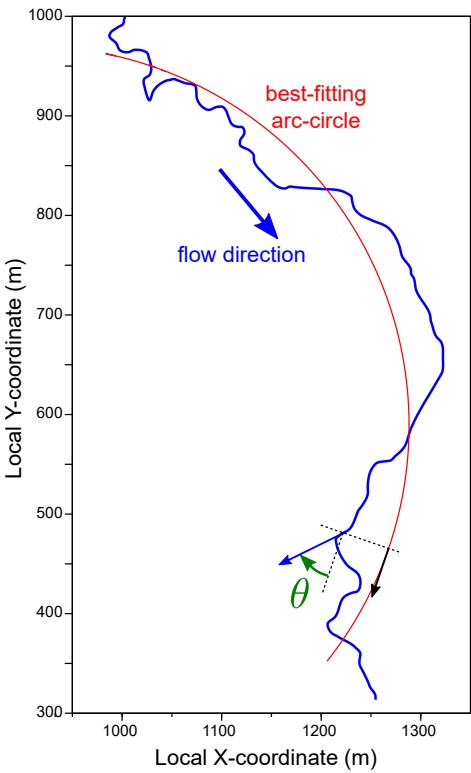

**Figure 3.** Definition of $\theta$, the local angular deviation of the channel direction from a lower-frequency behavior. Here this low-frequency planform is defined as an arc-circle (illustration on the Olivet River reach). It is worth noting that $\theta$ is signed : at the location pointed on the figure, $\theta$ is negative.

## 3  Wavelet method

Classical mathematical methods, such as Fourier analysis, extract the wavelengths in the frequency domain for stationary signals, but can also be used for nonstationary signals using an "evolutive" methodology based on spectral estimators (Thomson, 1982; Pasternack and Hinnov, 2003). Wavelet transform standardly does the same for nonstationary signals : analyzing a signal basically consists in looking for local similarity between the signal and a given waveform (the wavelet). In this paper, we use the continuous wavelet transform with the Morlet wavelet (Gabor, 1946) (Fig. 4) applied to spatial series instead of time series, so periods and frequencies in time series are replaced by wavelengths (in m) and wavenumbers (in rad.m$^{-1}$). The choice of the Morlet wavelet is justified by the analytical properties in its derivation and its flexibility due to the exponential form (see Appendix B).

The wavelet transform uses a whole family of "daughter" wavelets generated by scaling and translating the mother wavelet $\psi$; the value of the transform at location $x$ and scale $s$ is the scalar product of the signal and this daughter wavelet $\psi_{s,x}$.

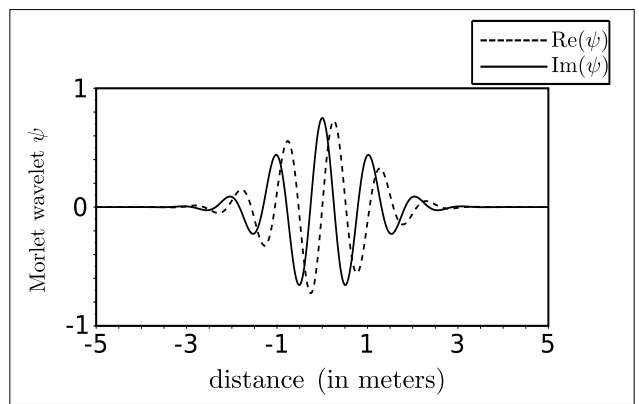

**Figure 4.** Wavelet Morlet mother function, the plot gives the real part and the imaginary part of the wavelets in the space domain (distance).

Wavelet analysis is very popular in many fields such as fluid mechanics (e.g., Schneider and Vasilyev (2010); Higuchi et al. (1994); Katul et al. (1994); Katul and Parlange (1995b, a)), meteorology (e.g., Kumar and Foufoula-Georgiou (1993); Kumar (1996)), geophysics (e.g., Ng and Chan (2012); Grinsted et al. (2004)), hydrology (e.g., Rossi et al. (2011); Schaefli et al. (2007); Nourani et al. (2014)), and geomorphology (Gangodagamage et al., 2007; Lashermes and Foufoula-Georgiou, 2007; McKean et al., 2009). In the literature of the alternating bedforms identification, McKean et al. (2009) used Derivative of a Gaussian wavelets (DOG) of order 6 to investigate the spatial patterns (pools and riffles) of channel morphology and salmon spawning using a one-dimensional elevation profile of the channel bed morphology.

In this study, we use another application of the wavelet analysis called the wavelet ridge extraction method (Mallat, 1999; Lilly and Olhede, 2010). This analysis is based on the existence of special space/wavenumber curves, called wavelet ridge curves or simply ridges (Lilly and Olhede, 2010), where the signal concentrates most of its energy (Carmona et al., 1999; Ozkurt and Savaci, 2005). Along such a curve, the signal can be approximated by a single component modulated both in amplitude and frequency. So, the rationale behind the method is that the existence of alternating morphological units along a reach (such as pools-riffles sequences) could be translated into a pseudo-periodicity in geometric and flow variables. Hence, identifying these bedforms requires to identify a local wavenumber $K(x)$ and phase $\Phi(x)$ for each variable, a task that can be performed by wavelet analysis and especially Wavelet Ridge Extraction (Mallat, 1999; Lilly and Olhede, 2010).

### 3.1 Wavelet analysis and ridge extraction

Few methods in the literature have been trying to identify river characteristics with wavelets. For example, Gangodagamage et al. (2007) used Wavelet Transform Modulus Maxima (WTMM, Muzy et al. (1993)) in a fractal analysis to extract multiscale

statistical properties of a corridor width. Procedures such as the WTMM consist in extracting components of the signal, but they are not specifically designed to identify pseudo-periodic components in a univariate, let alone in a multivariate case.

In the present study, we tested a new wavelet ridge analysis on spatial series with the Morlet mother basis function represented in Fig. 4. Its expression is :

$$\psi(\eta) = \pi^{-\frac{1}{4}} e^{i\beta\eta} e^{-\frac{\eta^2}{2}} \tag{3}$$

With $\psi$ is the mother wavelet function that depends on the dimensionless "position" parameter $\eta$ and $\beta$ is the dimensionless frequency, here taken to be 6 as recommended by Torrence and Compo (1998). Starting with this wavelet mother, a family $\psi_{s,x}$ called wavelet daughters is obtained by translating and scaling $\psi$.

$$\psi_{s,x}(\eta) = \frac{1}{\sqrt{s}} \psi\left(\frac{\eta - x}{s}\right), \quad x \in \mathbb{R}, \ s > 0 \tag{4}$$

With $x$ is the translation offset, which represents a position at which the signal is analyzed, and $s$ the dilation or scale factor. If $s > 1$, the daughter wavelet has a frequency lower than the mother wavelet, whereas if $s < 1$, a wavelet with a frequency higher than the mother wavelet is generated.

Given a spatial series $f(\eta)$, its continuous wavelet transform $\mathcal{W}[f](x,s)$ with respect to the wavelet $\psi$ is a function of two variables where :

$$\mathcal{W}[f] : \mathbb{R} \times \mathbb{R}_+^* \quad \rightarrow \quad \mathbb{C}$$

$$(x,s) \quad \mapsto \quad \frac{1}{\sqrt{s}} \int_{-\infty}^{+\infty} f(\eta)\psi^*\left(\frac{\eta-x}{s}\right) d\eta \tag{5}$$

(*) indicates the complex conjugate. This complex function can also be written as :

$$\mathcal{W}[f](x,s) = R(x,s)e^{i\phi(x,s)} \tag{6}$$

where $R$ is the absolute value (modulus) and $\phi$ the phase (argument) at position $x$ with the scale $s$.

$$R(x,s) = \left|\mathcal{W}[f](x,s)\right| \tag{7}$$

$$\phi(x,s) = \text{Im}\left(\ln \mathcal{W}[f(x)](x,s)\right) \tag{8}$$

To respect the nomenclature in the spatial definition and facilitate the extraction of wavelengths, we choose the angular wavenumber (in rad.m$^{-1}$) $k = \dfrac{2\pi}{\lambda}$ instead of the scale factor. We associate a wavelength $\lambda = 2\pi\alpha s$ with the scale parameter $s$, where $\alpha$ is the Fourier factor associated with the wavelet, and

$$\alpha = \frac{2}{\beta + \sqrt{2 + \beta^2}} \tag{9}$$

$$s = \frac{1}{\alpha k} \tag{10}$$

Thus, the wavelet transform of the function $f(x)$ is defined in the space-wavenumber as :

$$
\mathcal{W}[f] : \mathbb{R} \times \mathbb{R}_+^* \quad \rightarrow \quad \mathbb{C}
$$

$$
(x,k) \quad \mapsto \quad \sqrt{\alpha k} \int_{-\infty}^{+\infty} f(\eta)\psi^*\Big(\alpha k(\eta - x)\Big)d\eta \tag{11}
$$

Except for the channel angle, all input variables are always positive and may substantially vary in magnitude so we per-
form the wavelet transform on the Neperian logarithm of these variables. The whole analysis is performed in a simple *Scilab*
script, using the functions that compute the wavelet transform $\mathcal{W}[f]$. They were provided by Torrence and Compo (1998)
[*atoc.colorado.edu/research/wavelets/*]. To extract the wavelength, the procedure in Appendix B is followed to compute $\dfrac{\partial\phi}{\partial x}$
and extract the curves that satisfy Eq. 12 and 13.

The complex wavelet transform can be classically visualized using a scalogram, i.e., a colored map of the modulus $R(x,k)$
in the $(x,k)$ plane (Fig. 5 bottom). The wavelet analysis neglects parts of the signal at both extremities of the series : this is
*the cone of influence* (Torrence and Compo, 1998) that is the region of the wavelet spectrum in which edge effects become
important. However, as explained previously, the complex transform also yields a phase $\phi(x,k)$ in rad (Eq. 8) which can also
be plotted in the same plane (Fig. 5 top). In our study, we will search for space/wavenumber curves mainly using the phase
information, i.e., search for phase ridges as opposed to amplitude ridges (Lilly and Olhede, 2010).

In section 3.2, we give a rigorous definition of Wavelet Ridge points and curves in a univariate case (i.e., a single spatial series).
Then, in section 3.3, we generalize the definition to the multivariate i.e., when the series consists in several correlated variables.

## 3.2 Univariate case

In the univariate case, we choose a single variable $f$ (velocity, hydraulic radius, bed shear stress, or or local channel direction
angle). For the wavelet $\psi(\eta)$, the ridge point of $\mathcal{W}[f](x,k)$ is a space/wavenumber pair $(x,k)$ satisfying the *phase ridge point
conditions* (Lilly and Olhede, 2010) :

$$
\frac{\partial}{\partial x}\text{Im}\Big(\ln\mathcal{W}[f(x)](x,k)\Big) - k = 0 \tag{12}
$$

or, according to the definition of the phase (Eq. 8) :

$$
\left.\frac{\partial\phi}{\partial x}\right|_{(x,k)} - k = 0 \tag{13}
$$

This condition states that the rate of change of transform phase at scale $k$ exactly matches $k$ at location $x$; from this condition,
the instantaneous frequency of the signal can be derived (Lilly and Olhede, 2008, 2010).The sets of points satisfying the
condition form a parametric curve (ridge curve) noted $(x, K(x))$ implicitly defined by :

$$
\left.\frac{\partial\phi}{\partial x}\right|_{(x,K(x))} - K(x) = 0 \tag{14}
$$

This property is illustrated in Fig. 5, where a ridge curve is superposed both on the scalogram and on the phase map.

There may be several curves that verify Eq. 14; in practice we choose curves that cross continuously the domain of the wavelet

transform (from one cone of influence to another) and belong to the region where a maximum power of the wavelet is. This curve $K(x)$ also represents the local wavenumber, which is defined on a support $\ell < L$ named assessed length, with $L$ the total reach length.

The phase function $\Phi$ is then obtained by evaluating the function $\phi(x,k)$ along the curve $(x, K(x))$, in thick black in Fig. B1 (A) in Appendix B-1.

$$\Phi(x) = \phi(x, K(x)) \tag{15}$$

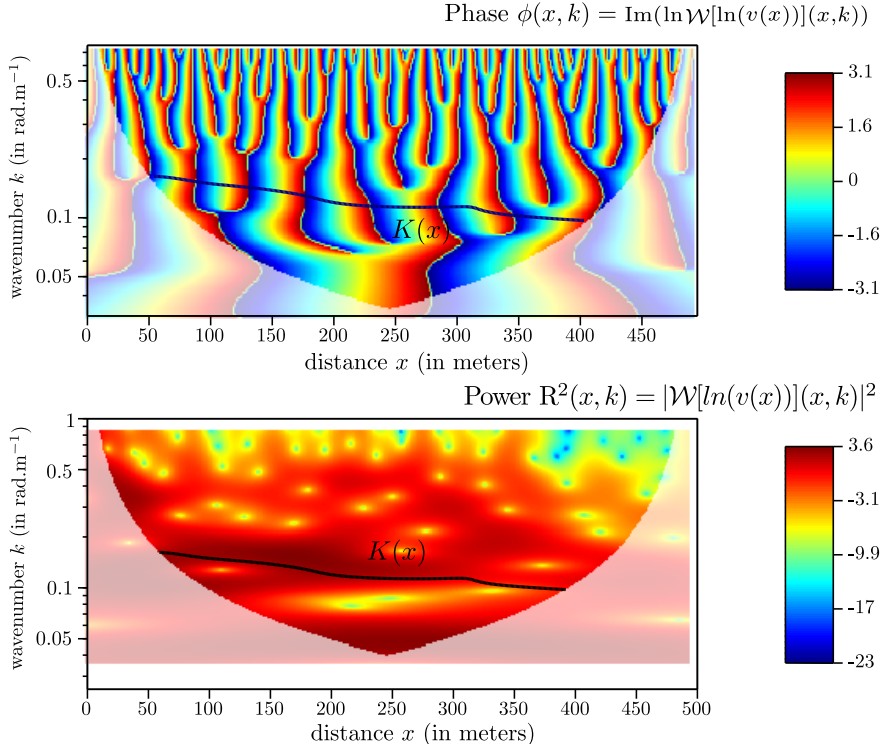

**Figure 5.** Top plot : the phase function from which we get the function $K(x)$; bottom plot : the power of the wavelet with the region where there is maximum variability depicted by the black curve $K(x)$ (ridge curve). These two figures are represented in a wavenumber/distance space for the Olivet River and the wavelet transform is performed on the logarithm of the velocity. The part of the figure with low opacity shows the cone of influence which is neglected in this study (edge effects are more important for short wavelengths than for long wavelengths).

In the end, we can extract the wavelength function of pool-riffle sequences, which corresponds to a pseudo-period function of the signal $f$, and which is :

$$\lambda(x) = \frac{2\pi}{K(x)} \tag{16}$$

Also, the shape's amplitude $A_m$, with which pools and riffles vary, is corrected by a coefficient $\sqrt{\dfrac{1}{\alpha K(x)}}$. This correction comes from the inversion of the direct transformation equation (Eq. 11) which holds the coefficient $\sqrt{\alpha K(x)}$.

$$A_m(x) = \left| \mathcal{W}[f](x, K(x)) \right| \sqrt{\frac{1}{\alpha K(x)}} = R(x, K(x)) \sqrt{\frac{1}{\alpha K(x)}} \tag{17}$$

The signal is locally similar to a sinusoid $f_{mod}$ of wavenumber $K$ in rad.m$^{-1}$ which model the variability $f$. We can define the pseudo-periodic variable as presented in the Fig. 6 with :

$$f_{mod}(x) = A_m(x)\cos(\Phi(x)) = A_m(x)\cos(\phi(x, K(x))) \tag{18}$$

In the example below (Fig. 6), the modeled velocity function follows the variability of the observed velocity, it is a pseudo-periodic, continuous function that approximates the first-order variability of this hydraulic parameter across pool-riffle sequences. The statistics of the $K(x)$ function can be translated into statistics of longitudinal spacings of alternating bed-forms, e.g., mean spacing $\lambda^*_{mean}$, median spacing $\lambda^*_{median}$ or spacing standard deviation $\sigma(\lambda^*)$. In Fig. 6 we would find $\lambda^*_{mean} \approx 8.7$, $\lambda^*_{median} \approx 9.12$, and $\sigma(\lambda^*) \approx 0.79$ if we were to analyze velocity only; The pseudo-periodicity of $v_{mod}$ yields to the identification of 6 pools (white) and 7 riffles (gray).

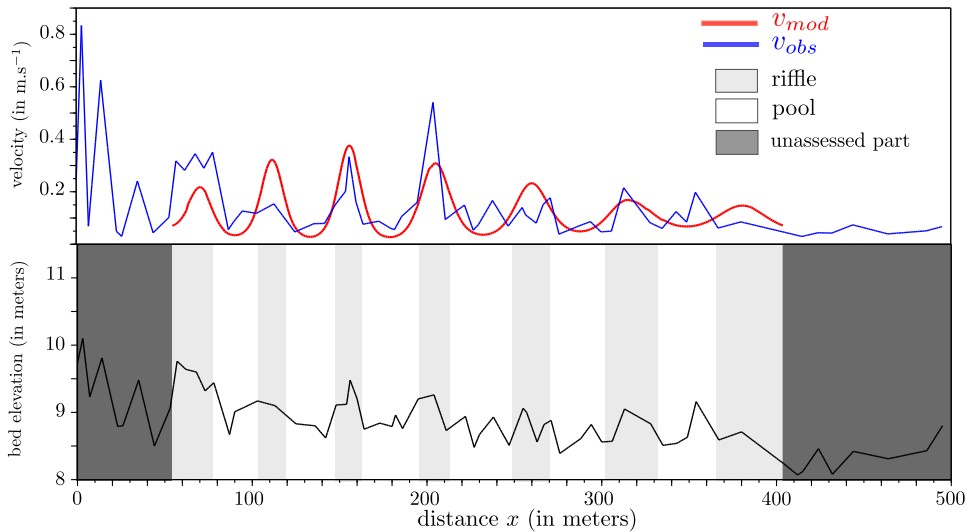

**Figure 6.** variation of the modeled function $f_{mod}$ which represent the pseudo-periodic variable (e.g., the velocity of the Olivet River) compared to the observed one. This pseudo-periodicity yields to the identification of pools (white) and riffles (gray) in the plot below. The not studied part is due to the cone of influence of the wavelet method.

In the next section, we will extend the definition of phase ridge points and ridges to the case where several variables are sampled along the reach, all of them potentially correlated and embedding information about the pseudo-periodicity of channel hydraulic behavior.

### 3.3 Multivariate case

The multivariate case is the extension of the univariate to a set of $N$ real-valued signals, we use the coevolution of more than one variable to extract the wavelength of the reach and therefore identify the pool-riffle sequences. We start by computing the wavelet transform for each variable $i = 1...N$ and extract their phase functions $\phi_i(x, k)$. According to the previous section, univariate ridges curves $K_i(x)$ would be defined by :

$$\left. \frac{\partial \phi_i}{\partial x} \right|_{(x, K_i(x))} - K_i(x) = 0 \tag{19}$$

But then the local wavenumber would be specific to a given variable. Otherwise, the multivariate case requires to determine a common wavenumber between all the variables such that :

$$\left. \frac{\partial \phi_i}{\partial x} \right|_{(x, K(x))} - K(x) \approx 0 \qquad \forall i \tag{20}$$

The identification of a "master" ridge point/curve is now a minimization problem. We will define it as a local minimum of the squared norm of the vector $\left( \left. \frac{\partial \phi_1}{\partial x} \right|_{(x,k)} - k, \left. \frac{\partial \phi_2}{\partial x} \right|_{(x,k)} - k, ..., \left. \frac{\partial \phi_N}{\partial x} \right|_{(x,k)} - k \right)$ :

$$E(x, k) = \sum_{i=1}^{N} \left( \left. \frac{\partial \phi_i}{\partial x} \right|_{(x,k)} - k \right)^2 \tag{21}$$

This minimum is calculated by searching for the wavenumbers and positions where the derivatives (Eq. 22) of this quantity satisfies these two conditions bellow :

$$\frac{\partial E(x, k)}{\partial k} = \sum_{i=1}^{N} \left( \left. \frac{\partial^2 \phi_i}{\partial k \partial x} \right|_{(x,k)} - 1 \right) \left( \left. \frac{\partial \phi_i}{\partial x} \right|_{(x,k)} - k \right) = 0$$

$$\frac{\partial^2 E(x, k)}{\partial k^2} = \sum_{i=1}^{N} \left[ \left. \frac{\partial^3 \phi_i}{\partial k^2 \partial x} \right|_{(x,k)} \left( \left. \frac{\partial \phi_i}{\partial x} \right|_{(x,k)} - k \right) + \left( \left. \frac{\partial^2 \phi_i}{\partial k \partial x} \right|_{(x,k)} - 1 \right)^2 \right] > 0 \tag{22}$$

The procedure is applied to a set of variables $[v, R_h, \tau_b, \theta]$ and the goal is to seek for the common wavenumber between all these variables. In the Fig. 7 we illustrate the result of this procedure applied on the Olivet River for all the four variables. A unique wavenumber is extracted which represents a co-evolution of all these variables.

As a result, the phase shift of every variable is calculated by :

$$\Phi_i(x) = \phi_i(x, K(x)) \tag{23}$$

This ridge curve $K(x)$ is common between all variables, yet $\Phi_i$ varies according to each variable. Therefore, each one can be represented as a pseudo-periodic function $f_{i,mod}$ with the pair $(K(x), \Phi_i(x))$.

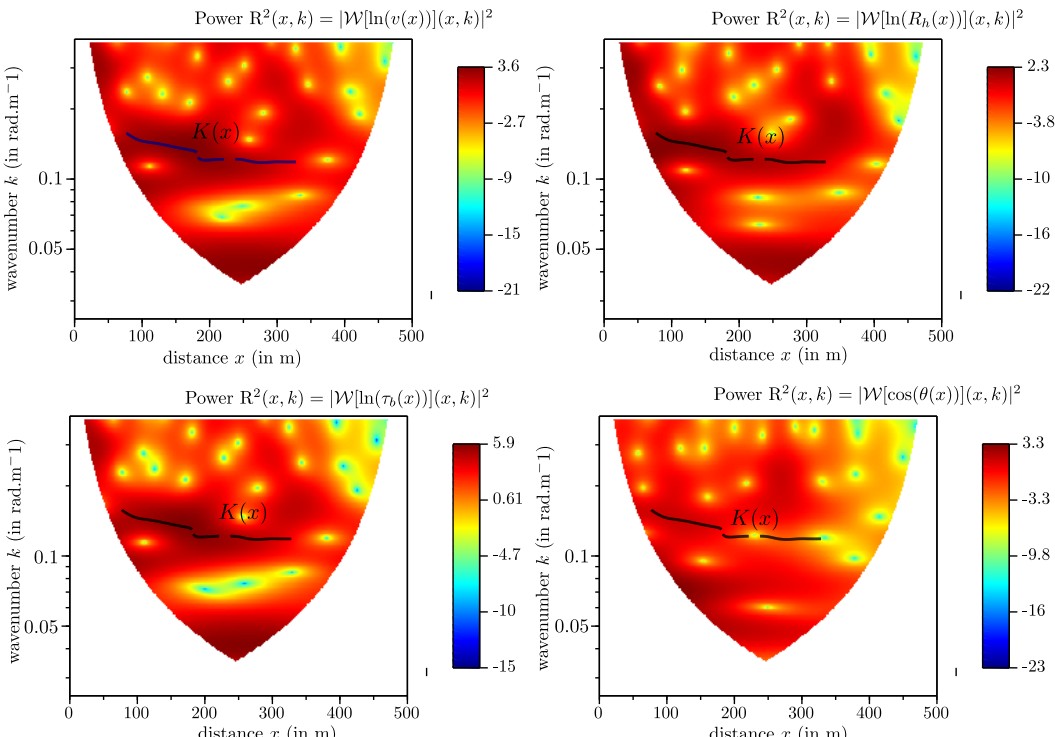

**Figure 7.** Power of the wavelet of the four variables : velocity, hydraulic radius, bed shear stress, and local channel direction angle. The black curve $K(x)$ is the extracted ridge curve of the Olivet River in the multivariate case.

In our case, after calculating the phase and amplitude, we modelled each variable as in the Eq. 24 and represented them in Fig. 8.

$$f_{i,mod}(x) = A_{i,m}(x)\cos(\Phi_i(x)) = A_{i,m}(x)\cos(\phi_i(x, K(x))) \tag{24}$$

The amplitude shape of the modeled variable is calculated by the same way in the univariate case :

$$5 \quad A_{i,m}(x) = |\mathcal{W}[f_i](x, K(x))|\sqrt{\frac{1}{\alpha K(x)}} \tag{25}$$

The results in Fig. 8 show that a common pseudo-period has been successfully identified and allows a consistent pseudo-periodic representation of all four variables.

Fig. 9 shows the correlations between these variables which are well respected between the three flow variables; an anti-correlated hydraulic radius with bed shear stress and velocity and a strong correlation between bed shear stress and velocity.

10 However, with regard to the angle, the results show a small phase shift which is corrected in the following $x$ positions. But generally a deviation (clockwise or counterclockwise) from the average direction of the channel (i.e., $\cos(\theta)$ much smaller than 1) is associated with a low hydraulic radius and large values of $\tau_b$ and $v$, a consistent characterization of a riffle. This gives us

an identification reach features : pools (in white) and riffles (in grey).

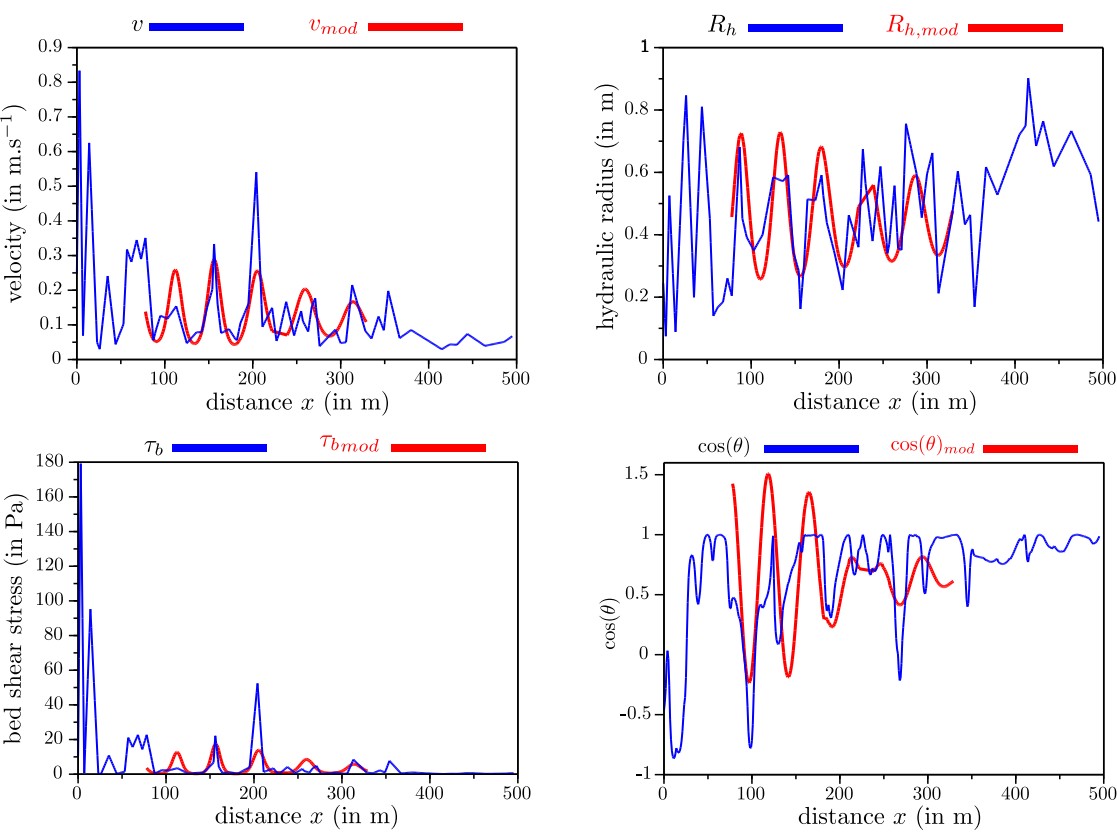

**Figure 8.** Variation of the modeled function $f_{i,mod}$ which represent the pseudo-periodic variable (in red) for the velocity, the hydraulic radius, the bed shear stress, and the local channel direction angle of the Olivet River compared to the observed ones (in blue).

As already mentioned in section 3.2 (univariate analysis), the statistics of the $K(x)$ function can be translated into statistics of local wavelength $\lambda(x) = \dfrac{2\pi}{K(x)}$, which can in turn be interpreted as statistics of longitudinal spacings of alternating bedforms, e.g., mean spacing $\lambda^*_{mean}$, median spacing $\lambda^*_{median}$ or spacing standard deviation $\sigma(\lambda^*)$. In the example of the Olivet river (Fig. 9) $\lambda^*_{mean} \approx 8.16$, $\lambda^*_{median} \approx 8.62$, and $\sigma(\lambda^*) \approx 0.70$. The pseudo-periodicity of the set $[v_{mod}, R_{h,mod}, \tau_{b,mod}, \cos(\theta)_{mod}]$ yields to the identification of 5 pools and 5 riffles.

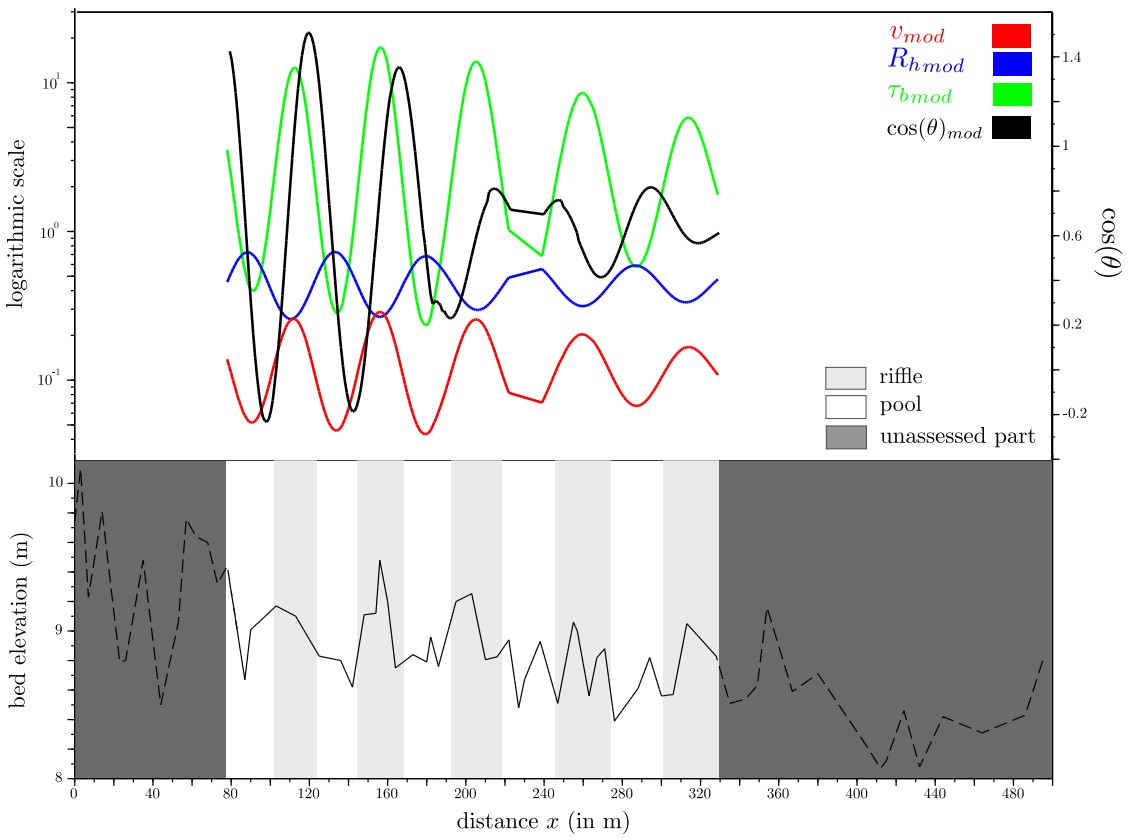

**Figure 9.** Correlation between the modeled functions $f_{i,mod}$ which represent the pseudo-periodic variables (velocity in red, hydraulic radius in blue, bed shear stress in green, and local channel direction angle in black) of the Olivet river.

## 4    Results

In this section, we present the results of the analysis on the six reaches presented in section 2. We present the comparison between the univariate and the multivariate approaches and a comparison of the multivariate with the benchmark method. The methods are compared in terms of the statistics (mean, median, etc.) they yield. Second, we present the benchmark method called BDT (Bedform differencing technique) and compare their results of the six reaches with the multivariate case.

### 4.1    Univariate vs Multivariate

First, both approaches were employed on all reaches to extract statistics such as the mean, mode and standard deviation wavelengths of morphological units (pool-riffle sequences). The wavelet method extracts the wavelength for an assessed length $\ell$ (which is the $K(x)$ support in Fig. 6 and 9) that is generally small compared to the total length of the reach. Consequently, we have results that are valuable only for the lengths shown in Table 3. In this table, we give the values of these lengths for each approach and with the variables used in it. These values generally depend on the number of alternating bed forms and

also on the total length of the reach. The greater the number of alternating bed forms and the reach length are, the greater the assessed length is.

Moreover, the multivariate approach takes into account all the variables and therefore looks for a single pseudo-periodicity between the four variables and then we are going to have a pseudo-periodicity that represents the reach and not the chosen variable.

**Table 3.** Assessed length by the wavelet analysis for all reaches in the univariate case using the velocity, hydraulic radius, bed shear stress, or local channel direction angle and in the multivariate case using all these four variables.

| Reaches | Reach length (m) | Assessed length $\ell$ (m) (Univariate) | | | | Assessed length $\ell$ (m) (Multivariate) |
|---|---|---|---|---|---|---|
| | | Velocity | Hydraulic radius | Bed shear stress | Local channel direction angle | $[v, R_h, \tau_b, \cos(\theta)]$ |
| **1: Graulade** | 160 | 88 | 67 | 72 | 102 | 67 |
| **2: Semme** | 177 | 87 | 70 | 89 | 110 | 37 |
| **3: Olivet** | 495 | 349 | 366 | 363 | 365 | 251 |
| **4: Ozanne** | 319 | 215 | 157 | 151 | 125 | 77 |
| **5: Avenelles** | 155 | 76 | 70 | 79 | 64 | 60 |
| **6: Orgeval** | 318 | 142 | 200 | 163 | 140 | 158 |

Table 4 gives some statistics on both approaches. Longitudinal spacing is calculated using the wavelengths extracted automatically by the wavelet ridge method from $K(x)$.

We compare the methods in terms of longitudinal spacing ($\lambda^*$). In each reach, there seems to be one variable which drives the wavelength identified in the multivariate approach :

  – in the Graulade River, the longitudinal spacing identified using the multivariate approach matches closely the one associated with the hydraulic radius (in the mean and the median with deviation of $0.05w_{bf}$) and also with the local channel direction angle (in the median with a deviation of $0.06w_{bf}$);

  – in the Semme River, it matches those of the local channel direction angle (in the mean and the median with a deviation of $0.14w_{bf}$ and $0.12w_{bf}$ consecutively);

  – in the Olivet River, it matches the bed shear stress (in the mean with a deviation of $0.25w_{bf}$) and the velocity (in the median with a deviation of $0.5w_{bf}$);

  – in the Ozanne River, it matches those of the hydraulic radius and the velocity (in the mean and the median with a deviation less than $0.6w_{bf}$);

  – in the Avenelles, it matches those of the velocity, hydraulic radius, and the bed shear stress (in the mean with a deviation
20
    less than $0.15w_{bf}$);

- in the Orgeval River, it matches those of the hydraulic radius (in the mean with a deviation of $0.28w_{bf}$ and the median with $0.06w_{bf}$) and also with the local channel direction angle (in the mean with a deviation of $0.23w_{bf}$ and in the median with $0.11w_{bf}$).

**Table 4.** Summary of results for all reaches in the univariate case using the velocity, hydraulic radius, bed shear stress, or local channel direction angle and in the multivariate case using all these four variables. For each variable we compute the mean, median, and the standard deviation $\sigma$ of the wavelength and the longitudinal spacing. This one $\lambda^*$ is calculated by $\frac{\lambda}{w_{bf}}$ , and $w_{bf}$ is taken from Table 2.

| | | | | 1: Graulade | 2: Semme | 3: Olivet | 4: Ozanne | 5: Avenelles | 6: Orgeval |
|---|---|---|---|---|---|---|---|---|---|
| Univariate | Velocity | $\lambda$(m) | Mean | 23.47 | 13.86 | 52.20 | 37.18 | 27.32 | 51.66 |
| | | | median | 24.17 | 13.93 | 54.74 | 36.74 | 27.17 | 51.29 |
| | | | $\sigma(\lambda)$ | 1.69 | 0.53 | 7.75 | 2.97 | 1.60 | 1.70 |
| | | $\lambda^*$(m) | Mean | 5.87 | 1.15 | 8.70 | **3.10** | **3.03** | 5.17 |
| | | | median | 6.04 | 1.16 | **9.12** | **3.06** | **3.02** | 5.13 |
| | | | $\sigma(\lambda^*)$ | 0.42 | 0.04 | 1.29 | 0.25 | 0.18 | 0.17 |
| | Hydraulic radius | $\lambda$(m) | Mean | 21.74 | 39.28 | 47.19 | 37.73 | 25.72 | 45.46 |
| | | | median | 21.41 | 39.43 | 46.60 | 38.47 | 25.47 | 48.23 |
| | | | $\sigma(\lambda)$ | 0.71 | 1.19 | 4.74 | 2.40 | 0.66 | 8.73 |
| | | $\lambda^*$(m) | Mean | **5.43** | 3.27 | 7.86 | **3.14** | **2.86** | **4.55** |
| | | | median | **5.35** | 3.29 | 7.76 | **3.20** | **2.83** | **4.82** |
| | | | $\sigma(\lambda^*)$ | 0.18 | 0.10 | 0.79 | 0.20 | 0.07 | 0.87 |
| | Bed shear stress | $\lambda$(m) | Mean | 26.07 | 32.29 | 47.47 | 36.43 | 27.47 | 51.70 |
| | | | median | 25.92 | 32.66 | 45.54 | 36.30 | 27.95 | 51.26 |
| | | | $\sigma(\lambda)$ | 1.12 | 1.68 | 5.36 | 1.70 | 0.73 | 1.54 |
| | | $\lambda^*$(m) | Mean | 6.52 | 2.69 | **7.91** | 3.04 | **3.05** | 5.17 |
| | | | median | 6.48 | 2.72 | 7.59 | 3.02 | **3.11** | 5.13 |
| | | | $\sigma(\lambda^*)$ | 0.28 | 0.14 | 0.89 | 0.14 | 0.09 | 0.15 |
| | Cosine of local channel direction angle | $\lambda$(m) | Mean | 21.14 | 23.45 | 40.87 | 66.31 | 28.79 | 50.58 |
| | | | median | 21.32 | 23.30 | 39.44 | 62.98 | 28.73 | 49.93 |
| | | | $\sigma(\lambda)$ | 0.75 | 0.95 | 3.57 | 7.49 | 1.47 | 4.35 |
| | | $\lambda^*$(m) | Mean | 5.28 | **1.95** | 6.81 | 5.52 | 3.20 | **5.06** |
| | | | median | **5.33** | **1.94** | 6.57 | 5.25 | 3.19 | **4.99** |
| | | | $\sigma(\lambda^*)$ | 0.19 | 0.08 | 0.60 | 0.62 | 0.16 | 0.43 |

| | | | | 1: Graulade | 2: Semme | 3: Olivet | 4: Ozanne | 5: Avenelles | 6: Orgeval |
|---|---|---|---|---|---|---|---|---|---|
| **Multivariate** | $[v, R_h, \tau_b, \cos(\theta)]$ | $\lambda$(m) | Mean | 21.54 | 21.74 | 48.98 | 43.89 | 26.59 | 48.29 |
| | | | median | 21.55 | 21.84 | 51.70 | 43.49 | 26.54 | 48.78 |
| | | | $\sigma(\lambda)$ | 0.38 | 0.85 | 4.22 | 0.98 | 0.40 | 3.42 |
| | | $\lambda^*$(m) | Mean | **5.38** | **1.81** | **8.16** | **3.66** | **2.95** | **4.83** |
| | | | median | **5.39** | **1.82** | **8.62** | **3.62** | **2.95** | **4.88** |
| | | | $\sigma(\lambda^*)$ | 0.09 | 0.07 | 0.70 | 0.08 | 0.04 | 0.34 |

Consequently, the multivariate estimates of $\lambda^*$ compares with univariate estimates in a similar way :

- The distribution of $\lambda^*$ in the multivariate case is included in the envelope of univariate distributions,

- The dispersion of this multivariate distribution, measured by $\sigma(\lambda^*)$, is always close to the minimum value that can be achieved by any of the univariate distributions.

Hence, the multivariate method improves the identification of the wavelength : it is less sensitive to a local high frequency variation of a given variable if this variation is not associated with a variation of the others variables. However, there is no direct way of validating the estimates from these raw results : a way of doing so would be to build a synthetic, equivalent periodic geometry parameterized by the identified wavelength in order to verify that it yields, for example, a similar reach-average rating curve. This will be the subject of further work.

In the following section, we will compare the wavelet method with a benchmark method using talweg elevation.

## 4.2   Comparison with benchmark method

In this section, we compare our method's results with a selected benchmark method from the literature (i.e., BDT). This method shows good results in the identification of these bedfroms according to some researches (e.g., Frothingham and Brown (2002); Krueger and Frothingham (2007).

The technique of O'Neill and Abrahams (1984) (BDT) uses a tolerance value ($T$), which defines the minimum absolute value needed to identify a pool or a riffle (Krueger and Frothingham, 2007). It is calculated using the standard deviation $S_D$ of the series of bed elevation differences from upstream to downstream for each reach and corrected by a coefficient chosen according to the reach. For this, we test several tolerance values, and for the Graulade (1), Ozanne (4), Avenelles (5) and Orgeval (6) reaches we find the same results. We choose to check one tolerance value for each reach with $T = S_D$. This method gives pools and riffles positions by assigning a crest as a riffle and a bottom as a pool and therefore the computation of the wavelengths becomes a little difficult. So, we chose to calculate a series of pool-pool and riffle-riffle spacings, their medians, and standard deviations and then calculate their averages.

This was applied to all rivers and the results are depicted in the Fig. 10. Statistics of the BDT are shown in Table 5 which displays a comparison between these two types of morphological units' identification and mostly the identification of an

average wavelength of the reach.

Fig. 10 shows the BDT results on all reaches, this method relies only on topography to determine the positions of pools and riffles, moreover it also uses a threshold $T$ (tolerance) but the technique does not need a calibration reach or field investigation to know how to set this threshold. In this figure, Round points are pools or riffles and from these points we can calculate the wavelengths and longitudinal spacing of each reach as we stated before.

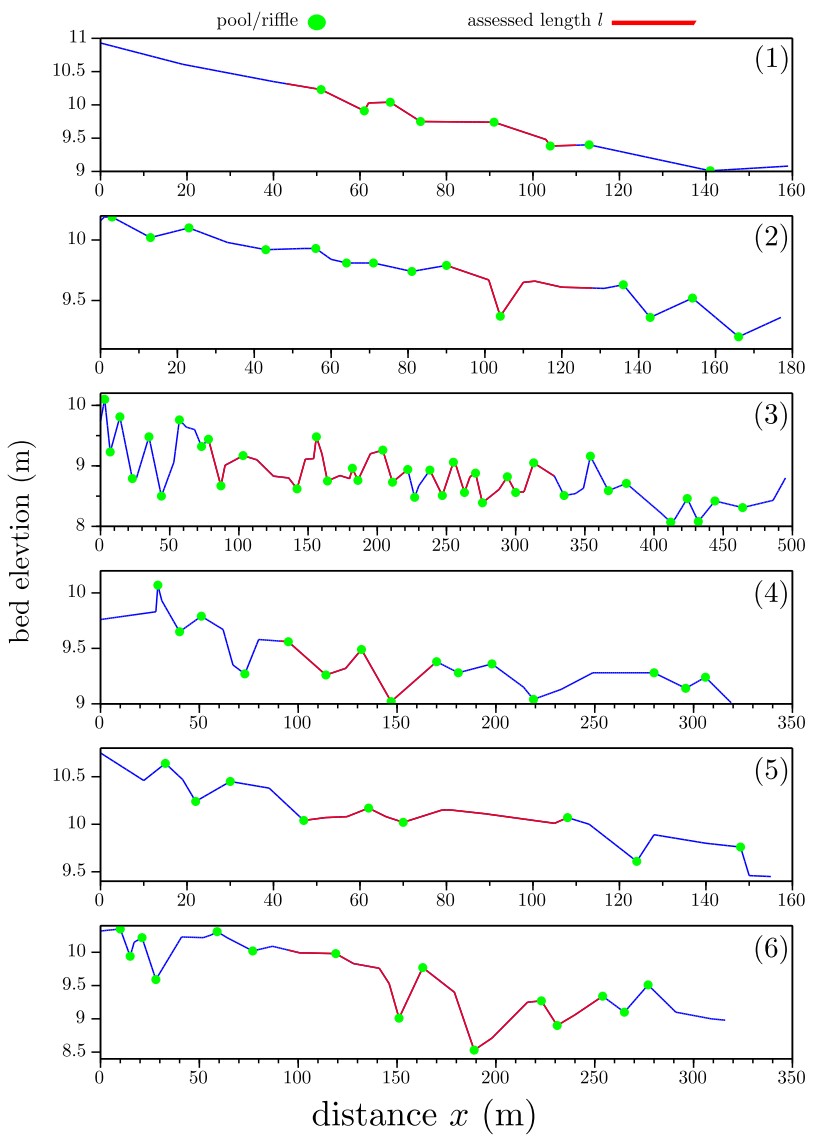

**Figure 10.** Results of the BDT method using a tolerance equal to the standard deviation on the total length and the assessed one (red) for all reaches (1 to 6). Round points are pools or riffles : pools are high and riffles are low points.

The work of the wavelet analysis is done on the assessed length $\ell$. However, the BDT method works on the total length of the reaches. This was done to determine how effective the wavelength extracted by the wavelet analysis can represent the entire reach even if an entire part is left unassessed.

For the wavelet method (Fig. 9), the wavelength extraction is among its objectives, while the BDT does not directly calculate the wavelength. It is computed by averaging the pool-to-pool and riffle-to-riffle distances. To compare these two methods, we will use only the longitudinal spacing ($\lambda^*$) as a criterion.

**Table 5.** Results of BDT and the multivariate wavelet methods for all reaches. $\lambda_{meth,mean}$ is the mean wavelength using one of the methods (for BDT, it used on the total length and on the assessed one $\ell$). $\lambda_{meth,median}$ is the median, and $\lambda^*_{meth,mean}$ is the mean longitudinal spacing, $\lambda^*_{meth,median}$ is the median, and $\sigma(\lambda_{meth,median})$ is the standard deviation related to the longitudinal spacing. (-) means that we find only one longitudinal spacing which is the mean and the median and there is no standard deviation.

| Reaches | 1: Graulade | 2: Semme | 3: Olivet | 4: Ozanne | 5: Avenelles | 6: Orgeval |
|---|---|---|---|---|---|---|
| **Total length $L$ (m)** | 160.0 | 177.0 | 495.0 | 319.0 | 155.0 | 318.0 |
| **Assessed length $\ell$ (m)** | 67.0 | 37.0 | 251.0 | 77.0 | 60.0 | 158.0 |
| $\lambda_{wav,mean}$ (m) | 21.54 | 21.74 | 48.98 | 43.89 | 26.59 | 48.29 |
| $\lambda_{BDT,mean}$ (m) | 23.67 | 25.33 | 24.94 | 41.12 | 33.63 | 39.90 |
| $\lambda_{\ell BDT,mean}$ (m) | 20.75 | - | 28.00 | 35.00 | - | 46.00 |
| $\lambda^*_{wav,mean}$ | 5.38 | 1.81 | 8.16 | 3.66 | 2.95 | 4.83 |
| $\lambda^*_{BDT,mean}$ | 5.92 | 2.11 | 4.16 | 3.43 | 3.74 | 3.99 |
| $\lambda^*_{\ell BDT,mean}$ | 5.19 | - | 4.66 | 2.92 | - | 4.60 |
| $\lambda_{wav,median}$ (m) | 21.55 | 21.84 | 51.70 | 43.49 | 26.54 | 48.78 |
| $\lambda_{BDT,median}$ (m) | 26.00 | 21.25 | 21.75 | 36.50 | 30.50 | 39.00 |
| $\lambda_{\ell BDT,median}$ (m) | 20.75 | - | 23.50 | 35.00 | - | 46.00 |
| $\lambda^*_{wav,median}$ | 5.39 | 1.81 | 8.62 | 3.62 | 2.95 | 4.88 |
| $\lambda^*_{BDT,median}$ | 6.5 | 1.77 | 3.63 | 3.04 | 3.39 | 3.90 |
| $\lambda^*_{\ell BDT,median}$ | 5.19 | - | 3.92 | 2.92 | - | 4.60 |
| $\sigma(\lambda^*_{wav,mean})$ | 0.09 | 0.07 | 0.70 | 0.08 | 0.04 | 0.34 |
| $\sigma(\lambda^*_{BDT,mean})$ | 2.06 | 0.82 | 1.83 | 1.56 | 1.71 | 1.91 |
| $\sigma(\lambda^*_{\ell BDT,mean})$ | 2.21 | - | 2.30 | - | - | 0.71 |

In Table 5, we present results of the BDT on the total length $L$ of all reaches and on the assessed length $\ell$. Using the total length $L$, the longitudinal spacings found with the BDT are close to the ones found with the wavelet analysis (deviation less than 1 time the bankfull width for the median), in all the reaches except the Olivet (deviation of $4w_{bf}$). Over the assessed length $\ell$ we find very similar results with a deviations less than one time the bankfull width. However, the shortening of the length ($\ell < L$) reduces the number of pools and riffles identified (Graulade (1) and Avenelles (5)) and therefore introduces bias. This indicates that a length greater than two cycles (pool-riffle) is always required to produce a pseudo-periodicity of the reach by

both methods, a condition which is clearly not fulfilled for all reaches of our dataset. But for the other rivers except Olivet (3) and Orgeval (6) reaches, there is no much improvement if we replace the total length with the assessed one. In this comparison, we found that the wavelengths extracted by the multivariate wavelet analysis are generally included in the variance intervals of the wavelengths found by the BDT. This was verified in all reaches except the Olivet River (3) where there is a big difference between the longitudinal spacings found by BDT and by wavelets. This difference is due to the choice of the tolerance value, which is low in our case to the point of not filtering out the high-frequency variability of bed elevation and therefore gives a lower periodicity compared to the wavelets.

## 5   Discussion

In this study, we consider the BDT method as a benchmark method. We do not consider a specific method to be the "true" one, we only apply these methods to have a general idea on the uncertainties in the identification of morphological units. This method was chosen not because it is the 'best' method for pool-riffle identification, but because it does not use thresholds (except for the tolerance $T$ which does not depend on field data). It means that it does not require a preliminary calibration of thresholds on velocity, hydraulic radius, etc. on an independent reach (e.g., Wyrick and Pasternack (2014); Hauer et al. (2009)). These thresholds vary from one reach to another and according to characteristics of each river. For this reason, we didn't compare our method with threshold methods on this dataset. In contrast, the results of the longitudinal spacing intervals will be compared with literature.

For a long time, researchers have found common interval of longitudinal spacings that vary between 5 and 7 times the channel width (Leopold et al., 1964; Keller, 1972; Richards, 1976a; Gregory et al., 1994). Keller (1972) found that the median is less and varies between 3 and 5 the channel width. O'Neill and Abrahams (1984), using BDT method, found the same results but with a median close to 3 the channel width and this value can vary according to the tolerance $T$. Carling and Orr (2000) found lower values than before at about $3w$. Recent studies (e.g., Wyrick and Pasternack (2014)) have calculated the longitudinal spacing of six morphological units using 2D identification methods. The average of these pool and riffle spacings are, respectively, 3.3 and 4.3 the channel width, which is less than the commonly accepted values of 5–7 $w$.

In this study, the longitudinal spacing vary in the mean and the median from ∼1.8 to 8.6 times the bankfull width, supporting the conclusion of Carling and Orr (2000) that pools are spaced approximately three to seven times the channel width. However, the quoted longitudinal spacing relationships should be considered in the context that the bankfull width and spacing distance are inherently variable even for short length reaches. To illustrate this inherent variability, we found the example of Keller and Melhorn (1978) where the pool-pool spacing values ranged from 1.5 to 23.3 channel widths, with an overall mean of 5.9 (Gregory et al., 1994; Knighton, 2014). This variability in longitudinal spacing is probably related to a short assessed length, a small number of cross-sections surveyed, or other factors such as geology, bank characteristics (cohesion), grain size of the river bed, artificial channel modifications, etc.

We worked with a dataset that contains cross-sections spaced 0.46 to 2.9 times bankfull width. Other studies have used much shorter spacings (e.g., Pasternack and Arroyo (2018); Legleiter (2014a)) to identify morphological units. Of course, the larger

the number of cross-sections, the more robust the identified correlations will be. In addition, we worked with irregularly spaced cross-sections, which will normally lead to biases in the results. Despite this, the "biased"placement does not impairs the overall methodology. This methodology has provided good results in terms of longitudinal spacings and therefore it can be applied for a shorter cross-section spacings to clearly identify these alternate morphologies. The short lengths we found raise questions about the *naturality* of the rivers. In our case, the rivers are subject to artificial modifications (e.g., bridges, weirs) and rehabilitations, which will have a significant impact on the hydro-morphological parameters (width, depth, meandering, etc.). This can have a very important impact on the identification of pseudo-periods.

The wavelet ridge analysis is powerful in identifying pseudo-periods, amplitude and phase while respecting the correlations between parameters. We can thus identify alternating morphological units in a more objective way in terms of frequency/wavenumber.

This wavelength can be used to represent the variability of the bathymetry in hydraulic models in cases where we do not have a full access to the geometry of the channel (e.g., remote sensing data as the overcoming Surface Water and Ocean Topography Mission) and the morphology can be modelled by pseudo-periodic functions. Furthermore, it can be implemented in synthetic geometry generators (e.g., River Builder, Pasternack and Zhang (2020)) where the bathymetry and sinuosity wavelengths extracted by the wavelets can be used to model meandering rivers with alternating morphologies. Ultimately, hydraulic modeling will be the true test of the potential of a pseudo-periodic equivalent geometry (e.g., for simulating a reach-average rating curve).

On the other hand, it presents drawbacks compared to other methods. First, the cone of influence that ignores a large part of the river and sometimes biases the results (in the case of the Graulade (1) and Semme (2) reaches) in the case of small total lengths. Similarly for reach length and number of morphological units as for the number of cross-sections : the larger it is, the more robust the results will be, and the smaller the relative portion of "unassessed length" will be. Still, the method remains a powerful tool for non-stationary analysis. Another problem is the amplitude which is sometimes overestimated in some regions of the topography. We visualized this in several cases in our study, since we used the Neperian logarithm to avoid negative values and therefore the inverse function (exponential) will give slightly larger values. However, this does not bias the identified wavelength of the reach.

## 6 Conclusions

In this study, we present an automatic procedure based on Wavelet Ridge extraction to identify some characteristics of alternating morphological units (MUs), such as their longitudinal spacing and amplitude. The method does not rely on any a priori thresholds to identify MUs sequences. It was applied to six rivers with a maximum length of 500 meters. We chose to work with classical hydro-morphological variables (velocity, hydraulic radius, bed shear stress) in addition to the planform channel direction angle that evaluates the impact of river sinuosity in the determination of the wavelength.

As a result, identified wavelengths are consistent with values of the literature (mean in 3-7 $w_{bf}$). The use of a multivariate approach yields more robust results than the univariate approaches, by ensuring a consistent covariance of flow variables in the

pseudo-periodic behavior.

Given the short length of several reaches, the relatively small number of cross-sections for each reach, and the possible impacts of artificial modifications, this paper is mainly a proof-of-concept of the wavelet approach. It does not preclude the long-term possibility of extending the work to other rivers with other types of MUs, other longer reaches with a large number of cross-sections.

## Appendix A: List of symbols

$A(x)$ : Cross-sectional area (m$^2$)

$A_{i,mod}$ : Signal amplitude of the shape of the modeled variable number $i$

$A_m$ : Signal amplitude of the shape

$\cos(\theta)$ : Cosine of local channel direction angle

$\cos(\theta)_{mod}$ : Modeled cosine of local channel direction angle

$f$ : Space series function (m)

$f_i$ : Measured space series function number $i$

$f_{i,mod}$ : Modeled space series function number $i$ with multivariate wavelet analysis

$f_{mod}$ : Modeled variable with the univariate wavelet analysis

$g$ : Acceleration due to gravity and its value is 9.81 m.s$^{-2}$

$k(x)$ : Wavenumber (rad.m$^{-1}$)

$K(x)$ : Local wavenumber that corresponds to the maximum variance of the signal (rad.m$^{-1}$)

$K_i(x)$ : Local wavenumber that corresponds to the maximum variance of the variable $i$ (rad.m$^{-1}$)

$\ell$ : $K(x)$ support (m)

$L$ : Total reach length (m)

$N$ : Number of total chosen variables

$n$ : Manning's roughness coefficient

$P(x)$ : Wetted perimeter (m)

$Q_{min}$ : Minimum discharge modeled (m$^3$.s$^{-1}$)

$R(x,s)$ or $R(x,k)$ or $R$ : Absolute value or modulus of the wavelet transform at a position $x$ and with a scale $s$ or wavenumber $k$

$R_h$ : Hydraulic radius (m)

$R_{h,mod}$ : Modeled hydraulic radius (m)

$s$ : Dilation or scale factor

$S$ : Reach slope (m/m)

$S_D$ : Standard deviation of the bed elevation diffrence series (m)

$T$ : Tolerance value used in BDT mehtod (m)

$v(x)$ : Velocity (m.s$^{-1}$)

$v_{mod}$ : Modeled velocity (m.s$^{-1}$)

$v_{obs}$ : Measured velocity (m.s$^{-1}$)

$w$ : Reach width (m)

$w_{bf}$ : Reach bankfull width (m)

$w_m$ : Mean bankfull width (m)

$x$ : Translation factor in the wavelet transform or the abscissa position (m)

| | | |
|---|---|---|
| $y = y_{max}$ | : | Water depth measured from the the talweg elevation $y = z_{ws} - z$ (m) |
| $y_m$ | : | Mean depth (m) |
| $z$ | : | Measured bed elevation or talweg elevation (m) |
| $z_{ws}$ | : | Water surface elevation measured from the 0NGF (m) |
| $\alpha$ | : | Fourier factor associated with the wavelet (m.rad$^{-1}$) |
| $\beta$ | : | Dimensionless frequency taken to be 6 recommended by Torrence and Compo (1998) |
| $\lambda$ | : | Reach wavelength (m) |
| $\lambda^*$ | : | Typical pool (riffle) spacing or dimensionless reach wavelength |
| $\rho$ | : | Water density (997kg.m$^{-3}$) |
| $\theta$ | : | local channel direction angle in rad |
| $\sigma(w)$ | : | Standard deviation of the width along the reach (m) |
| $\sigma()$ | : | Standard deviation |
| $\tau_b(x)$ | : | Bed shear stress in the x abscissa (Pa) |
| $\tau_{b,mod}(x)$ | : | Modeled bed shear stress in the x abscissa (Pa) |
| $\Phi$ | : | Corresponding phase at the position x and the wavenumber $K$ with $\Phi(x) = \phi(x, K(x))$ (rad) |
| $\Phi_i$ | : | Phase at the position $x$ and the wavenumber $K$ for the variable number $i$ |
| $\phi(x,s)$ or $\phi(x,k)$ or $\phi$ | : | Phase or argument at a position $x$ and with a scale $s$ or wavenumber $k$ (rad) |
| $\phi_i$ | : | Phase of the variable number $i$ |
| $\psi$ | : | Mother wavelet function |
| $\psi_{s,x}$ | : | Daughter wavelet function |
| $\eta$ | : | Dimensionless position parameter |
| $\mathbb{C}$ | : | Complex numbers |
| $\mathbb{R}$ | : | Real numbers |
| $\mathbb{R}_+^*$ | : | Positive real numbers |
| $\mathcal{W}[f](x,s)$ | : | Continuous wavelet transform of $f(x)$ with the wavelet $\psi$ |

## Appendix B: Mathematical calculus for the wavelet transform

### B1 The univariate case

The conjugate form of the mother wavelet is :

5   $\psi^*(\eta) = \pi^{-\frac{1}{4}} e^{-i\beta\eta - \frac{\eta^2}{2}}$ (B1)

Its derivative in relation to the mute variable $\eta$ is :

$$\psi^{'*}(\eta) = -\pi^{-\frac{1}{4}}\,(i\beta+\eta)\,e^{-i\beta\eta-\frac{\eta^2}{2}} \tag{B2}$$

$$= -(i\beta+\eta)\psi^*(\eta) \tag{B3}$$

In section 3-1, $\eta$ is a mute integration variable and $x$ appears only in the argument $\alpha k(\xi - x)$ of the function $\psi^*$. By applying the derivation formula of a composite function, the derivative of the wavelet transform is expressed by :

$$\frac{\partial}{\partial x}\mathcal{W}[f(x)](x,k) = \sqrt{\alpha k}\int_{-\infty}^{+\infty} f(\eta)\frac{\partial}{\partial x}\Big[\psi^*\Big(\alpha k(\eta-x)\Big)\Big]d\eta$$

$$= \sqrt{\alpha k}\int_{-\infty}^{+\infty} f(\eta)\cdot(-\alpha k)\cdot\psi^{'*}\Big(\alpha k(\eta-x)\Big)d\eta$$

$$= (\alpha k)\sqrt{\alpha k}\int_{-\infty}^{+\infty} f(\eta)\cdot\Big(i\beta+\alpha k(\eta-x)\Big)\cdot\psi^*\Big(\alpha k(\eta-x)\Big)d\eta$$

$$= (\alpha k)\sqrt{\alpha k}\int_{-\infty}^{+\infty}\Big[(i\beta-\alpha k x)f(\eta)+\alpha k\eta f(\eta)\Big]\cdot\psi^*\Big(\alpha k(\eta-x)\Big)d\eta$$

$$= (\alpha k)(i\beta-\alpha k x)\mathcal{W}[f(x)](x,k) \quad + \quad (\alpha k)^2\,\mathcal{W}[xf(x)](x,k)$$

On the other hand, we have :

$$\frac{\partial}{\partial x}\mathcal{W}[f(x)](x,k) = \frac{\partial}{\partial x}\Big(R(x,k)e^{i\phi(x,k)}\Big)$$

$$= \Big[\frac{1}{R}\frac{\partial R}{\partial x}+i\frac{\partial\phi}{\partial x}\Big]Re^{i\phi}$$

$$\frac{\partial}{\partial x}\mathrm{Re}\left(\ln\mathcal{W}[f(x)](x,k)\right) = \frac{1}{R}\frac{\partial R}{\partial x} = \mathrm{Re}\left(\frac{1}{\mathcal{W}[f(x)](x,k)}\frac{\partial}{\partial x}\mathcal{W}[f(x)](x,k)\right)$$

$$\frac{\partial}{\partial x}\mathrm{Im}\left(\ln\mathcal{W}[f(x)](x,k)\right) = \frac{\partial\phi}{\partial x} = \mathrm{Im}\left(\frac{1}{\mathcal{W}[f(x)](x,k)}\frac{\partial}{\partial x}\mathcal{W}[f(x)](x,k)\right)$$

Finally :

$$\frac{\partial\phi}{\partial x} = \mathrm{Im}\left\{(\alpha k)(i\beta-\alpha k x)+(\alpha k)^2\frac{\mathcal{W}[xf(x)](x,k)}{\mathcal{W}[f(x)](x,k)}\right\} \tag{B4}$$

$$\frac{\partial\phi}{\partial x} = (\alpha k)\beta \quad + \quad (\alpha k)^2\,\mathrm{Im}\left\{\frac{\mathcal{W}[xf(x)](x,k)}{\mathcal{W}[f(x)](x,k)}\right\}$$

The previous expression numerically avoids the derivative of the function $\phi(x,k)$, which varies quickly for large wave numbers.

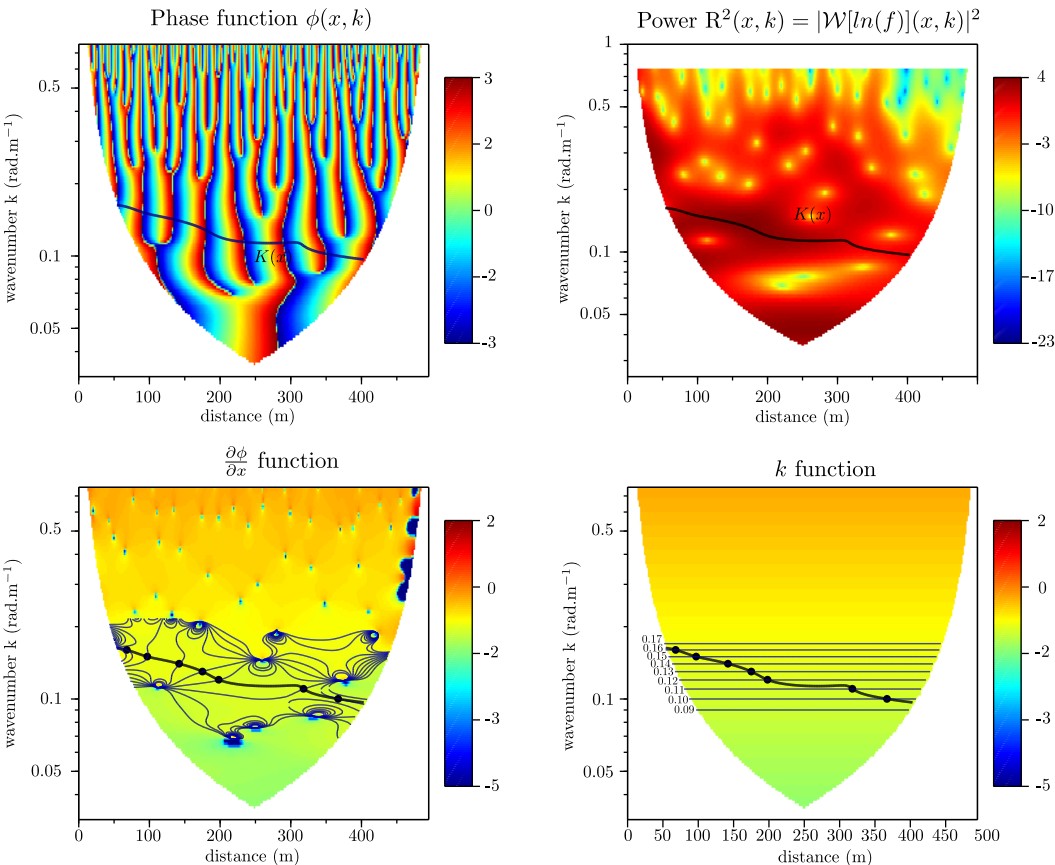

**Figure B1.** Steps of determining the local wavenumber $K(x)$ using the wavelet univariate ridge analysis of the the velocity of the Olivet (3) reach, represented in the four panels. (A) The phase function $\phi(x,k)$; (B) the power's cone of influence of the wavelet to characterize the region where there is a maximum variability of the velocity in Neperian logarithm; (C) the function $\frac{\partial \phi}{\partial x}$; (D) the function $k$

## B2    The multivariate case

In the multivariate case, we should resolve the equation (20) which contain three derivatives to compute. The first one is already done in the univariate case which is :

$$\frac{\partial \phi_i(x,k)}{\partial x} = (\alpha k)\,\beta \quad + \quad (\alpha k)^2 \operatorname{Im}\left\{ \frac{\mathcal{W}\big[x f_i(x)\big](x,k)}{\mathcal{W}\big[f_i(x)\big](x,k)} \right\} \tag{B5}$$

5    The second one is the computation of $\dfrac{\partial^2 \phi_i(x,k)}{\partial k \partial x}$ :

$$\frac{\partial^2 \phi_i(x,k)}{\partial k \partial x} = (\alpha \beta) \quad + \quad 2\alpha^2 k \operatorname{Im}\left\{ \frac{\mathcal{W}\big[x f_i(x)\big](x,k)}{\mathcal{W}\big[f_i(x)\big](x,k)} \right\} \quad + \quad (\alpha k)^2 \operatorname{Im}\left\{ \frac{\partial}{\partial k}\left( \frac{\mathcal{W}\big[x f_i(x)\big](x,k)}{\mathcal{W}\big[f_i(x)\big](x,k)} \right) \right\} \tag{B6}$$

For that we should develop $\dfrac{\partial}{\partial k}\left(\dfrac{\mathcal{W}\big[xf_i(x)\big](x,k)}{\mathcal{W}\big[f_i(x)\big](x,k)}\right)$

$$\frac{\partial}{\partial k}\left(\frac{\mathcal{W}\big[xf_i(x)\big](x,k)}{\mathcal{W}\big[f_i(x)\big](x,k)}\right) = \frac{1}{\mathcal{W}\big[f_i(x)\big](x,k)}\frac{\partial \mathcal{W}\big[xf_i(x)\big](x,k)}{\partial k} - \frac{\mathcal{W}\big[xf_i(x)\big](x,k)}{\mathcal{W}\big[f_i(x)\big](x,k)^2}\frac{\partial \mathcal{W}\big[f_i(x)\big](x,k)}{\partial k} \tag{B7}$$

We calculate each derivative :

$$\frac{\partial \mathcal{W}\big[f_i(x)\big](x,k)}{\partial k} = \left(\frac{1}{\sqrt{k}}\frac{\partial\sqrt{k}}{\partial k}\right)\mathcal{W}\big[f_i(x)\big](x,k) + \sqrt{\alpha k}\int\limits_{-\infty}^{+\infty} f(\eta)\frac{\partial}{\partial k}\Big[\psi^*\big(\alpha k(\eta-x)\big)\Big]d\eta$$

$$= \left(\frac{1}{2k}\right)\mathcal{W}\big[f_i(x)\big](x,k) + \sqrt{\alpha k}\int\limits_{-\infty}^{+\infty} f(\eta)\alpha(\eta-x)\psi^{*'}\big(\alpha k(\eta-x)\big)d\eta$$

$$= \left(\frac{1}{2k}\right)\mathcal{W}\big[f_i(x)\big](x,k) + \sqrt{\alpha k}\int\limits_{-\infty}^{+\infty} f(\eta)\alpha(\eta-x)\big(i\beta+\alpha k(\eta-x)\big)\psi^*\big(\alpha k(\eta-x)\big)d\eta$$

$$= \left(\frac{1}{2k}\right)\mathcal{W}\big[f_i(x)\big](x,k) + \sqrt{\alpha k}\int\limits_{-\infty}^{+\infty}\Big[(i\beta-\alpha^2 kx^2)+(-i\beta\alpha+2\alpha^2 kx)\eta-(\alpha^2 k)\eta^2\Big]f(\eta)\psi^*\big(\alpha k(\eta-x)\big)d\eta$$

$$= \left(\frac{1}{2k}+i\beta\alpha x-\alpha^2 kx^2\right)\mathcal{W}\big[f_i(x)\big](x,k) + \big(2\alpha^2 kx - i\beta\alpha\big)\mathcal{W}\big[xf_i(x)\big](x,k) - \big(\alpha^2 k\big)\mathcal{W}\big[x^2 f_i(x)\big](x,k)$$

We find a general formulation with $p = 0...N-2$ :

$$\frac{\partial \mathcal{W}\big[f_i(x)\big](x,k)}{\partial k} = \left(\frac{1}{2k}+i\beta\alpha x-2\alpha^2 kx^2\right)\mathcal{W}\big[x^p f_i(x)\big](x,k)+\big(2\alpha^2 kx-i\beta\alpha\big)\mathcal{W}\big[x^{p+1} f_i(x)\big](x,k)-\big(\alpha^2 k\big)\mathcal{W}\big[x^{p+2} f_i(x)\big](x,k)$$

$$\tag{B8}$$

The third one is the computation of $\dfrac{\partial^3\phi_i(x,k)}{\partial k^2\partial x}$

$$\frac{\partial^3\phi_i(x,k)}{\partial k^2\partial x} = 2\alpha^2\mathrm{Im}\left\{\frac{\mathcal{W}\big[xf_i(x)\big](x,k)}{\mathcal{W}\big[f_i(x)\big](x,k)}\right\} \;+\; 4\alpha^2 k\,\mathrm{Im}\left\{\frac{\partial}{\partial k}\left(\frac{\mathcal{W}\big[xf_i(x)\big](x,k)}{\mathcal{W}\big[f_i(x)\big](x,k)}\right)\right\} \;+\; (\alpha k)^2\,\mathrm{Im}\left\{\frac{\partial^2}{\partial k^2}\left(\frac{\mathcal{W}\big[xf_i(x)\big](x,k)}{\mathcal{W}\big[f_i(x)\big](x,k)}\right)\right\}$$

$$\tag{B9}$$

With :

$$\frac{\partial^2}{\partial k^2}\left(\frac{\mathcal{W}\big[xf_i(x)\big](x,k)}{\mathcal{W}\big[f_i(x)\big](x,k)}\right) = \frac{1}{\mathcal{W}\big[f_i(x)\big](x,k)}\frac{\partial^2 \mathcal{W}\big[xf_i(x)\big](x,k)}{\partial k^2} - \frac{\mathcal{W}\big[xf_i(x)\big](x,k)}{\mathcal{W}\big[f_i(x)\big](x,k)^2}\frac{\partial^2 \mathcal{W}\big[f_i(x)\big](x,k)}{\partial k^2}$$

$$- \frac{2}{\mathcal{W}\big[f_i(x)\big](x,k)^2}\frac{\partial \mathcal{W}\big[f_i(x)\big](x,k)}{\partial k}\frac{\partial \mathcal{W}\big[xf_i(x)\big](x,k)}{\partial k}$$

and :

$$\frac{\partial^2 \mathcal{W}\big[x^p f_i(x)\big](x,k)}{\partial k^2} = \left(-\frac{1}{2k^2} - \alpha^2 x^2\right) \mathcal{W}\big[x^p f_i(x)\big](x,k) + \left(2\alpha^2 x\right) \mathcal{W}\big[x^{p+1} f_i(x)\big](x,k) - \left(\alpha^2\right) \mathcal{W}\big[x^{p+2} f_i(x)\big](x,k)$$
$$+ \left(\frac{1}{2k} + i\beta\alpha x - \alpha^2 k x^2\right) \frac{\partial \mathcal{W}\big[x^p f_i(x)\big](x,k)}{\partial k} + \left(2\alpha^2 k x - i\beta\alpha\right) \frac{\partial \mathcal{W}\big[x^{p+1} f_i(x)\big](x,k)}{\partial k}$$
$$- \left(\alpha^2 k\right) \frac{\partial \mathcal{W}\big[x^{p+2} f_i(x)\big](x,k)}{\partial k}$$

5   *Acknowledgements.*  The authors acknowledge financial support from the French National Space Agency (Centre National d'Etudes Spatiales, CNES) and Sorbonne University through the Ph.D. grant of M. Mahdade. We would also like to thank G.B. Pasternack, the second anonymous reviewer, and the editor for the valuable suggestions and comments that greatly helped improve this paper.

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
