# Peer review of "Automatic identification of alternating morphological units in river channels using wavelet analysis and ridge extraction"

_Hydrology and Earth System Sciences, 2018_

## Referee Comment (RC1) · Anonymous Referee #1 · 3 Feb 2019

The manuscript "Wavelet and index methods for the identification of pool–riffle sequences" by Mahdade et al. presents two novel methods for the identification of pools and riffles in natural streams. These methods also allow the assessment of the main geometrical features of pools and riffles. The manuscript states that appropriate geometric description of pools and riffles is pivotal for flood modelling. I think this statement is correct when modelling floods (and flash floods) at the local scale. Conversely, previous studies have shown that simplified representations of river geometry can be a cost-effective solutions for flood modelling at the large (basin to continental scale). In fact, I believe that an accurate representation of river geometry is essential for the implementation of hydrodynamic models used for the investigation of local flow conditions and sediment transport. The scope of the paper could thus be extended to biological

and environmental modelling (oxygen exchange, fish habitat, sediment transport) and not only limited to flood forecasting.

The paper is interesting, sections 1 and 2 provide a comprehensive literature review; sections 4 and 5 provide a detailed explanation of the methodologies; the presentation and discussion of the results in section 6 is quite extensive. However, I think that a number of major modifications should be introduced before the publication of this study.

Firstly, I think that the research gap and the novelty of this study should be clearly stated. Why did the authors propose two novel methods for the identification of pools and riffles? What are the advantages of these two novel methodologies when compared to the existing ones? I believe that these aspects should be clearly stated in the manuscript.

Second, the results of the new methods are compared to the results of the BDT method. Is the BDT method used as benchmark or to validate the new methods? Is the BDT method considered more accurate than the new methods? If so, why? What are the advantages of using the two methods rather than using the BDT method? Would it be possible to validate the results of this study using field data?

Third, the computation of the index method relies on the results of the numerical model. Have the authors considered the impact of the uncertainties in the results of the numerical model on the results of the index method?

Furthermore, I suggest discussing the transferability of the new methods to other reaches. In other words, how easy would be to implement the proposed methodologies to other study areas? Are the data and algorithms required easy to collect and implement? Can other researchers implement the proposed methodologies?

Moreover, I think the manuscript should clearly state which methods are recommended. A more explicit presentation of the conclusions of this study would highlight

its scientific and practical relevance.

Finally, I believe that some sentences are a bit convolute and difficult to understand (please see below).

Regarding the structure of the paper, I would like to recommend two modifications: - Section 2 lists a large number of studies and it is a bit hard to follow. More specifically, I think it is difficult to appreciate the differences between the large number of criteria listed in this section. The authors might consider adding a table to summarise their literature review. - I suggest adding "Section 7 – Conclusions". This section should clearly summarise the aim of the study, its results, its "take home messages" and, if appropriate, its limitations.

I hope the authors will find my questions and recommendations useful to improve their manuscript.

I listed below a number of minor recommendations.

Page 1, lines 7-8: the sentence "To better take this high-frequency variability in bedforms into account in hydraulic models" is a bit convolute. The authors might consider improving the structure of this sentence.

Page 1, abstract: the abstract should clearly state the research gap, the aim, and the novelty of the study.

Page 1, line 9: the abstract mentions "several methods", however, only three (two novel methods and one benchmarking method) are listed explicitly.

Page 1, lines 12-13: the authors might consider avoiding the repetition of the word "compared".

Page 2, lines 14-15: I am not sure whether this is the final format of the paper, however, I suggest positioning each figure after a full stop (Figure 1 is currently positioned in the middle of a sentence).

Page 2, line 15: please correct "dimensionless reach wavelength".

Page 3, lines 4-5: this sentence is a bit hard to understand. Do the authors mean that the overarching purpose of their study is to provide a methodology for the prediction/modelling/assessment of cross sections variability?

Page 3, line 8: words such as "methods" or "techniques" might be more appropriate than "studies".

Page 3, line 11: could please the authors clarify the meaning of "descriptions of the water surface characteristics"? Is "water surface slope" (mentioned in Line 8) included in this latter category?

Page 3, line 14: I suggest clarifying the sentence "because it changes less with discharge".

Page 3, line 20: please rephrase "goes with the notion".

Page 3, line 22: please rephrase "allows one to extract".

Page 3, line 30: please rephrase "using a threshold on a criterion index."

Page 4, Figure 2(A): I believe that this figure is not mentioned in the text.

Page 4, lines 3-4: I think this sentence should be moved to the section 6.2 as it motivates the choice of the benchmarking method.

Page 5, line 7: "the areal difference asymmetry index by Knighton" has not been mentioned before, the authors might consider adding more context to this statement.

Page 5, line 32: the manuscript states: "a common geomorphological and hydrological" methods, I suggest specifying these methods.

Page 6, line 8: was the channel width/channel bankfull width used to compute dimensionless values of wavelength? I think this sentence is not clear.

Page 6, line 9: what do the authors mean with "certainty"?

Page 6, line 14: I suggest avoiding colloquial expressions such as "a great deal".

Page 6, line 32: I suggest rephrasing this sentence and avoid the use of "we".

Page 7, line 5: I believe that information on slope has been previously provided in page 6, line 32. Could please the authors explain the added value of this sentence?

Page 8, line 11: please clarify the sentence "It is based on interpolations rather than extrapolations".

Page 8, line 13: "visually": do the authors mean that they performed a manual calibration of the hydraulic model?

Page 8, line 14: please remove the second full stop.

Page 8, line 14: "multi-section flows": do the authors mean that the numerical model is used to predict a number of quantities (e.g. the elevation of the water surface, wetted perimeter, wetted surface,...) at a number of cross sections?

Page 8, line 3: why is the minimum discharge used for the implementation of the method?

Page 8, line 6: does "it" stand for "relevant information"? The authors might consider editing the structure of this sentence.

Page 8, line 8: I believe that "the trend" has not been explained before. I suggest clarifying this sentence. What does "detrended variables" mean? How are these variables computed?

Page 8, line 13: "contain the most explained variances" do the authors mean that those directions can explain the variability of the data? I suggest clarifying this sentence.

Page 8, lines 18-20: does these results confirm/contradict previous studies?

Page 9, figure 4: could please the authors explain the meaning of Dimension 1...9?

Page 9, lines 5-6: I think this sentence is unclear. What is the relationship between bed

elevation and hydraulic radius? The statement seems to be contradictory. Moreover, I was wondering whether any correlation between bed elevation and hydraulic radius is meaningful. Bed elevation is a geometric characteristic at the point scale. The hydraulic radius depends on discharge, river bed slope, cross section area.

Page 9, lines 6-8: the explanation based on hydraulic radius and Froude number is reasonable and (almost) intuitive. I suggest to clarify the added value of this finding compared to the existing literature.

Page 9, line 9: I suggest clarifying the importance of bed elevation.

Page 9, line 10: what do the authors mean with "we smooth" the data?

Page 12, lines 11-13: I suggest improving the readability of this sentence.

Page 12, lines 16-18: please improve the structure of this sentence: "have been interested. . .but working", both the verbs have the same subject.

Page 12, line 18: "analysis" is repeated.

Page 14, line 12: I suggest replacing "evacuate" with something more appropriate (an option could be "remove").

Page 15, line 3: please clarify "It also represents" (what does "it" stand for?)

Page 15, line 11: could please the authors better explain why this correction is applied?

Page 15, line 15: please correct the structure of this sentence: "we limit the study only with univariate analysis". Moreover, could please the authors justify this choice?

Page 15, line 26: please clarify the meaning of "multivariate case".

Page 17, Table 3: Table 3 and Table 2 show the results of the two methods for the same river reach. The authors might consider displaying these tables in the same page in order to allow a straightforward comparison of the results.

Page 18, lines 2-4: I think that this sentence is unclear.

[Figure]

Page 19, line 3: I believe that these results demonstrate a good level of agreement between the two methods. In my opinion, these results do not provide explicit information on the accuracy of the methods.

Page 19, line 5: the BDT methods is used to "validate" the results of the proposed methodologies. This choice implies that the BDT method is more accurate than the new methods introduced in this manuscript. Is this correct? If so, what are the benefits/advantages of using the two proposed techniques?

Page 19, lines 19-20: please clarify this sentence.

Page 19, line 23: the manuscript states that the results of BDT "are closer to the other methods and to reality". I strongly recommend to better substantiating this sentence. Which are the "other methods"? What does "reality" mean? Was the BDT method compared with field data? In which case study?

Page 20, lines 3-4: could please the authors clarify this sentence?

Page 20, lines 6-7: please rephrase this sentence.

Page 21, line 2: a Froude number of 0.30 looks a bit large. Could please the authors explain this result?

Page 21, line 3: it seems that the average values are driven by the results of the Graulade river, Are the average values representative of the sample?

Page 21, line 10-17: these lines present a comparison between the results of this study and some of the previous studies. The authors might (or might not) consider using a table to summarise these comparisons.

Page 21, line 21: I suggest motivating this sentence. Why aren't the previous methods considered quantitative?

Page 22, line 1: is "crossing" the most appropriate word?

[Figure]

Page 22, line 3: please clarify this sentence.

---

## Referee Comment (RC2) · Gregory Pasternack (Referee) · 24 Jul 2019

I was accepted to undertake this review on June 24 and I completed the review today, July 22, so I did the job within the typical 4 weeks allotted.

Review Synopsis

The authors present two methods for mapping the locations and extents of 1D longitudinally arrayed riffles and pools and quantifying the spacing between successive pools and that between successive riffles. The method are applied to up to 6 reaches and then they are inter-compared as well as compared to a classic method from the 1980s. The study conclusion is that all 3 methods yield roughly similar results, with the proposed wavelet method also providing riffle-pool relief, though those this can be

obtained from a variety of pre-existing methods, too. There are no scientific conclusions about riffles and pools in the study reaches presented or discussed relative to the literature on the topic. Upon very thorough inspection and close reading, I have many questions and discussion points about the methods that would need to be clarified by better and more thorough writing in a major revision, likely including a meaningful supplementary materials file rather than blasting the primary manuscript to an unreasonable length. I think the index method as applied with the selected variables is technically unsound, but I would grant the authors a chance to explain better and justify their choices. The structure of the manuscript also needs to be overhauled to fit a better framework with a clear experimental design. I cannot come to a final conclusion without first having all questions answered, as detailed below in the broad narrative review and then the detailed specific comments I provide. Therefore, I recommend major revision and further external review, ideally from other viewpoints than my own by others who also work on spatial series of river topographic data.

Narrative Review.

As I understand the open review process for HESS and its sibling journals, the goal is to have an open discussion among the community about a manuscript to bring to light a wide range of issues related to the submission that can help make it better, while also performing a critical analysis to aid the journal's decision to accept or reject an article. As a long-serving associated editor, reviewer, and author for several journals, my preference is to offer the journal and authors my best effort at thinking through a manuscript in great detail to give the best insights and ideas I can offer to help to constructively improve the work and have it come to its fullest potential. My reviews are long and substantive, which I think is good for making a high-quality open discussion and final manuscript, but I know it can feel burdensome to authors, because then they have to reflect on all the issues I raise. In this case, this manuscript is right in the center of the scope of research I do on fluvial geomorphology and I have published several articles that use other methods to achieve the same goals as here, and more.

Therefore, I definitely have a lot of experience and insight about the contents of this manuscript. To be fair and open, I do mention my own research articles in this review, because it is one of the topics I am a recognized expert in, but I also do balance that by citing excellent articles by other experts from around the world. It is up to authors to decide if they want to cite my articles or not, I cannot expect them to, but I do think all the articles I mention in my review from myself and others do provide important insights that bear on the article here and it would be wise for the authors to at least consider their merits given that the manuscript's literature review of morphological unit analysis methods pretty much ends at 2001, not 2019.

In this manuscript the authors undertake analyses of cross-sectional river channel topographic and hydraulic data to achieve 2 methodological goals: (a) map the locations and extents of 1D longitudinally arrayed riffles and pools and (b) quantify the spacing between successive pools and that between successive riffles. The manuscript provides a literature review about how to do these two things beginning on the bottom of page two and continuing through section 2 ending on page 6, almost exclusively recalling articles from the 20th century. The fundamental premise of the article and the strategies taken are similarly rooted in 20th century data collection methods, which then constrain the analysis methods, results, and comparisons. In the 21 century, the emergency of meter-resolution digital elevation models (DEMs) of entire river corridors has fundamentally transformed the breadth and depth of not only analyses but the very questions that can be asked. On a practical level, there have been two developments that the literature review misses that are very important to the study's context and should be addressed to have a modern summary of the state of the science. First, meter-resolution topo-bathymetric DEMs of rivers have enabled for not only higher resolution bed elevation spatial series to be developed showing many more sub-reach scale fluctuations than considered in the 20th century, but also stage-dependent river width series (at elevations associated with various lateral geometric slope breaks and/or discharges) and other variables. These spatial series can now be produced for many variables in vastly higher resolution, such as a spacing of 1-5% of bankfull width

rather than one every $\sim$ 0.5-3 bankfull widths used in this study. Further, the advantage of joint analysis of depth and width is that the spatial series of the "geomorphic covariance structure" (sensu Pasternack et al., 2018a,b and preceding work) between them controls the morphodynamic process of flow convergence routing, as explained with citations in the detailed comments below. My own lab group's research program pursues this course of inquiry to not only map morphological units and their spacings, but to also link topographic patterns to morphodynamic processes, and even do that in a way that transcends spatial scale for the first time. Second, meter-resolution topo-bathymetric DEMs of rivers enable fully spatially explicit (lateral and longitudinal) analyses of river corridor terrain. This comes in three varieties. First, Prof. Martin Thom's research group is inventing entirely new 3D DEM statistical metrics to characterize whole surfaces, which may eventually lead to new ideas about morphological unit segregation or abandonment of that concept in favor of continuous surface analysis. Second, Dr. Carl Legleiter's research group has been using geostatistical methods to also evaluate river corridor DEMs, including their spatial autocorrelation and related topics. This has the same conceptual potential as Martin Thom's research but using different statistics. Finally, my own group and that of my former student Prof. Joe Wheaton are pursing methods at making spatially explicit morphological unit maps- in my group's case taking advantage of 2D hydraulic modeling to obtain depth and velocity grids that can be classified with decision-tree analysis at the proper discharge and in his group's case by doing direct topographic analysis to segregate fluvial landforms using 3D topographic geometry rule sets. In short, the 21st century is seeing an exciting and rapid expansion of approaches to mapping fluvial landforms, evaluating their spacings, and going beyond that to link patterns to processes. None of these 21st century developments are mentioned in the literature review, which means the authors may not be aware of them. In the detailed comments below, I have tried to cite examples from the above mentioned new works specifically, and of course that means I'm also citing my own group's research, but I only do that because it is directly producing the same kinds of results as here, just with some newer ideas and high-resolution datasets. So the literature review
needs work as the methods for riffle-pool mapping and spacing quantification did not end in 2001, but what about this study as a whole, where does it stand with its ideas?

This study lays out two methods and presents their results. The so called index method and then wavelet analysis, both applied to identify locations and extents of riffles ands pools, and then to quantify their spacings. Considering just these methods in isolation, there are both positive developments and several concerns to be addressed. Going in reverse order, I totally agree that wavelet analysis is an important and meaningful tool for analyzing spatial series, though the real potential comes from analyzing meter-resolution river corridor DEMs. Wavelets are not new to hydrological or geomorphic research (see citations below), so merely applying them is not novel though it is meaningful. The wavelet study and associated developments in the manuscript are meaningful and really constitute the merit of the manuscript, with a variety of detailed methodological questions raised below notwithstanding. I must point out though an important criticism and caveat in the results that the authors unfortunately gloss over too much, which is that the results show that the wavelet method could only characterize units for no more than 50% of the main test reach (#6), and that makes it far inferior to pretty much all other methods one can use. It is unclear if the method was applied to all six reaches or not because of some poor methodological explanations, but if so, then the authors should explain more about how much of each spatial series get a riffle-pool characterization in each case given the different cross-sectional densities in each reach. Given better and more data, it can do more in theory, but it will always have to trim the ends of the series. Still, the real value of wavelet analysis in my viewpoint is that it can provide the required parameters to drive procedural river design of sub reach scale fluctuations in detrended bed elevation and in all the lateral offsets of lateral reach brake (e.g., bank top, floodplain top, Terrance toes and tops, etc.). Such parameterization is the basis of the procedural river generation software my lab group has published called "River Builder", but we have not yet formalized a wavelet parameterization. These authors could make a helpful contribution by using their study results to report parameters that would specify how to make rivers naturally fluctuation in a

non stationary way down their length. That is the exciting potential of their research when mindful directed toward that purpose.

My main concern has more to do with the so-called index method. Conceptually, I have absolutely no problem with the idea of defining riffles and pools on the basis of spatial series of multiple variables, rather than just that of bed elevation. In fact, my group's research articles already do that. The problem is that the choice of variables used in this study (detrended bed elevation, hydraulic radius, and Froude number) seems to me to be technically unsound, because they are highly co-dependent. In fact, both hydraulic radius and Froude number can be derived as a function of detrended bed elevation, while Froude number can also be derived as a function of hydraulic radius as well. Whatever independent information (i.e., unrelated to detrended bed elevation) is contained in hydraulic radius and Froude number is not native to those variable, but comes from the underlying variables that determine them, which, for a fixed discharge along a reach (as the study here is about changes in a variable along the reach), comes down to width, slope, and bed roughness (e.g., surface substrate grain size, form roughness, etc). Together, detrended bed elevation (essentially the inverse of depth but just with a shifted vertical datum), width, slope, and surface roughness entirely define and explain Fr and Rh, as can be derived analytical for a simple channel geometry. For a complex geometry, width is dependent on cross-sectional geometry more generally, but this is a minor technicality- if one has to extract and analyze spatial series to understand riffles and pools, then metrics for depth and width are what matter. To go further, if the river is meandering, the of course one would also want to add the spatial series of thalweg planform curvature, which is a topic the authors do not address even though I think their channels do meander. Meanwhile Fr and Rh are just dependent response variables, so they are not requisite. Further, the significance comes down to the fact that one can use spatial series of detrended bed elevation and width to not only assess locations of riffles and pools, but more importantly subdivide those two landforms in two types each with respect to how they will evolve under the direction of the morphodynamic process of flow convergence routing, per the journal articles my lab group published in

2018 in ESPL, as cited above and below. There are other technical questions as well about the index method I raise below, but the fundamental concern I raise comes down to this issue of the selection of 3 variables that are highly co-dependent. I argue those are the wrong choices and I am based my arguments on physics, whereas the authors provide some exploratory multivariate statistics using PCA to try to substantiate their selections, which doesn't hold up against physics foundations. The fact that one can map riffles and pools with the index simply derives from the fact that we already know one can do that using detrended bed elevation alone, so any variable that is derivative of it will also work. Whether adding the other variables helps more or not, is clouded by the fact that the other variables are not the true independent ones that should be used, but are dependent response variables. Thus, my perspective is that the index method is not sound, whereas the wavelet method is. Therefore, my recommendation to the authors is to cut out the index method form the article and expand the work using the wavelet analysis per my comments because that is the more important method that will carry into the future.

Looking beyond the concepts of the methods, I also find that the methods are not describe well enough for readers to understand them and accept them. I have raised many methodological questions in the detailed comments below. Of course, a manuscript can not explain every nuance, and that is why many people now submit supplementary materials files along with their manuscripts. I am unaware of one for this article and in searching the manuscript text there is no reference to a supplemental file, so I do not think there is one. Before I can really evaluate the full soundness of both methods, there are several fundamental questions that have to be answered by the authors to clarify the data itself (what is it and how was it obtained, for example) and the analysis methods. I recommend moving some of the current text to a supplementary materials file to shorten the main manuscript but then adding in all the details I raise with questions. Remember, if a study cannot be replicated, then it is not science, as is now commonly understood through the devastating replication crisis in science. Ideally, people should be able to apply these methods but to do that they have to be

explained well.

Considering the results of the study, the article focuses on inter-comparison of the two methods and between them and one pre-existing method. That is ok insofar as it goes for a methodological study. However, I always prefer to see a blend between a methodological development and a new scientific development, because why adopt a new method over the old one(s) if it does no better? In other words, what have the authors learned about rivers from the new results they have obtained that might inspire others to care about these methods? If their method is only as good as the older method, then it is hardly significant or worth considering. The new method has to bring something new and valuable. The introductory motivation is purported to be for flood forecast modeling and I do not see anything that suggests to me the new methods offer value toward that end that other methods do not already provide, especially considering the 21st century methods I mentioned above that look at spatially explicit analysis or depth-width geomorphic covariance structures. These newer methods use the same DEMs as modern 2D flood forecasting models, which is why they can have more bearing on that application. As I previously mentioned, I do see value in wavelet parameterization for sub-reach-scale river design though. In short, I think the authors need to add some results and discussion that states the scientific significance of their findings and relates that to research in the literature. This is all the more true when one deletes the index method and focuses on the value in the wavelet analysis. Can anything be learned about riffle-pool rivers especially form wavelet analysis that we do not already know? I am confident the answer is "yes", but the authors need to explain that.

The structure of the manuscript needs to be re-organized to fit the scientific method more strictly, as this would be clearer and better for readers. Right now there is too much blending of introductory text, methods, results, and discussion in many sections, rather than having mindful segregation with a clear "experimental design" to the study and its writing.

Finally, as the authors point out in the manuscript, while some studies are at the fore-front of science with the most-advanced, highest-quality, highest-resolution topographic datasets and analysis methods, the reality is that many rivers around the world are not mapped in such detail and will continue to rely on a small number of cross-sections for the foreseeable future. It may be that for such widespread yet limited datasets, classic approaches from the 1980s and derivative concepts will still be relevant and used. In that light, I think there is value in further pursing this manuscript to see what value it can offer after a major revision accounting for the detailed comments provided here. I'd like to see a major revision and find out where it goes. Can the authors further explain and justify their methods? Can the index method and the specific selection of variables be defended successfully or can the authors amend the method to use better variables with a better outcome? What is the science discovered about the riffles and pools of the 6 rivers studied, or is this just a methods study? I'm open to seeing how the authors go from here and I always think authors deserve the opportunity to respond and produce a better manuscript. I always appreciate when I am given the chance.

-Prof. Greg Pasternack UC Davis

Specific comments:

It is unfortunate that the manuscript does not have continuous line numbering to aid reviewers and editors with referring to locations easily, even the repeating page numbers are only every 5th value, which is not convenient. Actually, based on page 8 where there is new numbering at the onset of section 4, I am totally confused as to how line numbering is done and it makes it harder to review the paper in a discussion format that requires me to write out all my comments rather than simply mark up a manuscript. In future manuscripts, always include full and continuous line numbering.

First 2 paragraphs of the introduction. It seems odd to me that the main reason why anyone should be interested in understanding the sub-reach variability of river topography is because of the potential application of such information to flood forecast model-

ing. Even in the applied realm that is only 1 of many applications that could be referred to. In my own research, the primary motivations are that such data is required for river design for a wide variety of purposes including river rehabilitation and enhancement and also because it informs fluvial ecohydraulics. In light of systemic global ecological collapse, these are more important to society than flood forecasting, in my professional opinion. At a minimum, I think the authors should identify a few more reasons why knowing topographic variability matters and add a citation for each. Also, of course, geomorphologists want to understand it in its own right as a basic scientific question that requires no justification, and of course it is also the case that this variability controls fluvial processes, so the lack of knowledge about it means that we really know little about processes; less than I think most people realize.

P. 2, lines 3-8. While this is generally true, the authors seem to be unaware that my lab group has already published theory and code that is the first to procedurally generate river terrains exactly to specification from the equations and parameters, and this methodology does include sub-reach-scale variability that can go to as high of a frequency as one wants to make it, so quite small scale. There is always more to do, but I think this is relevant to the claim of this paragraph. I see that this paragraph has 4 citations for the first sentence alone, which seems like too many, so removing 1-2 of those could make way for citing this relevant work if the authors agree that what we published does in fact do what they say is an important thing to do, even if not perfectly, but still more than anyone else thus far. The journal citation is Brown, R. A., Pasternack, G. B., Wallender, W. W. 2014. Synthetic river valleys: creating prescribed topography for form-process inquiry and river rehabilitation design. Geomorphology 214: 40-55. 10.1016/j.geomorph.2014.02.025. The code is open-source and free to the world presently coded in R as "River Builder". The R package and user's manual can be downloaded from the CRAN website at https://cran.r-project.org/package=RiverBuilder. The code also includes the Perlin function that can create very small scale features, and that is a common method for generating landscape terrains in the video game and animation industries. In the future we hope to
add the capability to parametrize the sub-reach-scale fluctuations in spatial series of detrended bed elevation and lateral topographic breaklines using wavelet parameterization.

The third paragraph of the introduction serves no required purpose and neither does Figure 1. Both can be deleted with no loss of understanding. Yes, rivers come in different types, but the main thing readers need to know is that this is a study of riffle-pool reaches and that the method can apply to other reaches; these ideas can be promoted without any of this paragraph, as is indicated by the first sentence of the very next paragraph just fine.

p.2, lines 15-16. The objective of what? The writing is unclear here. I disagree that the main purpose of quantitative analysis of channel topography is just to get pool spacing. In support of our River Builder software, one normally wants to analyze many aspects of reach-scale topographic variability so that they can all be parameterized and used to make realistic synthetic rivers. Other important variables would be parameterizations of thalweg planform curvature, base flow and bankfull channel width undulations, flood-plain width undulations, and then how all of these are phased relative to each other (in time series that's "coherence" and "cross-phase"). Thus, pool spacing is certainly one useful data output, but not alone or necessarily most important. I also note that tat the authors never use their reach site results to present any conclusions about the science of pool spacing, so if it is sop important than its value should be evident in how the results are used to advance science.

p. 3, lines 1-4. No need to define wbf twice. Remove one of them.

p. 3, lines 7-17. A major problem with the historic work cited here that its all pre-2001and how it is presented is that the authors are not addressing the equal importance of channel width undulation to channel depth undulation. Richards in the 1970s understood it and wrote about the importance of width. However, because people didn't tend to make width profiles down rivers, the focus wrongly got limited to depth

undulation in the literature of the late 20th century. Of course, authors studying velocity reversal concepts did start to understand this problem pretty well by 1990. With modern high resolution DEMs since 2000, that problem is over and now we are in the era of looking at how depth and width co-vary to control pool and riffle topography and morphodynamics vis-a-vis the "flow convergence routing" mechanism explained by MacWilliams et al (WRR, 2006) and explored further by Prof. Jose Rodriguez in recent WRR papers as well by my lab group in several articles (Sawyer et al., Geomorph., 2010; Brown et al., Env. Man., 2015; Strom et al, Hyd. Proc., 2016; etc). My lab group has published a series of papers on the importance of linked depth and width undulations that has culminated in a new sub-reach scale channel unit classification relevant to this paragraph and this study. See these two articles, the rest leading up to these are cited in them: -Pasternack, G. B., Baig, D., Webber, M., Brown, R. 2018. Hierarchically nested river landform sequences. Part 1: Theory. Earth Surface Processes and Landforms. DOI: 10.1002/esp.4411. -Pasternack, G. B., Baig, D., Webber, M., Brown, R. 2018. Hierarchically nested river landform sequences. Part 2: Bankfull channel morphodynamics governed by valley nesting structure. Earth Surface Processes and Landforms. DOI: 10.1002/esp.4410.

p. 3, lines 20-26. Yes, I agree with all of this, though I don't think wavelet analysis cannot be called "new" as it has been published in geo/hydro journals for decades now; what's new is high quality topo data to apply it to, though that is present intros study. I'm surprised by the citations the authors offer here, as they are not very relevant compared to other options, such as (most importantly) Gangodagamage et al. (Geomorph., 2007) but also Lashermes et al. (WRR, 2007) and McKean et al. (Rem. Sens., 2009). One can use spatially evolutive Fourier analysis and autocorrelation analysis or, if one limits the analysis to a single reach, regular Fourier analysis where the average parameterizations are reasonable. One might even argue that the locations where the Wavelet analysis indicates a change in parameters could be a reach break. Certainly wavelet analysis is a very good way to go for this to objectively delineate reach breaks, but preferably with a multivariate strategy using both depth and width variables. A good

comparison would be to look at the riffle-pool quasi periodicity analyses of Brown, R. A., Pasternack, G. B. 2017. Bed and width oscillations form coherent patterns in a partially confined, regulated gravel–cobble-bedded river adjusting to anthropogenic disturbances, Earth Surface Dynamics, 5, 1-20, doi:10.5194/esurf-5-1-2017.

p.5, lines 19-20. This explanation is incorrect on two levels. First, energy gradient is more than just water surface slope, because energy also accounts for the velocity head that is not in that term. Often velocity isn't changing over long distances or is assumed to not change, but along a riffle crest and in the transition to a pool it definitely changes quite a bit, so strictly speaking one has to account for that. Second, the energy gradient is stage dependent, because the steepest gradient is always associated with the vicinity of the smallest cross-sectional area, all other things being equal. At low discharge the way the authors describe it is true, because at low discharge riffles have the smallest XS area. However, once the discharge exceeds the value for the minimum cross-sectional area of the reach to be elsewhere, then it is not at the riffle any more. At some high flow it will become at the pool location, and of course this is the main reason why pools scour and riffles aggrade to maintain relief in alluvial channels, all other things being equal (especially substrate). This stage dependence is a key issue to account for in any scheme to evaluate where riffles and pools are located and it its why considering only depth and ignoring width has always been a mistake by the river science community. Now that we have width data commensurate to depth data, we can move on to the proper treatment of the problem considering their linked co-variance.

p. 5, lines 22-37. All of these methods retain the limiting viewpoint that they put a primacy on riffles and pools, either ignoring other morphological units (MUs) or treating them as irrelevant. Thankfully, 2D and 3D hydrodynamic modeling ends that mistake and enables objective mapping of all MUs with decision-tree analysis. This approach was explained by Wyrick et al. (Geomorph, 2014) and then applied in Wyrick and Pasternack (Geomorph., 2014) to not only show the greater diversity of MUs beyond riffles and pools, but also to compute simple metrics like pool spacing. Thus, Wyrick and

Pasternack (Geomorph.,2014) presented a novel methodology to extract pool spacing from 2D hydrodynamic model outputs of MUs using GIS tools. That is very relevant to this literature review, because it shows recent progress in automated extraction of this metric. The authors are arguing that their methods are more automated and better than pre-existing methods, but they have not actually considered more recent auto-mated methods. Meanwhile, the sentences about the outstanding work by Almeida and Rodriguez as well as Parker goes off topic from pool spacing to get into the sep-arate topic of riffle-pool morphodynamics, of which there is a very long and illustrious literature not addressed. Best to cut those at this location and stay focused on the directly relevant literature about pool spacing that is the focus of this study. They may be relevant if the revised manuscript ever addresses processes explicitly.

p.6., lines 5-27. Very good literature review and written well, just not accounting for many recent studies since 2001.

p. 7, line 6. The sentence about having surveyed "many" cross-sections is poorly constructed and, in my view, not accurate. Terms like "many" are relative, so it could be that for one person any arbitrarily small number of cross-sections would still seem like many; that makes it hard to argue the point. However, the key metric here is that one cannot analyze for topographically significant spatial frequencies at resolutions smaller than the minimum XS spacing, and that's already quite conservative. For that reason, my lab group uses vastly denser cross-sectional spacing than that used here. For example, in Pasternack et al. (ESPL, 2018b) we used a spacing of 3% of bankfull width. That's "many". For another group, Legleiter (Geomorph., 2014b) spaced a XS every quarter channel width. In contrast, in this study, an analysis of Table 1 finds that cross-sections are spaced between 0.46 to 2.9 times bankfull width, with two reaches not even having 1 XS every bankfull width. These numbers of cross-sections are more like the amount used in a conventional reach survey to obtain reach-average depth and width metrics, not to identify the underlying nature of variability. I think if the authors refer to previously cited articles above about spatial series analysis of rivers topography

plus Legleiter, they'll get a better sense of what is needed to get at the detailed patterns of fluvial topo spatial series at the su-reach scale. This issue doesn't invalidate the study, but just recommends to back off the "many" and get to saying "a normal number of cross-sections typical for a 1D hydraulic modeling study" or something like that. Also, these cited works could be referred to in the discussion section to help compare and contrast undulation metrics from different studies, including when undulations may not have high enough amplitude to become a "riffle" or "pool" but are still big enough to make a difference for the intermediate morphological units that are mentioned but not investigated in this study.

p. 7, line 6. I think a bigger questionmark for the technical soundness has to do with the mindful decision to not have all cross-sections regularly spaced, but to place them primarily at hydraulic controls and morphological breaks. The authors then interpolate to get a grid, but the source data is not uniform. I fully understand why that would be done for a 1D hydraulic modeling study and given perhaps limited resources and no lidar data, but there is no question whatsoever that biased (aka mindful) XS placement impairs and calls into question spatial series analysis as far as objective identification of parameters. By placing the XS where the authors think important hydraulic and mor-phological things are happening, then necessarily the wavelet analysis and any other method is also forced to bias results toward the same outcome of where significant things are happening. On the other hand, when I put an XS every 3% of bankfull width along the series, then there is no chance anything will be missed and the algorithm can decide for itself what the frequencies, amplitudes, and phases (and other parameters) are for that reach. Equal spacing of XS is the best approach for unbiased results. I think there are some things that can still be analyzed with a small number of mindfully selected XS positions, but I would never take this approach. I do understand the lack of availability of lidar and other remote sensing data to facilitate high-resolution mapping though, but then one has to be thoughtful about what one can reasonably achieve. I think the way forward would be for the authors to explain their viewpoint on why they have a sufficient number of XS for the goals of their study in comparison to the highest

density used by the references cited above.

p. 8, section 4, line 1. "Hydrological" should be "hydraulic". I believe. These are not interchangeable. Hydrologic would be rainfall-runoff and water balance related, could be purely discharge but discharge alone does not identify riffles vs pools. Hydraulic means on the basis of the depths, velocities, and other flow kinematics.

p. 8, lines 8-16. I am confused by the writing. On line 8 it says hydraulic data were "surveyed" at 3 discharges. Please clarify that the data were measured in the field and then it is necessary to also describe how the data were measured. There are many different methods possible and one cannot undertake analyses of data without stating how it was collected. Moving on from there, if the data was actually measured, then I have absolutely no idea why the authors mention a method involving 1D hydraulic modeling of the sites. Given field observed cross-sections and hydraulic data, once could use a pure XS analyzer like the old, free software WinXSPro and many other GUIs to extract geometric variables like hydraulic radius with no numerical modeling. If the derivative variables like Rh and Fr are not based on field data, but instead are coming from a 1D hydraulic model, then it opens up a whole can of worms regarding the accuracy of the model outputs, which then necessitates an explanation of model calibration and validation performance. All of this is written unclearly and needs to be revised to explain to readers what is going on. This has profound consequences for evaluating the study.

Section 4. In the previous section it was stated that hydraulic variables were "surveyed" at 2 low discharges and 1 near bankful discharge. As the relative magnitudes of the variables between rifles and pools are stage dependent, it matters which flow was used to for the analysis in this section. The authors should state that. If somehow all three discharges were used, clarify how. In fact, at-a-station hydraulic geometry is an important tool for identifying riffles and pools more holistically considering the totality of the bankfull channel, so it is too bad few people take note of that and apply it fo this purpose.
p. 8, section 4, lines 4-5. Most people use PCA for challenging multivariate problems with complex interrelationships that are unknown and thus this is the first way to get a sense of how variables interrelate. That does not characterize the situation for riffle-poole geometry and open channel hydraulics. A wiser strategy here could be to use Buckingham Pi theorem dimensional analysis to create the variables of interest. Also, one can easily reason out that really the variables that matter are those that control or respond directly to morphodynamic processes, such as flow convergence routing or meander migration. That can then guide wise variable selection that is process based. Returning to this list of variables, several of these variables are highly correlated or define each other, so it does not make sense to throw them all into one multivariate analysis as if it is a mystery. For example, bed elevation, max depth, and hydraulic radius are all highly correlated and redundant. Meanwhile, A and P define Rh, so those 3 are also highly correlated. Similarly, Fr is defined by y and u, so the same situation arises. This "throw everything into the soup" strategy of multivariate analysis is not wise and possibly not technically sound, but the authors can review the PCA assumptions and limitations to evaluate that- not worth my time to re-study up on PCA. Even if it is technically ok, it still doesn't make any sense as a strategy as if we do not already know how these variables relate to each other- we do know exactly how they relate.

p. 8, section 4, line 8. The topic of detrending is a huge issue that requires a bit of unpacking in the writing here, because the outcome of riffle-pool delineation can be largely depending on this very choice based on my own sensitivity analysis of this situation using different detrending methods. Earlier in the manuscripts the authors wisely commented about all the different way different authors measure and analyze pool spacing data (e.g., p.6, line 22). Well, the same challenge arises with detrending. There is no universally right or wrong way given the diversity of purposes for detrending, but each option has consequences for the scientific outcome for a specific purpose, especially for identifying the magnitude and length of residual highs and lows in a bed profile. Without going into all the options, what I request is that the authors state what

type of detrending they did. If linear, then was it one line per site (presumably no reach breaks within a site, but there could be) and was care taken to insure that the line began and ended at the same relative elevation to avoid biasing the slope, which is a significant problem

p.9. I am just not understanding why anything in Figure 4 and the associated results text is actually new results or anything other than trivial findings. By definition of variables, A, Rh, and y are positively correlated, while Fr is going to be negatively correlated to y and positively correlated to u. Also, Z has to be negatively correlated to A, Rh, and y. This is all be definition. PCA is not required to know this. Further, I do not agree that the PCA is adding any fundamentally new or useful information for riffle-pool delineation compared to wisely selecting the few independent variables underlying the physics-based analytical relations, especially bed elevation, width, and possibly slope, as together these three control relative velocity between riffle and pool units for a fixed discharge. If the channels are meandering, then thalweg planform curvature would be important, too, as it is well known in the physics to control meander migration. In fact, it is unclear and technically unsound to exclude metrics of channel width from this analysis, as width is the underlying independent variable influencing all the other variables in the list except for detrended bed elevation and depth (which of course are the same thing just inverted and with different vertical datums). The authors need to set up this methodology better to justify why it is necessary and better than what I am proposing as an easier, more process-based approach or else I do not see how this PCA analysis is meritorious.

p.9, lines 3-4. The claim that each descriptor adds additional information about the bedforms is easy to show as not true. Rh is defined by A and P, so how is Rh fundamentally new and additional as opposed to using a combination of A and P, unless one defines the mathematical operation of division as adding new content, which it does not. This continues the theme of my last few comments. The authors are applying blind statistical methods to what is a pure analytical problem with 100% defined and

known elements. There is no additional information beyond the independent variables and the math operators to combine them into A, Rh, and Fr.

p.9, lines 7-8. These claims apply only to low discharges, due to the flow-dependent nature of riffle-pool hydraulics. How they develop as discharge increases depends on the shape of the cross-sections (especially depth vs width "geomorphic covariance", per flow convergence routing theory.

p.9, lines 10-12. This single long sentence attempting to explain a sequence of mathematical steps applied to some data is opaque to me as a reader, as is plot (a) in Figure 5. This should be written out more thoroughly and clearly in steps. For example, presumably the smoothed data is each XS spatial series, but then what constitutes the "sampling" that is "homogenized"? I neither understand the samples nor what homogenization is and why it is needed. Is homogenization the same or different from normalization in this study? If so, why call it two different things that creates reader confusion, but if not then what is it? Sometimes normalization means the strict application of the function that makes the data fit the normal probability distribution while more often it just means to divide variable by another.

p. 10, first line. Why is this line bold?

p. 10, equation (5). This equation is an all-or-nothing type approach where every location is either classified as riffle or pool for an individual descriptor. This is in contrast to the aforementioned BDT approach that uses a standard deviation tolerance. Also, the method of Pasternack et al. (ESPL 2018a,b) uses a standard deviation tolerance. It would be useful to explain why no tolerance was applied.

p.10, line 10. From what I gather considering the equations and the potential values of I, the concept here is that for something to be defined as a riffle or pool versus an intermediate MU type, all three descriptor variables must agree and yield the same heavyside function value of 0 (pool) or 1 (riffle). Conceptually, the authors are substituting a cross-check among 3 variables as the countermeasure to cope with uncertainty

in place of tolerance within each variable as the countermeasure for uncertainty. I think putting the concept of the method in words like this would help readers understand the strategy and purpose of the math and procedure that is described. However, looking beyond that, one can ask if this actually works? In other words, is there a resiliency against uncertainty gained by using multiple variables and the specific ones chosen? The authors should address why they think this is so, because this is the kernel of new idea they are proposing but have not actually written out. I have to agree that using more than 1 INDEPENDENT variables would help serve as a check against uncertainty, so that is good idea, but (a) the variables chosen are not independent (both Fr and Rh depend on detrended bed elevation, which is a surrogate for the inverse depth and depth goes into both Fr and Rh) and (b) one can choose to use both a tolerance per BDT and multiple variables per this study. That would yield the best outcome. In Pasternack et al. (ESPL, 2018a,b), we do use both strategies, but for our choice of variables we limit our analysis to only detrended bed elevation and width, as these are the process-based controls on flow convergence routing, they are independent, and they underlie the derivative variables like Fr and Rh. However, we do not use slope, which independently controls velocity and Fr, and we make that choose for a specific process-based reason, but we do exclude it. We also do not look at thalweg planform curvature in those articles, though we have internally thus far. One could reasonably choose to include both slope and thalweg planform curvature. One could also choose to include grain size metrics, as I'm sure prof. Jose Rodriguez would be very insistent on given the importance of that variable to determining relative erosion and deposition on riffle sand pools. Unfortunately, it is incredibly difficult to obtain high-resolution spatial series of substrate grain size as of yet. In any case, I see both positive and negative to what is being done. At a minimum, the authors can explain the general idea in words as I have done, but then also some defense is needed if the authors stand by the decision of variables chosen, because I see the choice as technically unsound given that they are defining each other as explained.

p.11, line 2. I see that p.6 line 10 defined lambda-star as "dimensionless pool spacing",

yet here that variable has dimensions of m? Something is wrong.

p.12, lines 1-2. While most people only apply Fourier analysis to stationary series, the method is not in fact limited as thus, because it can be applied using the "evolutive" methodology to capture non stationary dynamics very similar to what one gets from wavelets. One can reasonably argue that wavelets are superior for non stationary data and because one can apply different wave forms, but to say that Fourier analysis cannot do non stationary analysis is wrong. Many applications of evolutive Fourier analysis exist, but for hydrological data see for example, Pasternack, G. B. and Hinnov, L. A. 2003. Hydro meteorological controls on water level in a vegetated Chesapeake Bay tidal freshwater delta. Estuarine, Coastal, and Shelf Science 58:2:373-393.

Section 5.1. I think there is too much redundancy between what was written about wavelets in section 1 (p. 3, lines 20-26) and this section. The introduction can more simply introduce the idea of it and state the scientific questions and hypotheses associated with using it, but then leave the literature review here, so there is only one literature review. My earlier comments about the literature of applying wavelets to geo/hydro data also apply to this section.

p.15, line 14. This sentence makes a key determination that flies against the same kind of decision-making applied to the index method of section 4. Specifically, the determination of riffles and pools is going to rely entirely on bed elevation. It seems odd that scientists who begin with the conjecture that multiple variables should be used to determine riffles and pools would now contradict themselves and only consider one variable. My view of it is that both decisions are arbitrary, as (a) the former was based on a questionably PCA analysis lacking a mechanistic basis and choosing interdependent variables rather than the proper independent ones and (b) the latter is likely based on the amount of work it takes to apply the wavelet methodology and so its application is being limited to only one variables and to only 1 reach instead of all the reaches. I am making my own guess with (b), but the authors provide no justification for limiting the analysis to only 1 reach after introducing so many reaches. Similarly why not do

all three variables the authors deem important. A quick check of the scientific literature confirms that multivariate wavelet analysis exists and is available for use. And then there is the issue of how the variables couple to affect riffle and pool occurrence, structure, and resultant processes. The decision-making here is too opaque and needs explanation per these issues. I expect the decisions cannot be justified, but the authors deserve a chance to try.

p. 15, line 15. Why choose the Orgeval reach, when it is not the longest or having the most XSs? I already deleted my table where I computed the XS density, so does this reach have the highest XS density? Otherwise, why? of course, why not analyze and compare all 6 reaches, as this is a scientific journal article and there could be interesting results in comparing the different reaches? The method itself of applying wavelet analysis to a spatial series is not so novel as to justify limiting to only 1 reach as a single case study.

Figure 8. This figure shows a fundamental problem with the wavelet methodology as the preferred tool for mapping riffle and pools as well as quantifying their spacings. Specifically, it cannot return results for some distance at the start and end of the spatial series. In the case shown, there is only results for the range of $\sim$ 81-241m out of 318 m. That leaves a whopping 50% of the reach unassessed. Wow. That's a lot of lost information. Of course, the longer the series and the more frequent the XS sampling, the less loss, but there will always be a loss. This makes the method less valuable than alternatives that retain the information.

p. 17, line 2. Again, why does lambda-star have units at all- it is supposed to be dimensionless.

p. 17, results header. Some authors like to blend methods and results in paired couplets working through a manuscript, and that is most appropriate when one couplet build son the results of another, but then one would not call section 6 here a results section, as many results have already been presented. If I was the associated editor for

this manuscript, I would require the authors to separate the methods content from the results content and go with the traditional ordering of the scientific method, because there is no reason not to. One can state the methods from sections 4 and 5 in one unified methods section and then state the results in a unified results section. As the two sections do not build on each other, then one does not need to use the couplet approach. Then, one can have methods and results subsections for the inter comparison analyses. Finally, discussion should stand alone after all results are presented.

Section 6.1. Authors must clarify if the score technique is applied to the entire reach length or only the length for which results overlap. I think one must count the whole reach as it is a deficiency of the wavelet method that it leaves 50% of reach 6 unevaluated. Whatever the authors are doing, they should clarify that.

Section 6.2 This section now states that the comparison is limited to only 81<x<241 m. That's problematic because it's not a fair test of the actual utility of the wavelet method leaving half the reach unevaluated. This should be stated clearly. The comparison is still useful but it does have this huge caveat. A method that leaves half the reach unevaluated can never be better than one that assess the whole reach, if the goal is to characterize the whole reach.

Table 4. I do not understand. Previously it was stated twice that only 1 reach was assessed but now here are data comparing all six reaches. I think the writing of the manuscript should be improved to explain what is going on better. If all six reaches were in fact tested with wavelet analysis, then some comparison between reaches would be interesting for section 5.

Discussion section 6.3. These paragraph primarily consistent of more results not previously present, but there is a bit of discussion, too. Specifically, all the text in this section from page 19 line 18 to page 21 line 20 are purely results. In fact, p. 21 line 10 even says, "these results..." so the authors view these as results too. Really, there is no suitable discussion putting the results of this study into the larger context of methods

and results about riffle-pool ID-ing and quantifying their spacings. There should be such a discussion.

Section structure. I think there are problems with the way the manuscript's sections are structured. In general I can follow what the paper is trying to do, but the structure would be better following a traditional scientific method with all methods first then all results second, ands then all actual discussion last. By mixing them all up it is somewhat confusing and more importantly, impossible to tell what methods have answered what important scientific questions. For example, from the structure it is difficult to tell if this study is only a methodological comparison or also a scientific contribution presenting new results about pool spacings that can be compared with the results of other studies. It would be a shame to do all this work and have no contribution to the question of pool spacing in different river types. But getting back to my main concern here, the discussion, if present at all, iOS hidden in bits throughout the manuscript and would work better if isolated and thoroughly presented.

In conclusion, I have put many hours of work into thoroughly inspecting this manuscript to help the authors receive the best quality of feedback I can produce. That has resulted in a lengthy review with many issues raised, which can be demoralizing to authors, but the point of the effort tis to offer about as much discussion of a manuscript as anyone is every going to give about it and also to help the authors produce the best possible scientific article they can. I hope the authors can see what I am trying to do and I hope it will provide substantial value, though it will create more work.

Best regards

-Greg

---

## Author Comment (AC1) · 19 Sep 2019

**Reply to the Anonymous Referee #1**

Dear Referee,

We thank you for your comments, which will help improve the clarity of the manuscript as well as the choice of the methods.

According to both reviews we decided to make very substantial changes to the paper. This work is a methodological study that introduces relatively new wavelet analysis tools in the field of geomorphic analysis (namely, Wavelet Ridge Extraction), in order to identify the pseudo-periodicity of alternating morphological units from a general point of view (and not only pool-riffle morphology). We did initially introduce an index method as a benchmark, but this index was poorly designed due to a lack of physical basis for the choice of the variables. We also neglected some relevant literature on the identification of the morphological units using DEMs, which could be used as benchmark methods in this paper.

For that, we suggest changing the title of this paper to "***Automatic identification of alternating morphological units in river channel using wavelet analysis and ridge extraction***". This will be more general and focuses on the method and not on the pool-riffle morphology.

We have presented two methods in this article. The first one is the wavelet method which represents alternating morphological units (pools and riffles) as pseudo-periodic signals with a continuous wavenumber function K(x). The other one is the index method which is a benchmark method that gives a discrete identification of the morphological units.

With the suggestion of the second reviewer Prof. Gregory Pasternack, we will cut out the index method and replace it with an existing method "Mesohabitat Evaluation Model (MEM)" inspired from Hauer et al. (2009). For that, we will focus only on the wavelet analysis and ridge extraction in the univariate and the multivariate cases and compare its results with two benchmark methods: BDT (O'Neill and Abrahams, 1984) to the bed elevation data and MEM (Hauer et al., 2009) to three variables (velocity, hydraulic radius, and bed shear stress).

We will also minimize the use of modeled variables and apply the methods directly on field measurements (velocity and hydraulic radius variables at the lowest surveyed water level). We will use modeling results (Fluvia model) for bed shear stress only, as the energy slope cannot be determined in a sufficiently accurate manner with the measurements.

For the literature, we missed many recent studies and methods in relation to this work. So first we will add a table that summarizes examples of methods of identifying these morphologies and the variables chosen to do that. Second, we will change and add many recent works especially those working with meter-resolution digital elevation models (DEMs). Finally, we will clearly state the objectives of this study in the abstract and in the introduction.

Another important thing is that we propose a new structure of the paper:

**I- Introduction**:

First, we will state the scope of this study with adding more fields of its application. Second, we will introduce a literature review of metrics, variables used to identify and characterize the alternating morphological units. We will focus on two kinds of numerical criteria computed at reach scale:

- The distribution of spacings between morphological units (mean, mode, etc.),
- After computing the mean values of geometrical and flow properties (velocity, hydraulic radius, bed shear stress, etc.) in each class of morphological units (e.g. pools, riffles, runs, etc.) we will evaluate the covariance matrix of these parameters.

**II- State of art methods for a quantitative assessment of morphological variability within a reach**:

We will present some recent methods and works in the identification of these alternating morphological units (pool-riffle in our case) and state their objectives and limitations. We will start with the Bedform Differencing Technique (BDT, O'Neill and Abrahams, 1984), which is simple but uses bed elevation as the sole variable, and relies on a tolerance criterion on elevation differences. We will then review index methods like Mesohabitat Evaluation Model (MEM, Hauer et al., 2009) which classify each position in the reach into a given discrete morphological unit (pool, riffle, run, plane bed, etc.). These methods rely on expert judgement to define the thresholds that define parameter classes. Finally, geostatistical methods (e.g. Legleiter, 2014) give a continuous description of river channel properties in spatially stationary way, using longitudinal and transverse variography. For these reasons, we are searching for a method that gives a continuous description of geometrical and flow characteristics along the reach with a non-stationary description.

**III- Study objectives**

We will state that this work aims to introduce relatively new wavelet analysis tools in the field of geomorphic analysis, the Wavelet Ridge Extraction, in order to identify the pseudo-periodicity of alternating bedforms from a general point of view. In this study we will use a dataset that presents mainly pool-riffle morphologies, but the method can be applied to any morphology.

We will present the scheme of the paper which include a methodological section of the wavelet analysis and ridge extraction in the univariate and the multivariate cases, a section that presents the comparison method (with defining more explicitly the two benchmark methods: BDT and MEM index), a discussion section, and conclusions.

**IV- Data set and study reaches**:

We will present the six reaches, more explicit information about data collection and about the numerical modeling (Fluvia), and the data that will be used in this study.

**V- Wavelet method**
   **1) Wavelet analysis and ridge extraction**:

We will present a general introduction about wavelets including some methods such as the Wavelet Transform Modulus Maximum (WTMM, Gangodagamage et al., 2007) and other studies using the wavelets in the geomorphological field (Lashermes et al., 2007; McKean et al., 2009). Procedures such as the WTMM (Muzy et al., 1993) consist in extracting components of the signal, but they are not specifically designed to identify pseudo-periodic components in a univariate, let alone in a multivariate case. For this reason, we introduce the procedure called Wavelet Ridge Extraction (Lilly and Olhede, 2009).

**2) Univariate case**

We will present the methodology of this method in the univariate case using one of the three variables (velocity, hydraulic radius, and bed shear stress).

**3) Multivariate case**

We will present the methodology of this method in the multivariate case using the three variables (velocity, hydraulic radius, and bed shear stress).

**VI-      Results**

**1) Comparison method**:

We will define more precisely the two benchmark methods: BDT and MEM index, and their classification of the morphological units (pool-riffle). We will define also the metrics and their computing method.

**2) Application and results**

We will present results of all methods for the six reaches and apply the comparison.

**VII-     Discussion**

We will discuss results (longitudinal spacing, number of morphological units, etc.) with literature and with the two benchmark methods

**VIII-    Conclusions**

Kind regards,

The authors

**Major comments:**

*The manuscript "Wavelet and index methods for the identification of pool–riffle sequences" by Mahdade et al. presents two novel methods for the identification of pools and riffles in natural streams. These methods also allow the assessment of the main geometrical features of pools and riffles. The manuscript states that appropriate geometric description of pools and riffles is pivotal for flood modelling. I think this statement is correct when modelling floods (and flash floods) at the local scale. Conversely, previous*

*studies have shown that simplified representations of river geometry can be a cost-effective solutions for flood modelling at the large (basin to continental scale). In fact, I believe that an accurate representation of river geometry is essential for the implementation of hydrodynamic models used for the investigation of local flow conditions and sediment transport. The scope of the paper could thus be extended to biological and environmental modelling (oxygen exchange, fish habitat, sediment transport) and not only limited to flood forecasting.*

Response:

We agree with you that this statement is correct only at local/small scale, in which we can quantify geometric variability and especially alternating morphological units. In the new version of the paper, we will add examples of application of our study such as the design of a synthetic river topography which is implemented in river restoration (e.g., Wheaton et al., 2004a), habitat modeling, ecohydraulics (e.g., Pasternack and Brown, 2013), biological and environmental modeling (oxygen exchange, fish habitat …) and also that this variability controls fluvial processes as sediment transport, but not focusing only on flood forecasting.

*The paper is interesting, sections 1 and 2 provide a comprehensive literature review; sections 4 and 5 provide a detailed explanation of the methodologies; the presentation and discussion of the results in section 6 is quite extensive. However, I think that a number of major modifications should be introduced before the publication of this study.*

*Firstly, I think that the research gap and the novelty of this study should be clearly stated. Why did the authors propose two novel methods for the identification of pools and riffles? What are the advantages of these two novel methodologies when compared to the existing ones? I believe that these aspects should be clearly stated in the manuscript.*

Response:

The goal of the paper is to introduce a new method for the analysis of river morphology. The rationale behind the method is that the existence of alternating morphological units along a reach (such as pools-riffles sequences, or step-pool etc.) should translate as a *pseudo-periodicity* in geometric and flow variables. Hence, identifying these bedforms amounts to identifying a local wavenumber $K(x)$ and phase $\phi(x)$ for each variable, a task that can be performed by wavelet analysis and especially Wavelet Ridge Extraction (Mallat, 1999; Lilly and Olhede, 2010), in a multivariate framework.

In the initial version of the paper, we were comparing this wavelet-based method with two benchmark methods: the BDT (O'Neill and Abrahams, 1984), and an index method that consists in affecting a different numerical value for each class of a given variable/degree of freedom, and then sum these individual index functions into a composite one.

The second reviewer Prof. Gregory Pasternack has raised major concerns not about the index method in itself, but on the choice of the variables/degrees of freedom. Initially we used the first three axes of a Principal Component Analysis as the degrees of freedom, a choice which has very little physical meaning. We will entirely change this choice and build the index using the same variables/degrees of freedom as

in the existing "Mesohabitat Evaluation Model" (MEM, Hauer et al., 2009), which uses velocity, hydraulic radius (or the closely related cross-sectional averaged depth), and bed shear stress. We will also use the same threshold values for classifying each variable.

However, we intend to keep the last benchmark, the BDT method, as it is in the current version of the paper.

*Second, the results of the new methods are compared to the results of the BDT method. Is the BDT method used as benchmark or to validate the new methods? Is the BDT method considered more accurate than the new methods? If so, why? What are the advantages of using the two methods rather than using the BDT method? Would it be possible to validate the results of this study using field data?*

Response:

In this study, we consider the BDT method as a benchmark method. We do not consider a specific method to be the "true" or "reference" one, we only apply several methods to have a general idea on the uncertainties in the identification of morphological units. That being said, there is a substantial difference between the BDT and index methods on one side, and the wavelet ridge extraction on the other side:

- BDT and index methods classify each position in the reach into a given category (pool, riffle, run, plane bed, etc.); hence, in 1D, we have access to a discrete values of bedform lengths $L_i$ (i=1…N), and we can compute statistics of this discrete distribution such as **mean**, **mode**, **n-th order moments**, etc.;
- In contrary, the wavelet ridge extraction provides a continuous description of bedform spacing along the reach, through a continuous wavenumber function K(x). In turn, we can compute the statistics (again, mean, mode, n-th order moments, etc.) of this function in order to compare them with the values obtained in a discrete method.

Moreover, index methods use expert judgement in order to specify threshold values for each variable/degree of freedom. Since wavelet analysis is continuous in nature, such thresholds are not needed in our method.

*Third, the computation of the index method relies on the results of the numerical model. Have the authors considered the impact of the uncertainties in the results of the numerical model on the results of the index method?*

Response:

The use of model outputs is indeed a questionable choice that may add a lot of uncertainty in the results. The purpose of the numerical model used in the previous studies by Navratil et al. (2006) was simply to generate water surface profiles for discharge values other than the surveyed ones (i.e., interpolate/extrapolate the rating curves). In our revised paper, we will solely rely on measurements at the lowest surveyed discharge. However, since the calibration of the FLUVIA model on the reaches provides estimates of Strickler roughness coefficient $K_s$, we will use these $K_s$ in order to compute the

third degree of freedom, bed shear stress $\tau_b(x)$, along the reach: even if partly relies on calibration, it seems a more robust way of computing $\tau_b$ than through the finite differentiation of the total head function $U^2/2g + z_{surface}$ between adjacent cross-sections to get the energy slope J.

*Furthermore, I suggest discussing the transferability of the new methods to other reaches. In other words, how easy would be to implement the proposed methodologies to other study areas? Are the data and algorithms required easy to collect and implement? Can other researchers implement the proposed methodologies?*

Response:

As stated previously, the wavelet methods intends to be quite general and can be applied in any morphology that presents alternating bedforms (pool-riffle, step-pool, etc). The code comes in the form of a small number of Matlab functions, and the data has to be provided as values of flow variables sampled at successive locations along the reach. The choice of the set of variables/degrees of freedom is up to the user, in our case we chose the triplet [$U(x)$ , $Rh(x)$, $\tau_b(x)$] but we could pick other variables, and add planform variables as well.

*Moreover, I think the manuscript should clearly state which methods are recommended. A more explicit presentation of the conclusions of this study would highlight its scientific and practical relevance.*

*Regarding the structure of the paper, I would like to recommend two modifications: - Section 2 lists a large number of studies and it is a bit hard to follow. More specifically, I think it is difficult to appreciate the differences between the large number of criteria listed in this section. The authors might consider adding a table to summarise their literature review.*

*I hope the authors will find my questions and recommendations useful to improve their manuscript.*

Response:

As said before, the structure of the paper will be changed by splitting the results and discussion and adding a conclusions part, the later one will specify the added value of the wavelet method according to the comparison in the discussion section. In fact, we will compare the metrics computing (mean, mode, n-th order moments, etc of the distribution) using the three methods.

**Minor comments:**

I listed below a number of minor recommendations.

| Comment of the reviewer | Response of the authors |
|---|---|
| *Page 1, lines 7-8: the sentence "To better take this high-frequency variability in bedforms into account in hydraulic models" is a bit convolute. The authors might consider improving the structure of this sentence.* | We will replace it with: "To include/consider this high-frequency variability of the geometry in the hydraulic models" |
| *Page 1, abstract: the abstract should clearly state* | We will change the abstract by including that this |

| | |
|---|---|
| *the research gap, the aim, and the novelty of the study.* | work is a methodological study that introduces relatively new wavelet analysis tools in the field of geomorphic analysis (namely, Wavelet Ridge Extraction), in order to identify the pseudo-periodicity of alternating morphological units from a general point of view (and not only pool-riffle morphology).
We will state clearly the aim of this paper which is for example extracting some quantitative properties of these alternating bedforms such as the mean and the mode of their longitudinal spacings, with a "continuous" vision of the topography instead of a discrete classification. |
| *Page 1, line 9: the abstract mentions "several methods", however, only three (two novel methods and one benchmarking method) are listed explicitly.* | As stated above, we will clarify the presentation: we introduce one new method (wavelet ridge extraction) and we compare the results with two existing methods (BDT and MEM index method). |
| *Page 1, lines 12-13: the authors might consider avoiding the repetition of the word "compared".* | Corrected |
| *Page 2, lines 14-15: I am not sure whether this is the final format of the paper, however, I suggest positioning each figure after a full stop (Figure 1 is currently positioned in the middle of a sentence).* | This is not the final format of the paper. We will change that in the revised version. |
| *Page 2, line 15: please correct "dimensionless reach wavelength".* | Corrected |
| *Page 3, lines 4-5: this sentence is a bit hard to understand. Do the authors mean that the overarching purpose of their study is to provide a methodology for the prediction / modelling / assessment of cross sections variability?* | We will cut out this sentence and change from line 14 to 16 in page 2 with: "In this study, we focus mainly on alluvial pool-riffle sequences, even though the method presented here could be used to analyze any alternate morphological units. The objective is to provide a continuous description of geometric and flow patterns along a reach, a description that could be subsequently used to create a synthetic river as in the River Builder (Brown et al., 2014). To do that we calculate the dimensionless reach wavelength, which is the distance …". |
| *Page 3, line 8: words such as "methods" or "techniques" might be more appropriate than "studies".* | Corrected |
| *Page 3, line 11: could please the authors clarify the meaning of "descriptions of the water surface characteristics"? Is "water surface slope" (mentioned in Line 8) included in this latter category?* | Descriptions of the water surface characteristics means a method that describe pools and riffles from the combination of all characteristics of the water surface (water surface elevation, water surface slope, etc.) and which include effectively the slope as Mosely (1982) mentioned in his paper. We also corrected the reference.
*"Mosley, M. P. (1982). Procedure for characterising* |

| | |
|---|---|
| | *river channels, Water Soil Misc.”*
Instead of
*“Mosley, M. P.: Analysis of the effect of changing discharge on channel morphology and instream uses in a braided river, Ohau River, New Zealand, Water resources research, 18, 800–812, 1982.”* |
| *Page 3, line 14: I suggest clarifying the sentence “because it changes less with discharge”.* | It means that this morphological definition of pool-riffle sequences doesn't depend on discharge. |
| *Page 3, line 20: please rephrase “goes with the notion”.* | We will change it with “involves the use” |
| *Page 3, line 22: please rephrase “allows one to extract”.* | We will change it with “extract” |
| *Page 3, line 30: please rephrase “using a threshold on a criterion index.”* | We decided to cut out this method but we will use instead of it a method that uses thresholds on 3 variables. We explained that before. |
| *Page 4, Figure 2(A): I believe that this figure is not mentioned in the text.* | Corrected, we will mention it in the page 3, line 26 |
| *Page 4, lines 3-4: I think this sentence should be moved to the section 6.2 as it motivates the choice of the benchmarking method.* | We will move this sentence to the comparison methods section and modify it according the new structure of paper. |
| *Page 5, line 7: “the areal difference asymmetry index by Knighton” has not been mentioned before, the authors might consider adding more context to this statement.* | It was felt that there is no need to define this method because it's just an example of methods existing in the literature. However, we will add it in the table that summarizes all the previous methods and techniques. |
| *Page 5, line 32: the manuscript states: “a common geomorphological and hydrological” methods, I suggest specifying these methods.* | We will change the entire sentence because it wasn't clear enough (from line 32 to line 34): “Krueger and Frothingham (2007) identified pools and riffles in fifteen reaches of Ransom Creek, Clarence, New York with methods used in two different disciplines, geomorphology (BDT) and hydrology (Froude Number method), and compared their identification agreement.” |
| *Page 6, line 8: was the channel width/channel bankfull width used to compute dimensionless values of wavelength? I think this sentence is not clear.* | Yes, it's not clear. Here we are talking about the dimensionless pool spacing, in which there are researchers who use the definition $\lambda^* = \frac{\lambda}{W}$ (mean channel width) while others use $\lambda^* = \frac{\lambda}{W_{bf}}$ (mean bankfull channel width). We will change it in the revised version. |
| *Page 6, line 9: what do the authors mean with “certainty”?* | “certainty of these ratios” means their efficiencies to give more consistent results, so we will change it to “efficiency” |
| *Page 6, line 14: I suggest avoiding colloquial expressions such as “a great deal”.* | We changed it to: “Some researches have investigated” |
| *Page 6, line 32: I suggest rephrasing this sentence and avoid the use of “we”.* | We will change all the sentence to: “These reaches contains mainly pool-riffle morphologies, they |

| | |
|---|---|
| | have slopes …" |
| Page 7, line 5: I believe that information on slope has been previously provided in page 6, line 32. Could please the authors explain the added value of this sentence? | This line has been added to define the thalweg elevation and how it can be estimated. That's why we will delete it and add this information in the line 32 p6: "they have slopes less than 0.015 (estimated from the thalweg elevation which is the lowest point in the section) …" |
| Page 8, line 11: please clarify the sentence "It is based on interpolations rather than extrapolations". | As said before, the role of the 1D hydraulic model (Fluvia) was simply to generate water surface profiles for discharge values other than the surveyed ones (i.e., interpolate the rating curves between values of surveyed discharges, and extrapolate slightly above highest surveyed discharge). In our revised paper, we will solely rely on measurements at the lowest surveyed discharge and use the model to provide estimates of Strickler roughness coefficient $K_s$, we will use these $K_s$ in order to compute the third degree of freedom, bed shear stress $\tau_b(x)$, along the reach. So this part form line 10 to line 14 will be modified by an explicit description of the model and the data set. |
| Page 8, line 13: "visually": do the authors mean that they performed a manual calibration of the hydraulic model? | It is a typo, we checked the calibration visually, but we adjusted it with a minimization function. |
| Page 8, line 14: please remove the second full stop. | Corrected. |
| Page 8, line 14: "multi-section flows": do the authors mean that the numerical model is used to predict a number of quantities (e.g. the elevation of the water surface, wetted perimeter, wetted surface,: : :) at a number of cross sections? | The use of the numerical model (Fluvia) will be simply to generate calibrated estimates of Strickler roughness coefficient $K_s$ that we will use to compute the bed shear stress $\tau_b(x)$ along the reach. For the other cross-section variables, we will use only measurements at the lowest surveyed discharge. |
| Page 8, line 3: why is the minimum discharge used for the implementation of the method? | We chose the minimum discharge (low flow) in the development of the method because it is the discharge through which we can visualize the variability of the bathymetry (alternating morphological units). |
| Page 8, line 6: does "it" stand for "relevant information"? The authors might consider editing the structure of this sentence. | This section will be removed from this paper as suggested by the second reviewer Prof. Gregory Pasternack and we will replace it by a benchmark method in the comparison methods part. |
| Page 8, line 8: I believe that "the trend" has not been explained before. I suggest clarifying this sentence. What does "detrended variables" mean? How are these variables computed? | The only detrended variable was bed elevation: we computed a series of bed elevation anomalies $\varepsilon_z(x)$ such that: $z_{bed}(x) = -S_{bed} x + b + \varepsilon_z(x)$, where $S_{bed}$ is the mean slope of the reach and $\varepsilon_z(x)$ has zero mean. This part is not necessary anymore. |
| Page 8, line 13: "contain the most explained | The PCA analysis will be completely removed so |

| | |
|---|---|
| _variances" do the authors mean that those directions can explain the variability of the data? I suggest clarifying this sentence._ | this discussion is not relevant anymore. |
| _Page 8, lines 18-20: does these results confirm/contradict previous studies?_ | The PCA analysis will be completely removed so this discussion is not relevant anymore. |
| _Page 9, figure 4: could please the authors explain the meaning of Dimension 1: : :9?_ | The PCA analysis will be completely removed so this discussion is not relevant anymore. |
| _Page 9, lines 5-6: I think this sentence is unclear. What is the relationship between bed elevation and hydraulic radius? The statement seems to be contradictory. Moreover, I was wondering whether any correlation between bed elevation and hydraulic radius is meaningful. Bed elevation is a geometric characteristic at the point scale. The hydraulic radius depends on discharge, river bed slope, cross section area._ | Here we are not talking about the physical meaning of these variables but their variability. The hydraulic radius is the cross-sectional area divided by the wetted perimeter, so the hydraulic radius, the cross-sectional area, and the depth are positively correlated, while the water surface elevation is the depth plus the bed elevation, so the depth and the bed elevation are negatively correlated. So the bed elevation and the hydraulic radius are negatively correlated. It's just trivial findings. For that we will choose in the revised paper variables that are not dependent. |
| _Page 9, lines 6-8: the explanation based on hydraulic radius and Froude number is reasonable and (almost) intuitive. I suggest to clarify the added value of this finding compared to the existing literature._ | There is no added value of this finding, we were wrong about the justification of our choice of variables. We will change all this section as we mentioned it before. |
| _Page 9, line 9: I suggest clarifying the importance of bed elevation._ | Historically, bed elevation has been seen as the most relevant variable in order to characterize geometric and flow variability. Since water surface elevation cannot change in space as fast as bed elevation, local bed elevation (and slope) is an important driver of depth and velocity variations along the reach. However, width variations have been found to be important as well, so a multivariate approach must clearly be favored. |
| _Page 9, line 10: what do the authors mean with "we smooth" the data?_ | The formulation was wrong; in fact the processing mentioned in this sentence was only applied to bed elevation: since the trend of bed elevation is not necessarily linear, we performed a more general removal of very low frequency components (wavelength larger than 7 times the mean bankfull width) before applying thresholds. Since we will not use bed elevation anymore in the index method (MEM), this processing is no longer relevant (and it was not a smoothing anyway). |
| _Page 12, lines 11-13: I suggest improving the readability of this sentence._ | We suggest another clear sentence: "They are functions used in representing data by processing it at different scales or resolutions. If we look at a signal with a large –window-, we would notice gross features. Similarly, if we look at a signal with |

| | a small –window-, we would notice small features. The result in wavelet analysis is to see both of them (Graps, 1995)." |
|---|---|
| *Page 12, lines 16-18: please improve the structure of this sentence: "have been interested: : :but working", both the verbs have the same subject.* | Instead of "but working" we will put: "but they have been working" |
| *Page 12, line 18: "analysis" is repeated.* | Corrected. |
| *Page 14, line 12: I suggest replacing "evacuate" with something more appropriate (an option could be "remove").* | Corrected. |
| *Page 15, line 3: please clarify "It also represents" (what does "it" stand for?)* | "it" stands for "the curve that continuously crosses the domain" and also "K(x)". We will replace it by "This curve K(x) also represents …" |
| *Page 15, line 11: could please the authors better explain why this correction is applied?* | Equation (21) actually gives the amplitude of the pseudo-periodic signal through inverse wavelet transform. In this reverse transformation we need to multiply by $\sqrt{s} = \sqrt{\frac{1}{\alpha K(x)}}$ where $\alpha$ is the Fourier factor (Torrence and Compo, 1998), since we multiply by $\sqrt{\frac{1}{s}} = \sqrt{\alpha K(x)}$ in the direct transformation (Equation 14). |
| *Page 15, line 15: please correct the structure of this sentence: "we limit the study only with univariate analysis". Moreover, could please the authors justify this choice?* | As we said before, we will focus in the revised version of this paper on both the univariate and the multivariate analysis and we will compare their results with the BDT (O'Neill and Abrahams, 1984) and the MEM (Hauer et al., 2009). |
| *Page 15, line 26: please clarify the meaning of "multivariate case".* | The multivariate case is the extension of the univariate to a set of N real-valued signals; it is described in Lilly and Olhede (2009). We will describe this case and develop its transformations in our revised version. |
| *Page 17, Table 3: Table 3 and Table 2 show the results of the two methods for the same river reach. The authors might consider displaying these tables in the same page in order to allow a straightforward comparison of the results.* | We will display these tables to the results section. |
| *Page 18, lines 2-4: I think that this sentence is unclear.* | We will replace it with a clear sentence according to the new results that we will have. |
| *Page 19, line 3: I believe that these results demonstrate a good level of agreement between the two methods. In my opinion, these results do not provide explicit information on the accuracy of the methods.* | Yes, these results do not prove the accuracy of the methods. For that, we chose presenting one method (wavelet method) and discuss it with two benchmark methods. |
| *Page 19, line 5: the BDT methods is used to "validate" the results of the proposed* | In this study, we consider the BDT method as a benchmark method. We do not consider a specific |

| | |
|---|---|
| *methodologies. This choice implies that the BDT method is more accurate than the new methods introduced in this manuscript. Is this correct? If so, what are the benefits/ advantages of using the two proposed techniques?* | method to be the "true" or "reference" one, we only apply several methods to have a general idea on the uncertainties in the identification of morphological units. |
| *Page 19, lines 19-20: please clarify this sentence.* | We will delete this sentence in the revised version. |
| *Page 19, line 23: the manuscript states that the results of BDT "are closer to the other methods and to reality". I strongly recommend to better substantiating this sentence. Which are the "other methods"? What does "reality" mean? Was the BDT method compared with field data? In which case study?* | True, This sentence is not clear, we will delete it. |
| *Page 20, lines 3-4: could please the authors clarify this sentence?* | We will change all the discussion according to the results that we will have. |
| *Page 20, lines 6-7: please rephrase this sentence.* | This sentence is not clear, we will change it in the revised version. |
| *Page 21, line 2: a Froude number of 0.30 looks a bit large. Could please the authors explain this result?* | We will dismiss the Froude number In the revised version. But for example in the study of Jowett (1993) and Clifford et al. (2006), they found values close to 0.3, so we think that these values are a bit large but acceptable.

Fig. 1 Kernel density functions of velocity/depth ratio, Froude number, and water surface slope in pool, run, and riffle habitats. |
| *Page 21, line 3: it seems that the average values are driven by the results of the Graulade river, Are the average values representative of the sample?* | If we exclude the Graulade river we will find and average of 0.20 for the index and 0.17 for the wavelet method. These results are nearly close the 0.23 and 0.20. |
| *Page 21, line 10-17: these lines present a comparison between the results of this study and some of the previous studies. The authors might (or might not) consider using a table to summarise these comparisons.* | This is a good idea. |
| *Page 21, line 21: I suggest motivating this sentence. Why aren't the previous methods considered quantitative?* | We will delete all this sentence |
| *Page 22, line 1: is "crossing" the most appropriate word?* | We will delete all these sentences |
| *Page 22, line 3: please clarify this sentence* | We will change this sentence in the revised version |

**References:**

Brown, R. A., Pasternack, G. B., Wallender, W. W. 2014. Synthetic river valleys: creating prescribed topography for form-process inquiry and river rehabilitation design. Geomorphology 214: 40-55. 10.1016/j.geomorph.2014.02.025

Clifford, N. J., Harmar, O. P., Harvey, G., & Petts, G. E. (2006). Physical habitat, eco-hydraulics and river design: a review and re-evaluation of some popular concepts and methods. Aquatic conservation: marine and freshwater ecosystems, 16(4), 389-408.

Gangodagamage, C., Barnes, E., & Foufoula-Georgiou, E. (2007). Scaling in river corridor widths depicts organization in valley morphology. Geomorphology, 91(3-4), 198-215.

Graps, A.: An introduction to wavelets, IEEE computational science and engineering, 2, 50–61, 1995.

Hauer, C., Mandlburger, G., & Habersack, H. (2009). Hydraulically related hydro-morphological units: description based on a new conceptual mesohabitat evaluation model (MEM) using LiDAR data as geometric input. River Research and Applications, 25(1), 29-47.

Jowett, I. G.: A method for objectively identifying pool, run, and riffle habitats from physical measurements, New Zealand journal of marine and freshwater research, 27, 241–248, 1993.

Krueger, A. and Frothingham, K.: Application and comparison of geomorphological and hydrological pool and riffle quantification methods, Geogr Bull, 48, 85–95, 2007.

Lashermes, B., & Foufoula-Georgiou, E. (2007). Area and width functions of river networks: New results on multifractal properties. Water Resources Research, 43(9).

Legleiter, C. J. (2014). A geostatistical framework for quantifying the reach-scale spatial structure of river morphology: 2. Application to restored and natural channels. Geomorphology, 205, 85-101

Lilly, J. M. and Olhede, S. C.: Wavelet ridge estimation of jointly modulated multivariate oscillations, in: Signals, Systems and Computers, 2009 Conference Record of the Forty-Third Asilomar Conference on, pp. 452–456, IEEE, 2009.

Lilly, J. M., & Olhede, S. C. (2010). On the analytic wavelet transform. IEEE Transactions on Information Theory, 56 (8), 4135–4156.

Mallat, S., A wavelet tour of signal processing, 2nd edition. New York: Academic Press, 1999

McKean, J., Nagel, D., Tonina, D., Bailey, P., Wright, C. W., Bohn, C., & Nayegandhi, A. (2009). Remote sensing of channels and riparian zones with a narrow-beam aquatic-terrestrial LIDAR. Remote Sensing, 1(4), 1065-1096.

Mosley, M. P. (1982). Procedure for characterising river channels, Water Soil Misc.

Muzy, J. F., Bacry, E., & Arneodo, A. (1993). Multifractal formalism for fractal signals: The structure-function approach versus the wavelet-transform modulus-maxima method. Physical review E, 47(2), 875.

Navratil, O., Albert, M., Herouin, E., and Gresillon, J.: Determination of bankfull discharge magnitude and frequency: comparison of methods on 16 gravel-bed river reaches, Earth Surface Processes and Landforms, 31, 1345–1363, 2006.

O'Neill, M. P., & Abrahams, A. D. (1984). Objective identification of pools and riffles. Water resources research, 20(7), 921-926.

Pasternack, G. B., & Brown, R. A. (2013, January). Ecohydraulic Design of Riffle-Pool Relief and Morphological Unit Geometry in Support of Regulated Gravel-Bed River Rehabilitation. In Ecohydraulics (p. 337).

Torrence, C., and G. P. Compo (1998), A Practical Guide to Wavelet Analysis, Bull. Amer. Meteor. Soc., 79, 61–78.

Wheaton, J.M., Pasternack, G.B., Merz, J.E., 2004a. Spawning habitat rehabilitation - 1. Conceptual approach & methods. International Journal of River Basin Management 2(1), 3-20.

---

## Author Response (AR1)

**Reply to the Anonymous Referee #1**

Dear Referee,

We thank you for your comments, which will help improve the clarity of the manuscript as well as the choice of the methods.

According to both reviews we decided to make very substantial changes to the paper. This work is a methodological study that introduces relatively new wavelet analysis tools in the field of geomorphic analysis (namely, Wavelet Ridge Extraction), in order to identify the pseudo-periodicity of alternating morphological units from a general point of view (and not only pool-riffle morphology). We did initially introduce an index method as a benchmark, but this index was poorly designed due to a lack of physical basis for the choice of the variables. We also neglected some relevant literature on the identification of the morphological units using DEMs, which could be used as benchmark methods in this paper.

For that, we suggest changing the title of this paper to "***Automatic identification of alternating morphological units in river channel using wavelet analysis and ridge extraction***". This will be more general and focuses on the method and not on the pool-riffle morphology.

We have presented two methods in this article. The first one is the wavelet method which represents alternating morphological units (pools and riffles) as pseudo-periodic signals with a continuous wavenumber function K(x). The other one is the index method which is a benchmark method that gives a discrete identification of the morphological units.

With the suggestion of the second reviewer Prof. Gregory Pasternack, we will cut out the index method. We will focus only on the wavelet analysis and ridge extraction in the univariate and the multivariate cases and compare its results with the benchmark method: BDT (O'Neill and Abrahams, 1984) to the bed elevation data. We didn't compare it with other recent methods (e.g. Hauer et al., 2009; Wyrick et al., 2014) because they require thresholds (expert judgment) collected from the field, which is not possible in our case.

We will also minimize the use of modeled variables and apply the methods directly on field measurements (velocity, hydraulic radius variables at the lowest surveyed water level and planform curvature angle). We will use modeling results (Fluvia model) for bed shear stress only, as the energy slope cannot be determined in a sufficiently accurate manner with the measurements.

For the literature, we missed many recent studies and methods in relation to this work. So first we will add a table that summarizes examples of methods of identifying these morphologies and the variables chosen to do that. Second, we will change and add many recent works especially those working with meter-resolution digital elevation models (DEMs). Finally, we will clearly state the objectives of this study in the abstract and in the introduction.

Another important thing is that we propose a new structure of the paper:

**I- Introduction**:

First, we will state the scope of this study with adding more fields of its application. Second, we will introduce a literature review of metrics, variables used to identify and characterize the alternating morphological units. We will focus on two kinds of numerical criteria computed at reach scale:

- The distribution of spacings between morphological units (mean, mode, etc.),
- After computing the mean values of geometrical and flow properties (velocity, hydraulic radius, bed shear stress, etc.) in each class of morphological units (e.g. pools, riffles, runs, etc.) we will evaluate the covariance matrix of these parameters.

**1) State of art methods for a quantitative assessment of morphological variability within a reach**:

We will present some recent methods and works in the identification of these alternating morphological units (pool-riffle in our case) and state their objectives and limitations. We will start with the Bedform Differencing Technique (BDT, O'Neill and Abrahams, 1984), which is simple but uses bed elevation as the sole variable, and relies on a tolerance criterion on elevation differences. We will then review index methods like Mesohabitat Evaluation Model (MEM, Hauer et al., 2009) which classify each position in the reach into a given discrete morphological unit (pool, riffle, run, plane bed, etc.). These methods rely on expert judgement to define the thresholds that define parameter classes. Finally, geostatistical methods (e.g. Legleiter, 2014) give a continuous description of river channel properties in spatially stationary way, using longitudinal and transverse variography. For these reasons, we are searching for a method that gives a continuous description of geometrical and flow characteristics along the reach with a non-stationary description.

**2) Study objectives**

We will state that this work aims to introduce relatively new wavelet analysis tools in the field of geomorphic analysis, the Wavelet Ridge Extraction, in order to identify the pseudo-periodicity of alternating bedforms from a general point of view. In this study we will use a dataset that presents mainly pool-riffle morphologies, but the method can be applied to any morphology.

We will present the scheme of the paper which include a methodological section of the wavelet analysis and ridge extraction in the univariate and the multivariate cases, a section that presents the comparison between our method and the BDT, a discussion section, and conclusions.

**II- Data set and study reaches**:

We will present the six reaches, more explicit information about data collection, planform curvature angle computing and about the numerical modeling (Fluvia).

**III-    Wavelet method**

**1) Wavelet analysis and ridge extraction**:

We will present a general introduction about wavelets including some methods such as the Wavelet Transform Modulus Maximum (WTMM, Gangodagamage et al., 2007) and other studies using the wavelets in the geomorphological field (Lashermes et al., 2007; McKean et al., 2009). Procedures such as the WTMM (Muzy et al., 1993) consist in extracting components of the signal, but they are not specifically designed to identify pseudo-periodic components in a univariate, let alone in a multivariate case. For this reason, we introduce the procedure called Wavelet Ridge Extraction (Lilly and Olhede, 2009).

**2) Univariate case**

We will present the methodology of this method in the univariate case using one of the four variables (velocity, hydraulic radius, bed shear stress, and planform curvature angle).

**3) Multivariate case**

We will present the methodology of this method in the multivariate case using the four variables (velocity, hydraulic radius, bed shear stress, and planform curvature angle).

**IV-    Results**

**1) Univariate vs Multivariate**:

We will compare the univariate with the multivariate results with computing some statistics. And we will use the multivariate wavelength to model the bed elevation of the reaches without using it as a variable.

**2) Comparison with the benchmark method**

We will compare our method's results in the multivariate case with the selected benchmark method from the literature: BDT.

**V-    Discussion**

We will discuss results (longitudinal spacing, number of morphological units, etc.) with literature and with the benchmark method.

**VI-    Conclusions**

Kind regards,

The authors

**Major comments:**

*The manuscript "Wavelet and index methods for the identification of pool–riffle sequences" by Mahdade et al. presents two novel methods for the identification of pools and riffles in natural streams. These methods also allow the assessment of the main geometrical features of pools and riffles. The manuscript states that appropriate geometric description of pools and riffles is pivotal for flood modelling. I think this statement is correct when modelling floods (and flash floods) at the local scale. Conversely, previous studies have shown that simplified representations of river geometry can be a cost-effective solutions for flood modelling at the large (basin to continental scale). In fact, I believe that an accurate representation of river geometry is essential for the implementation of hydrodynamic models used for the investigation of local flow conditions and sediment transport. The scope of the paper could thus be extended to biological and environmental modelling (oxygen exchange, fish habitat, sediment transport) and not only limited to flood forecasting.*

Response:

We agree with you that this statement is correct only at local/small scale, in which we can quantify geometric variability and especially alternating morphological units. In the new version of the paper, we will add examples of application of our study such as the design of a synthetic river topography which is implemented in river restoration (e.g., Wheaton et al., 2004a), habitat modeling, ecohydraulics (e.g., Pasternack and Brown, 2013), biological and environmental modeling (oxygen exchange, fish habitat …) and also that this variability controls fluvial processes as sediment transport, but not focusing only on flood forecasting.

*The paper is interesting, sections 1 and 2 provide a comprehensive literature review; sections 4 and 5 provide a detailed explanation of the methodologies; the presentation and discussion of the results in section 6 is quite extensive. However, I think that a number of major modifications should be introduced before the publication of this study.*

*Firstly, I think that the research gap and the novelty of this study should be clearly stated. Why did the authors propose two novel methods for the identification of pools and riffles? What are the advantages of these two novel methodologies when compared to the existing ones? I believe that these aspects should be clearly stated in the manuscript.*

Response:

The goal of the paper is to introduce a new method for the analysis of river morphology. The rationale behind the method is that the existence of alternating morphological units along a reach (such as pools-riffles sequences, or step-pool etc.) should translate as a *pseudo-periodicity* in geometric and flow variables. Hence, identifying these bedforms amounts to identifying a local wavenumber $K(x)$ and phase $\phi(x)$ for each variable, a task that can be performed by wavelet analysis and especially Wavelet Ridge Extraction (Mallat, 1999; Lilly and Olhede, 2010), in a multivariate framework.

In the initial version of the paper, we were comparing this wavelet-based method with two benchmark methods: the BDT (O'Neill and Abrahams, 1984), and an index method that consists in affecting a different numerical value for each class of a given variable/degree of freedom, and then sum these individual index functions into a composite one.

The second reviewer Prof. Gregory Pasternack has raised major concerns not about the index method in itself, but on the choice of the variables/degrees of freedom. Initially we used the first three axes of a Principal Component Analysis as the degrees of freedom, a choice which has very little physical meaning. We will entirely ignore this choice and use a classical variables/degrees of freedom, which are velocity, hydraulic radius (or the closely related cross-sectional averaged depth), bed shear stress in addition to the new variable: planform curvature angle.

However, we intend to keep the last benchmark, the BDT method, as it is in the current version of the paper instead of adding other recent methods (e.g. Hauer et al., 2009; Wyrick et al., 2014) because they require thresholds (expert judgment) collected from the field, which is not possible in our case.

*Second, the results of the new methods are compared to the results of the BDT method. Is the BDT method used as benchmark or to validate the new methods? Is the BDT method considered more accurate than the new methods? If so, why? What are the advantages of using the two methods rather than using the BDT method? Would it be possible to validate the results of this study using field data?*

Response:

In this study, we consider the BDT method as a benchmark method. We do not consider a specific method to be the "true" or "reference" one, we only apply several methods to have a general idea on the uncertainties in the identification of morphological units. That being said, there is a substantial difference between the BDT and index methods on one side, and the wavelet ridge extraction on the other side:

- BDT and index methods classify each position in the reach into a given category (pool, riffle, run, plane bed, etc.); hence, in 1D, we have access to a discrete values of bedform lengths $L_i$ (i=1...N), and we can compute statistics of this discrete distribution such as **mean**, **mode**, **n-th order moments**, etc.;
- In contrary, the wavelet ridge extraction provides a continuous description of bedform spacing along the reach, through a continuous wavenumber function K(x). In turn, we can compute the statistics (again, mean, mode, n-th order moments, etc.) of this function in order to compare them with the values obtained in a discrete method.

Moreover, index methods use expert judgement in order to specify threshold values for each variable/degree of freedom. Since wavelet analysis is continuous in nature, such thresholds are not needed in our method.

*Third, the computation of the index method relies on the results of the numerical model. Have the authors considered the impact of the uncertainties in the results of the numerical model on the results of the index method?*

Response:

The use of model outputs is indeed a questionable choice that may add a lot of uncertainty in the results. The purpose of the numerical model used in the previous studies by Navratil et al. (2006) was simply to generate water surface profiles for discharge values other than the surveyed ones (i.e., interpolate/extrapolate the rating curves). In our revised paper, we will solely rely on measurements at the lowest surveyed discharge. However, since the calibration of the FLUVIA model on the reaches provides estimates of Manning coefficient n, we will use these values in order to compute the third degree of freedom, bed shear stress $\tau_b(x)$, along the reach: even if partly relies on calibration, it seems a more robust way of computing $\tau_b$ than through the finite differentiation of the total head function $U^2/2g + z_{surface}$ between adjacent cross-sections to get the energy slope J.

*Furthermore, I suggest discussing the transferability of the new methods to other reaches. In other words, how easy would be to implement the proposed methodologies to other study areas? Are the data and algorithms required easy to collect and implement? Can other researchers implement the proposed methodologies?*

Response:

As stated previously, the wavelet methods intends to be quite general and can be applied in any morphology that presents alternating bedforms (pool-riffle, step-pool, etc). The code comes in the form of a small number of Matlab functions, and the data has to be provided as values of flow variables sampled at successive locations along the reach. The choice of the set of variables/degrees of freedom is up to the user, in our case we chose the set $[U(x), Rh(x), \tau_b(x), \theta(x)]$ but we could pick other variables.

*Moreover, I think the manuscript should clearly state which methods are recommended. A more explicit presentation of the conclusions of this study would highlight its scientific and practical relevance.*

*Regarding the structure of the paper, I would like to recommend two modifications: - Section 2 lists a large number of studies and it is a bit hard to follow. More specifically, I think it is difficult to appreciate the differences between the large number of criteria listed in this section. The authors might consider adding a table to summarise their literature review.*

*I hope the authors will find my questions and recommendations useful to improve their manuscript.*

Response:

As said before, the structure of the paper will be changed by splitting the results and discussion and adding a conclusions part, the later one will specify the added value of the wavelet method according to the comparison in the discussion section. In fact, we will compare the metrics computing (mean, mode, n-th order moments, etc of the distribution) using the two methods.

**Minor comments:**

I listed below a number of minor recommendations.

| Comment of the reviewer | Response of the authors |
|---|---|
| Page 1, lines 7-8: the sentence "To better take this high-frequency variability in bedforms into account in hydraulic models" is a bit convolute. The authors might consider improving the structure of this sentence. | We will replace it with: "To include/consider this high-frequency variability of the geometry in the hydraulic models" |
| Page 1, abstract: the abstract should clearly state the research gap, the aim, and the novelty of the study. | We will change the abstract by including that this work is a methodological study that introduces relatively new wavelet analysis tools in the field of geomorphic analysis (namely, Wavelet Ridge Extraction), in order to identify the pseudo-periodicity of alternating morphological units from a general point of view (and not only pool-riffle morphology). We will state clearly the aim of this paper which is for example extracting some quantitative properties of these alternating bedforms such as the mean and the mode of their longitudinal spacings, with a "continuous" vision of the topography instead of a discrete classification. |
| Page 1, line 9: the abstract mentions "several methods", however, only three (two novel methods and one benchmarking method) are listed explicitly. | As stated above, we will clarify the presentation: we introduce one new method (wavelet ridge extraction) in univariate and multivariate cases and we compare the results with an existing method (BDT). |
| Page 1, lines 12-13: the authors might consider avoiding the repetition of the word "compared". | Corrected |
| Page 2, lines 14-15: I am not sure whether this is the final format of the paper, however, I suggest positioning each figure after a full stop (Figure 1 is currently positioned in the middle of a sentence). | This is not the final format of the paper. We will change that in the revised version. |
| Page 2, line 15: please correct "dimensionless reach wavelength". | Corrected |
| Page 3, lines 4-5: this sentence is a bit hard to understand. Do the authors mean that the overarching purpose of their study is to provide a methodology for the prediction / modelling / assessment of cross sections variability? | We will cut out this sentence and change from line 14 to 16 in page 2 with: "In this study, we focus mainly on alternating alluvial channels especially pool-riffle sequences, even though the method presented here could be used to analyze any alternate morphological units (MUs). The objective is to provide a continuous description of geometric and flow patterns along a reach, a description that could be subsequently used to create a synthetic river as in the RiverBuilder (Brown et al., 2014). To do that we calculate the dimensionless reach |

| | |
|---|---|
| | wavelength $\lambda*$, which is the distance…". |
| Page 3, line 8: words such as "methods" or "techniques" might be more appropriate than "studies". | Corrected |
| Page 3, line 11: could please the authors clarify the meaning of "descriptions of the water surface characteristics"? Is "water surface slope" (mentioned in Line 8) included in this latter category? | Descriptions of the water surface characteristics means a method that describe pools and riffles from the combination of all characteristics of the water surface (water surface elevation, water surface slope, etc.) and which include effectively the slope as Mosely (1982) mentioned in his paper. We also corrected the reference. *"Mosley, M. P. (1982). Procedure for characterising river channels, Water Soil Misc."* Instead of *"Mosley, M. P.: Analysis of the effect of changing discharge on channel morphology and instream uses in a braided river, Ohau River, New Zealand, Water resources research, 18, 800–812, 1982."* |
| Page 3, line 14: I suggest clarifying the sentence "because it changes less with discharge". | It means that this morphological definition of pool-riffle sequences doesn't depend on discharge. |
| Page 3, line 20: please rephrase "goes with the notion". | We will change it with "involves the use" |
| Page 3, line 22: please rephrase "allows one to extract". | We will change it with "extract" |
| Page 3, line 30: please rephrase "using a threshold on a criterion index." | We decided to cut out this method |
| Page 4, Figure 2(A): I believe that this figure is not mentioned in the text. | Corrected, we will mention it in the page 2 |
| Page 4, lines 3-4: I think this sentence should be moved to the section 6.2 as it motivates the choice of the benchmarking method. | We will move this sentence to the comparison methods section and modify it according the new structure of paper. |
| Page 5, line 7: "the areal difference asymmetry index by Knighton" has not been mentioned before, the authors might consider adding more context to this statement. | It was felt that there is no need to define this method because it's just an example of methods existing in the literature. However, we will add it in the table that summarizes all the previous methods and techniques. |
| Page 5, line 32: the manuscript states: "a common geomorphological and hydrological" methods, I suggest specifying these methods. | We will change the entire sentence according to the new structure of the paper |
| Page 6, line 8: was the channel width/channel bankfull width used to compute dimensionless values of wavelength? I think this sentence is not clear. | Yes, it's not clear. Here we are talking about the dimensionless pool spacing, in which there are researchers who use the definition $\lambda^* = \frac{\lambda}{W}$ (mean channel width) while others use $\lambda^* = \frac{\lambda}{W_{bf}}$ (mean bankfull channel width). We will change it in the revised version. |
| Page 6, line 9: what do the authors mean with | "certainty of these ratios" means their efficiencies |

| | |
|---|---|
| *"certainty"?* | to give more consistent results, so we will change it to "efficiency" |
| *Page 6, line 14: I suggest avoiding colloquial expressions such as "a great deal".* | We changed it to: "Some researches have investigated" |
| *Page 6, line 32: I suggest rephrasing this sentence and avoid the use of "we".* | We will change all the sentence to: "These reaches contains mainly pool-riffle morphologies, they have slopes …" |
| *Page 7, line 5: I believe that information on slope has been previously provided in page 6, line 32. Could please the authors explain the added value of this sentence?* | This line has been added to define the thalweg elevation and how it can be estimated. That's why we will delete it and add this information in the lines 1 and 2 p7: "they have slopes between 0.002 and 0.013 m.m$^{-1}$ (estimated from the talweg elevation which is the lowest point in the section)…" |
| *Page 8, line 11: please clarify the sentence "It is based on interpolations rather than extrapolations".* | As said before, the role of the 1D hydraulic model (Fluvia) was simply to generate water surface profiles for discharge values other than the surveyed ones (i.e., interpolate the rating curves between values of surveyed discharges, and extrapolate slightly above highest surveyed discharge). In our revised paper, we will solely rely on measurements at the lowest surveyed discharge and use the model to provide estimates of Manning coefficient n, we will use these values in order to compute the third degree of freedom, bed shear stress $\tau_b(x)$, along the reach. So this part form line 10 to line 14 will be modified by an explicit description of the model and the data set. |
| *Page 8, line 13: "visually": do the authors mean that they performed a manual calibration of the hydraulic model?* | It is a typo, we checked the calibration visually, but we adjusted it with a minimization function. |
| *Page 8, line 14: please remove the second full stop.* | Corrected. |
| *Page 8, line 14: "multi-section flows": do the authors mean that the numerical model is used to predict a number of quantities (e.g. the elevation of the water surface, wetted perimeter, wetted surface,: : :) at a number of cross sections?* | The use of the numerical model (Fluvia) will be simply to generate calibrated estimates of Manning coefficient n that we will use to compute the bed shear stress $\tau_b(x)$ along the reach. For the other cross-section variables, we will use only measurements at the lowest surveyed discharge. |
| *Page 8, line 3: why is the minimum discharge used for the implementation of the method?* | We chose the minimum discharge (low flow) in the development of the method because it is the discharge through which we can visualize the variability of the bathymetry (alternating morphological units). |
| *Page 8, line 6: does "it" stand for "relevant information"? The authors might consider editing the structure of this sentence.* | This section will be removed from this paper as suggested by the second reviewer Prof. Gregory Pasternack. |
| *Page 8, line 8: I believe that "the trend" has not* | The only detrended variable was bed elevation: we |

| | |
|---|---|
| *been explained before. I suggest clarifying this sentence. What does "detrended variables" mean? How are these variables computed?* | computed a series of bed elevation anomalies $\varepsilon_z(x)$ such that: $z_{bed}(x) = -S_{bed}\,x + b + \varepsilon_z(x)$, where $S_{bed}$ is the mean slope of the reach and $\varepsilon_z(x)$ has zero mean. This part is not necessary anymore. |
| *Page 8, line 13: "contain the most explained variances" do the authors mean that those directions can explain the variability of the data? I suggest clarifying this sentence.* | The PCA analysis will be completely removed so this discussion is not relevant anymore. |
| *Page 8, lines 18-20: does these results confirm/contradict previous studies?* | The PCA analysis will be completely removed so this discussion is not relevant anymore. |
| *Page 9, figure 4: could please the authors explain the meaning of Dimension 1: : :9?* | The PCA analysis will be completely removed so this discussion is not relevant anymore. |
| *Page 9, lines 5-6: I think this sentence is unclear. What is the relationship between bed elevation and hydraulic radius? The statement seems to be contradictory. Moreover, I was wondering whether any correlation between bed elevation and hydraulic radius is meaningful. Bed elevation is a geometric characteristic at the point scale. The hydraulic radius depends on discharge, river bed slope, cross section area.* | Here we are not talking about the physical meaning of these variables but their variability. The hydraulic radius is the cross-sectional area divided by the wetted perimeter, so the hydraulic radius, the cross-sectional area, and the depth are positively correlated, while the water surface elevation is the depth plus the bed elevation, so the depth and the bed elevation are negatively correlated. So the bed elevation and the hydraulic radius are negatively correlated. It's just trivial findings. For that we will choose in the revised paper variables that are not dependent. |
| *Page 9, lines 6-8: the explanation based on hydraulic radius and Froude number is reasonable and (almost) intuitive. I suggest to clarify the added value of this finding compared to the existing literature.* | There is no added value of this finding, we were wrong about the justification of our choice of variables. We will change all this section as we mentioned it before. |
| *Page 9, line 9: I suggest clarifying the importance of bed elevation.* | Historically, bed elevation has been seen as the most relevant variable in order to characterize geometric and flow variability. Since water surface elevation cannot change in space as fast as bed elevation, local bed elevation (and slope) is an important driver of depth and velocity variations along the reach. However, width variations have been found to be important as well, so a multivariate approach must clearly be favored. |
| *Page 9, line 10: what do the authors mean with "we smooth" the data?* | The formulation was wrong; in fact the processing mentioned in this sentence was only applied to bed elevation: since the trend of bed elevation is not necessarily linear, we performed a more general removal of very low frequency components (wavelength larger than 7 times the mean bankfull width) before applying thresholds. Since we will not use bed elevation anymore, this processing is no longer relevant (and it was not a smoothing anyway). |

| | |
|---|---|
| *Page 12, lines 11-13: I suggest improving the readability of this sentence.* | We will change all the structure of this paragraph |
| *Page 12, lines 16-18: please improve the structure of this sentence: "have been interested: : :but working", both the verbs have the same subject.* | We delete this sentence in the revised paper. |
| *Page 12, line 18: "analysis" is repeated.* | Corrected. |
| *Page 14, line 12: I suggest replacing "evacuate" with something more appropriate (an option could be "remove").* | Corrected. |
| *Page 15, line 3: please clarify "It also represents" (what does "it" stand for?)* | "it" stands for "the curve that continuously crosses the domain" and also "K(x)".
 We will replace it by "This curve K(x) also represents …" |
| *Page 15, line 11: could please the authors better explain why this correction is applied?* | Equation (21) actually gives the amplitude of the pseudo-periodic signal through inverse wavelet transform. In this reverse transformation we need to multiply by $\sqrt{s} = \sqrt{\frac{1}{\alpha K(x)}}$ where $\alpha$ is the Fourier factor (Torrence and Compo, 1998), since we multiply by $\sqrt{\frac{1}{s}} = \sqrt{\alpha K(x)}$ in the direct transformation (Equation 14). |
| *Page 15, line 15: please correct the structure of this sentence: "we limit the study only with univariate analysis". Moreover, could please the authors justify this choice?* | As we said before, we will focus in the revised version of this paper on both the univariate and the multivariate analysis and we will compare their results with the BDT (O'Neill and Abrahams, 1984). |
| *Page 15, line 26: please clarify the meaning of "multivariate case".* | The multivariate case is the extension of the univariate to a set of N real-valued signals; it is described in Lilly and Olhede (2009). We will describe this case and develop its transformations in our revised version. |
| *Page 17, Table 3: Table 3 and Table 2 show the results of the two methods for the same river reach. The authors might consider displaying these tables in the same page in order to allow a straightforward comparison of the results.* | In the revised version we will cut out these kind of results. |
| *Page 18, lines 2-4: I think that this sentence is unclear.* | We will replace it with a clear sentence according to the new results that we will have. |
| *Page 19, line 3: I believe that these results demonstrate a good level of agreement between the two methods. In my opinion, these results do not provide explicit information on the accuracy of the methods.* | Yes, these results do not prove the accuracy of the methods. For that, we chose presenting one method (wavelet method) and discuss it with one benchmark methods as explained before. |
| *Page 19, line 5: the BDT methods is used to "validate" the results of the proposed methodologies. This choice implies that the BDT method is more accurate than the new methods* | In this study, we consider the BDT method as a benchmark method. We do not consider a specific method to be the "true" or "reference" one, we only apply several methods to have a general idea |

| | |
|---|---|
| *introduced in this manuscript. Is this correct? If so, what are the benefits/ advantages of using the two proposed techniques?* | on the uncertainties in the identification of morphological units. |
| *Page 19, lines 19-20: please clarify this sentence.* | We will delete this sentence in the revised version. |
| *Page 19, line 23: the manuscript states that the results of BDT "are closer to the other methods and to reality". I strongly recommend to better substantiating this sentence. Which are the "other methods"? What does "reality" mean? Was the BDT method compared with field data? In which case study?* | True, This sentence is not clear, we will delete it. |
| *Page 20, lines 3-4: could please the authors clarify this sentence?* | We will change all the discussion according to the results that we will have. |
| *Page 20, lines 6-7: please rephrase this sentence.* | This sentence is not clear, we will change it in the revised version. |
| *Page 21, line 2: a Froude number of 0.30 looks a bit large. Could please the authors explain this result?* | We will dismiss the Froude number In the revised version. But for example in the study of Jowett (1993) and Clifford et al. (2006), they found values close to 0.3, so we think that these values are a bit large but acceptable.  Fig. 1 Kernel density functions of velocity/depth ratio, Froude number, and water surface slope in pool, run, and riffle habitats. |
| *Page 21, line 3: it seems that the average values are driven by the results of the Graulade river, Are the average values representative of the sample?* | If we exclude the Graulade river we will find and average of 0.20 for the index and 0.17 for the wavelet method. These results are nearly close the 0.23 and 0.20. |
| *Page 21, line 10-17: these lines present a comparison between the results of this study and some of the previous studies. The authors might (or might not) consider using a table to summarise these comparisons.* | This is a good idea. |
| *Page 21, line 21: I suggest motivating this sentence. Why aren't the previous methods considered quantitative?* | We will delete all this sentence |
| *Page 22, line 1: is "crossing" the most appropriate word?* | We will delete all these sentences |
| *Page 22, line 3: please clarify this sentence* | We will change this sentence in the revised version |

**Reply to Prof. Gregory Pasternack**

Dear Prof. Gregory Pasternack,

We thank you for your comments, which will help improve the clarity of the manuscript as well as the choice of the methods.

According to both reviews we decided to make very substantial changes to the paper. This work is a methodological study that introduces relatively new wavelet analysis tools in the field of geomorphic analysis (namely, Wavelet Ridge Extraction), in order to identify the pseudo-periodicity of alternating morphological units from a general point of view (and not only pool-riffle morphology). We did initially introduce an index method as a benchmark, but this index was poorly designed due to a lack of physical basis for the choice of the variables. We also neglected some relevant literature on the identification of the morphological units using DEMs, which could be used as benchmark methods in this paper.

For that, we suggest changing the title of this paper to "***Automatic identification of alternating morphological units in river channel using wavelet analysis and ridge extraction***". This will be more general and focuses on the method and not on the pool-riffle morphology.

We have presented two methods in this article. The first one is the wavelet method which represents alternating morphological units (pools and riffles) as pseudo-periodic signals with a continuous wavenumber function $K(x)$. The other one is the index method which is a benchmark method that gives a discrete identification of the morphological units.

According to your suggestions, we will cut out the index method. We will focus only on the wavelet analysis and ridge extraction in the univariate and the multivariate cases and compare its results with the benchmark method: BDT (O'Neill and Abrahams, 1984) to the bed elevation data. We didn't compare it with other recent methods (e.g. Hauer et al., 2009; Wyrick et al., 2014) because they require thresholds (expert judgment) collected from the field, which is not possible in our case.

We will also minimize the use of modeled variables and apply the methods directly on field measurements (velocity, hydraulic radius variables at the lowest surveyed water level and planform curvature angle). We will use modeling results (Fluvia model) for bed shear stress only, as the energy slope cannot be determined in a sufficiently accurate manner with the measurements.

For the literature, we missed many recent studies and methods in relation to this work. So first we will add a table that summarizes examples of methods of identifying these morphologies and the variables chosen to do that. Second, we will change and add many recent works especially those working with meter-resolution digital elevation models (DEMs). Finally, we will clearly state the objectives of this study in the abstract and in the introduction.

Another important thing is that we propose a new structure of the paper:

**I-    Introduction**:

First, we will state the scope of this study with adding more fields of its application. Second, we will introduce a literature review of metrics, variables used to identify and characterize the alternating morphological units. We will focus on two kinds of numerical criteria computed at reach scale:

- The distribution of spacings between morphological units (mean, mode, etc.),
- After computing the mean values of geometrical and flow properties (velocity, hydraulic radius, bed shear stress, etc.) in each class of morphological units (e.g. pools, riffles, runs, etc.) we will evaluate the covariance matrix of these parameters.

**1) State of art methods for a quantitative assessment of morphological variability within a reach**:

We will present some recent methods and works in the identification of these alternating morphological units (pool-riffle in our case) and state their objectives and limitations. We will start with the Bedform Differencing Technique (BDT, O'Neill and Abrahams, 1984), which is simple but uses bed elevation as the sole variable, and relies on a tolerance criterion on elevation differences. We will then review index methods like Mesohabitat Evaluation Model (MEM, Hauer et al., 2009) which classify each position in the reach into a given discrete morphological unit (pool, riffle, run, plane bed, etc.). These methods rely on expert judgement to define the thresholds that define parameter classes. Finally, geostatistical methods (e.g. Legleiter, 2014) give a continuous description of river channel properties in spatially stationary way, using longitudinal and transverse variography. For these reasons, we are searching for a method that gives a continuous description of geometrical and flow characteristics along the reach with a non-stationary description.

**2) Study objectives**

We will state that this work aims to introduce relatively new wavelet analysis tools in the field of geomorphic analysis, the Wavelet Ridge Extraction, in order to identify the pseudo-periodicity of alternating bedforms from a general point of view. In this study we will use a dataset that presents mainly pool-riffle morphologies, but the method can be applied to any morphology.

We will present the scheme of the paper which include a methodological section of the wavelet analysis and ridge extraction in the univariate and the multivariate cases, a section that presents the comparison between our method and the BDT, a discussion section, and conclusions.

**II-    Data set and study reaches**:

We will present the six reaches, more explicit information about data collection, planform curvature angle computing and about the numerical modeling (Fluvia).

**III- Wavelet method**

**1) Wavelet analysis and ridge extraction:**

We will present a general introduction about wavelets including some methods such as the Wavelet Transform Modulus Maximum (WTMM, Gangodagamage et al., 2007) and other studies using the wavelets in the geomorphological field (Lashermes et al., 2007; McKean et al., 2009). Procedures such as the WTMM (Muzy et al., 1993) consist in extracting components of the signal, but they are not specifically designed to identify pseudo-periodic components in a univariate, let alone in a multivariate case. For this reason, we introduce the procedure called Wavelet Ridge Extraction (Lilly and Olhede, 2009).

**2) Univariate case**

We will present the methodology of this method in the univariate case using one of the four variables (velocity, hydraulic radius, bed shear stress, and planform curvature angle).

**3) Multivariate case**

We will present the methodology of this method in the multivariate case using the four variables (velocity, hydraulic radius, bed shear stress, and planform curvature angle).

**IV- Results**

**1) Univariate vs Multivariate:**

We will compare the univariate with the multivariate results with computing some statistics. And we will use the multivariate wavelength to model the bed elevation of the reaches without using it as a variable.

**2) Comparison with the benchmark method**

We will compare our method's results in the multivariate case with the selected benchmark method from the literature: BDT.

**V- Discussion**

We will discuss results (longitudinal spacing, number of morphological units, etc.) with literature and with the benchmark method.

**VI- Conclusions**

Kind regards,

The authors

*It is unfortunate that the manuscript does not have continuous line numbering to aid reviewers and editors with referring to locations easily, even the repeating page numbers are only every 5th value, which is not convenient. Actually, based on page 8 where there is new numbering at the onset of section 4, I am totally confused as to how line numbering is done and it makes it harder to review the paper in a discussion format that requires me to write out all my comments rather than simply mark up a manuscript. In future manuscripts, always include full and continuous line numbering.*

Response:

We used the Latex Template; maybe there is a problem in the numbering characteristics that we will modify in the revised version.

*First 2 paragraphs of the introduction. It seems odd to me that the main reason why anyone should be interested in understanding the sub-reach variability of river topography is because of the potential application of such information to flood forecast modeling. Even in the applied realm that is only 1 of many applications that could be referred to. In my own research, the primary motivations are that such data is required for river design for a wide variety of purposes including river rehabilitation and enhancement and also because it informs fluvial ecohydraulics. In light of systemic global ecological collapse, these are more important to society than flood forecasting, in my professional opinion. At a minimum, I think the authors should identify a few more reasons why knowing topographic variability matters and add a citation for each. Also, of course, geomorphologists want to understand it in its own right as a basic scientific question that requires no justification, and of course it is also the case that this variability controls fluvial processes, so the lack of knowledge about it means that we really know little about processes; less than I think most people realize.*

Response:

It's true that there are more reasons why knowing topographic variability matters like you mentioned above. For that we will modify the first paragraph of the introduction by adding examples of application of our study like the design of a synthetic river topography which is implemented in river restoration (e.g., Wheaton et al., 2004a), habitat modeling, ecohydraulics (e.g., Pasternack and Brown, 2013), and of environmental modelling (oxygen exchange, fish habitat) and also that this variability controls fluvial processes as sediment transport, but not focusing only on flood forecasting.

*P. 2, lines 3-8. While this is generally true, the authors seem to be unaware that my lab group has already published theory and code that is the first to procedurally generate river terrains exactly to specification from the equations and parameters, and this methodology does include sub-reach-scale variability that can go to as high of a frequency as one wants to make it, so quite small scale. There is always more to do, but I think this is relevant to the claim of this paragraph. I see that this paragraph has 4 citations for the first sentence alone, which seems like too many, so removing 1-2 of those could make way for citing this relevant work if the authors agree that what we published does in fact do what they say is an important thing to do, even if not perfectly, but still more than anyone else thus far. The journal citation is Brown, R.*

*A., Pasternack, G. B., Wallender, W. W. 2014. Synthetic river valleys: creating prescribed topography for form-process inquiry and river rehabilitation design. Geomorphology 214: 40-55. 10.1016/j.geomorph.2014.02.025. The code is open-source and free to the world presently coded in R as "River Builder". The R package and user's manual can be downloaded from the CRAN website at https://cran.rproject.org/package=RiverBuilder. The code also includes the Perlin function that can create very small scale features, and that is a common method for generating landscape terrains in the video game and animation industries. In the future we hope to add the capability to parametrize the sub-reach-scale fluctuations in spatial series of detrended bed elevation and lateral topographic breaklines using wavelet parameterization.*

Response:

This is true; we shouldn't ignore these studies because they are relevant and important to this literature. For that we will change the paragraph in the page 2 from the line 3 to 8 by: "Many researchers are working on determining the best simplified representation of channel geometry (Saleh et al., 2013; Grimaldi et al., 2018), based on the variability of cross sections but without the knowledge of the bed elevation variability or the river sinuosity. While other studies focused on the generating of river channels with taking into account the sub-reach scale variability using geostatistics and variogram tools (Legleiter, 2014a, 2014b) or a geometric framework modeling with geomorphic covariance structures (Brown et al., 2014). Longitudinal variability in river geometry has greater impact on the simulation of the water level than the cross-sectional shapes (Saleh et al., 2013) and it must be taken into account in the hydraulic models."

*The third paragraph of the introduction serves no required purpose and neither does Figure 1. Both can be deleted with no loss of understanding. Yes, rivers come in different types, but the main thing readers need to know is that this is a study of riffle-pool reaches and that the method can apply to other reaches; these ideas can be promoted without any of this paragraph, as is indicated by the first sentence of the very next paragraph just fine.*

Response:

It's true; we should focus on the alternating morphological units especially the riffle-pool sequences without including this paragraph and the figure 1. We will add to the line 14: "This topographic variability is related to the channel morphology types. In this study, we focus mainly on alternating alluvial channels especially pool-riffle sequences, even though the method presented here could be used to analyze any alternate morphological units (MUs).", and we will remove lines from 9 to 13 and figure 1.

*p.2, lines 15-16. The objective of what? The writing is unclear here. I disagree that the main purpose of quantitative analysis of channel topography is just to get pool spacing. In support of our River Builder software, one normally wants to analyze many aspects of reach-scale topographic variability so that they can all be parameterized and used to make realistic synthetic rivers. Other important variables would be parameterizations of thalweg planform curvature, base flow and bankfull channel width undulations, floodplain width undulations, and then how all of these are phased relative to each other (in time series*

*that's "coherence" and "cross-phase"). Thus, pool spacing is certainly one useful data output, but not alone or necessarily most important.*

Response:

This is true, the paper isn't about the pool spacing identification, but its purpose is for example extracting some quantitative properties of these alternating morphological units such as the mean and the mode of their longitudinal spacings, with a "continuous" vision of the topography instead of a discrete classification. This will be done by focusing on two kinds of numerical criteria computed at reach scale: The distribution of spacings between morphological units (mean, mode, etc.), and after computing the mean values of geometrical and flow properties (velocity, hydraulic radius, bed shear stress, planform curvature angle, etc.) in each class of morphological units (e.g. pools, riffles, runs, etc.) the evaluation of the covariance matrix of these parameters.

*I also note that tat the authors never use their reach site results to present any conclusions about the science of pool spacing, so if it is so important than its value should be evident in how the results are used to advance science.*

Response:

We should be clear in that point that our paper is methodological research that propose a new method with new developments. Of course we will add some conclusions about the longitudinal spacing results and the covariance between the chosen variables.

*p. 3, lines 1-4. No need to define wbf twice. Remove one of them.*

Response:

We will remove the second one and change the first at line 3 by: "the reach average bankfull width (wbf)".

*p. 3, lines 7-17. A major problem with the historic work cited here that its all pre-2001 and how it is presented is that the authors are not addressing the equal importance of channel width undulation to channel depth undulation. Richards in the 1970s understood it and wrote about the importance of width. However, because people didn't tend to make width profiles down rivers, the focus wrongly got limited to depth undulation in the literature of the late 20th century. Of course, authors studying velocity reversal concepts did start to understand this problem pretty well by 1990. With modern high resolution DEMs since 2000, that problem is over and now we are in the era of looking at how depth and width co-vary to control pool and riffle topography and morphodynamics vis-a-vis the "flow convergence routing" mechanism explained by MacWilliams et al (WRR, 2006) and explored further by Prof. Jose Rodriguez in recent WRR papers as well by my lab group in several articles (Sawyer et al., Geomorph., 2010; Brown et al., Env. Man., 2015; Strom et al, Hyd. Proc., 2016; etc). My lab group has published a series of papers on the importance of linked depth and width undulations that has culminated in a new sub-reach scale channel unit classification relevant to this paragraph and this study. See these two articles, the rest leading up to these are cited in them: -Pasternack, G. B., Baig, D., Webber, M., Brown, R. 2018.*

*Hierarchically nested river landform sequences. Part 1: Theory. Earth Surface Processes and Landforms. DOI: 10.1002/esp.4411. -Pasternack, G. B., Baig, D., Webber, M., Brown, R. 2018. Hierarchically nested river landform sequences. Part 2: Bankfull channel morphodynamics governed by valley nesting structure. Earth Surface Processes and Landforms. DOI: 10.1002/esp.4410.*

Response:

You're right, we didn't present our literature in that chronological way, which is interesting for the reader, we should focus not only on the identification methods but also on the science of alternating bedforms including pool-riffle, but honestly a methodological paper focusing on alternating bedforms should mention these works and shouldn't neglect any of them. As you suggested, we will change this entire paragraph and summarize it in a table and add studies done after 2000. In addition to that we will present the literature of depth and width undulations in relation to pool-riffle, the same thing for the modern high resolution DEMs, and the thresholds chosen in literature which would help us to discuss our results.

*p. 3, lines 20-26. Yes, I agree with all of this, though I don't think wavelet analysis cannot be called "new" as it has been published in geo/hydro journals for decades now; what's new is high quality topo data to apply it to, though that is present intros study. I'm surprised by the citations the authors offer here, as they are not very relevant compared to other options, such as (most importantly) Gangodagamage et al. (Geomorph., 2007) but also Lashermes et al. (WRR, 2007) and McKean et al. (Rem. Sens., 2009). One can use spatially evolutive Fourier analysis and autocorrelation analysis or, if one limits the analysis to a single reach, regular Fourier analysis where the average parameterizations are reasonable.*

Response:

What is new is the method presented itself and the identification of alternated morphologies, since it is never made with wavelets. But for the wavelets, yes it has been present for decades, however it is still less used compared to Fourier. Wavelet Transform Modulus Maximum (WTMM, Muzy et al., 1993; Gangodagamage et al., 2007) and other studies using the wavelets in the geomorphological field (Lashermes et al., 2007; McKean et al., 2009) consist in extracting components of the signal, but they are not specifically designed to identify pseudo-periodic components in a univariate, let alone in a multivariate case. For this reason, we introduce the procedure called Wavelet Ridge Extraction (Lilly and Olhede, 2009). These works will be presented in the revised version.

*One might even argue that the locations where the Wavelet analysis indicates a change in parameters could be a reach break. Certainly wavelet analysis is a very good way to go for this to objectively delineate reach breaks, but preferably with a multivariate strategy using both depth and width variables. A good comparison would be to look at the riffle-pool quasi periodicity analyses of Brown, R. A., Pasternack, G. B. 2017. Bed and width oscillations form coherent patterns in a partially confined, regulated gravel–cobble-bedded river adjusting to anthropogenic disturbances, Earth Surface Dynamics, 5, 1-20, doi:10.5194/esurf-5-1-2017.*

Response:

That's what we will do in the revised version, we will focus only on the wavelet analysis and ridge extraction in the univariate case using one of the four variables (velocity, hydraulic radius, bed shear stress, and planform curvature angle) and the multivariate case by using the all four of them and compare its results with the benchmark method: BDT.

*P.15, line 14. This sentence makes a key determination that flies against the same kind of decision-making applied to the index method of section 4. Specifically, the determination of riffles and pools is going to rely entirely on bed elevation. It seems odd that scientists who begin with the conjecture that multiple variables should be used to determine riffles and pools would now contradict themselves and only consider one variable. My view of it is that both decisions are arbitrary, as (a) the former was based on a questionably PCA analysis lacking a mechanistic basis and choosing interdependent variables rather than the proper independent ones and (b) the latter is likely based on the amount of work it takes to apply the wavelet methodology and so its application is being limited to only one variables and to only 1 reach instead of all the reaches. I am making my own guess with (b), but the authors provide no justification for limiting the analysis to only 1 reach after introducing so many reaches. Similarly why not do all three variables the authors deem important. A quick check of the scientific literature confirms that multivariate wavelet analysis exists and is available for use. And then there is the issue of how the variables couple to affect riffle and pool occurrence, structure, and resultant processes. The decision-making here is too opaque and needs explanation per these issues. I expect the decisions cannot be justified, but the authors deserve a chance to try.*

Response:

According to these comments:

(a) We totally agree with you that the choice of these 3 variables is unsound, so we will keep the multivariate case with a physical combination of variables. In the revised version, we will focus on the wavelet analysis using the univariate and the multivariate with four variables; we choose the classic three ones: velocity, hydraulic radius, and shear stress, in addition to the planform curvature angle that represent the planform variability.

(b) For the multivariate case, it isn't a problem of computation time, we already have an implementation of the wavelet ridge extraction in a multivariate case; however, we initially chose not to present it in the paper because we need to introduce a specific criterion to identify the local wavenumber K(x). Basically, in the univariate case, wavelet ridge points are those points of the (x,K) plane where the phase of the wavelet $\phi(x,K)$ changes in space exactly at rate K (i.e., $(\partial\phi/\partial x) - K = 0$ : the signal is locally similar to a sinusoid of wavenumber K in rad/s). In the multivariate case, we search not for an equality but for a local ***minimization of the norm of the vector*** [ $(\partial\phi 1/\partial x) - K$ ; $(\partial\phi 2/\partial x) - K$ ; $(\partial\phi 3/\partial x) - K$] with respect to K : the local wavenumber K(x) is such that ***all four variables locally look like sinusoids of same wavenumber K(x), but with potentially different phase shifts***. Clearly, this co-evolution is needed to identify morphological units. We will add the necessary mathematical developments in the appendix of the revised paper.

*p.5, lines 19-20. This explanation is incorrect on two levels. First, energy gradient is more than just water surface slope, because energy also accounts for the velocity head that is not in that term. Often velocity isn't changing over long distances or is assumed to not change, but along a riffle crest and in the transition to a pool it definitely changes quite a bit, so strictly speaking one has to account for that. Second, the energy gradient is stage dependent, because the steepest gradient is always associated with the vicinity of the smallest cross-sectional area, all other things being equal.*

Response:

The complete quotation of the paper by Yang (1971, p. 1567) reads: "For practical purposes, the energy gradient for most natural streams may be replaced by water surface slope without much error". So we are of course fully aware of the difference mentioned in the referee's comment, even though this difference between energy slope and water surface slope is usually small for "low Froude" rivers such as the ones studied here. But to avoid this misunderstanding we will refer only to the energy gradient.

*At low discharge the way the authors describe it is true, because at low discharge riffles have the smallest XS area. However, once the discharge exceeds the value for the minimum cross-sectional area of the reach to be elsewhere, then it is not at the riffle any more. At some high flow it will become at the pool location, and of course this is the main reason why pools scour and riffles aggrade to maintain relief in alluvial channels, all other things being equal (especially substrate). This stage dependence is a key issue to account for in any scheme to evaluate where riffles and pools are located and it its why considering only depth and ignoring width has always been a mistake by the river science community. Now that we have width data commensurate to depth data, we can move on to the proper treatment of the problem considering their linked co-variance.*

*p.9, lines 7-8. These claims apply only to low discharges, due to the flow-dependent nature of riffle-pool hydraulics. How they develop as discharge increases depends on the shape of the cross-sections (especially depth vs width "geomorphic covariance", per flow convergence routing theory.*

Response:

Absolutely true, that is why we chose low flow instead of high discharges. We completely agree that the maximum shear stress may be located in different morphological units at high discharges than at low discharges, and that it is very important to understand how relief is maintained. Exploring this near-bankfull behaviour is the reason why hydraulic modeling was needed in the first place in the study by Navratil et al. (2006), since it is difficult to obtain field measurements precisely at bankfull conditions. The wavelet ridge extraction could perfectly be applied at bankfull conditions, but since it would rely on modeling results if we want to perform it on our dataset, we will leave it out of the scope of the paper.

*p. 5, lines 22-37. All of these methods retain the limiting viewpoint that they put a primacy on riffles and pools, either ignoring other morphological units (MUs) or treating them as irrelevant. Thankfully, 2D and 3D hydrodynamic modeling ends that mistake and enables objective mapping of all MUs with decision-tree analysis. This approach was explained by Wyrick et al. (Geomorph, 2014) and then applied in Wyrick and Pasternack (Geomorph., 2014) to not only show the greater diversity of MUs beyond riffles and*

*pools, but also to compute simple metrics like pool spacing. Thus, Wyrick and Pasternack (Geomorph., 2014) presented a novel methodology to extract pool spacing from 2D hydrodynamic model outputs of MUs using GIS tools. That is very relevant to this literature review, because it shows recent progress in automated extraction of this metric. The authors are arguing that their methods are more automated and better than pre-existing methods, but they have not actually considered more recent automated methods.*

Response:

This study of Wyrick et al. (2014) is relevant and presents a method that should be mentioned in the literature of this paper and also should be discussed in the discussion part.

*Meanwhile, the sentences about the outstanding work by Almeida and Rodriguez as well as Parker goes off topic from pool spacing to get into the separate topic of riffle-pool morphodynamics, of which there is a very long and illustrious literature not addressed. Best to cut those at this location and stay focused on the directly relevant literature about pool spacing that is the focus of this study. They may be relevant if the revised manuscript ever addresses processes explicitly.*

Response:

True, we should focus only on alternating morphological units and all references linked to morphodynamics or pool-riffle processes should be cut out from the paper.

*p.6., lines 5-27. Very good literature review and written well, just not accounting for many recent studies since 2001.*

Response:

We will add more recent studies.

*p. 7, line 6. The sentence about having surveyed "many" cross-sections is poorly constructed and, in my view, not accurate. Terms like "many" are relative, so it could be that for one person any arbitrarily small number of cross-sections would still seem like many; that makes it hard to argue the point. However, the key metric here is that one cannot analyze for topographically significant spatial frequencies at resolutions smaller than the minimum XS spacing, and that's already quite conservative. For that reason, my lab group uses vastly denser cross-sectional spacing than that used here. For example, in Pasternack et al. (ESPL, 2018b) we used a spacing of 3% of bankfull width. That's "many". For another group, Legleiter (Geomorph., 2014b) spaced a XS every quarter channel width. In contrast, in this study, an analysis of Table 1 finds that cross-sections are spaced between 0.46 to 2.9 times bankfull width, with two reaches not even having 1 XS every bankfull width. These numbers of cross-sections are more like the amount used in a conventional reach survey to obtain reach-average depth and width metrics, not to identify the underlying nature of variability. I think if the authors refer to previously cited articles above about spatial series analysis of rivers topography plus Legleiter, they'll get a better sense of what is needed to get at the detailed patterns of fluvial topo spatial series at the sub-reach scale. This issue*

*doesn't invalidate the study, but just recommends to back off the "many" and get to saying "a normal number of cross-sections typical for a 1D hydraulic modeling study" or something like that.*

Response:

We agree with you in that point, the use of "many" in this sentence is relative, so we will change it to "Cross-sections were surveyed".

*Also, these cited works could be referred to in the discussion section to help compare and contrast undulation metrics from different studies, including when undulations may not have high enough amplitude to become a "riffle" or "pool" but are still big enough to make a difference for the intermediate morphological units that are mentioned but not investigated in this study.*

Response:

We will cite these works in the literature review and we will discuss our work according to it to make some conclusions on the parts where we have pools and riffles without investigating the other MUs and also about longitudinal spacings.

*p. 7, line 6. I think a bigger questionmark for the technical soundness has to do with the mindful decision to not have all cross-sections regularly spaced, but to place them primarily at hydraulic controls and morphological breaks. The authors then interpolate to get a grid, but the source data is not uniform. I fully understand why that would be done for a 1D hydraulic modeling study and given perhaps limited resources and no lidar data, but there is no question whatsoever that biased (aka mindful) XS placement impairs and calls into question spatial series analysis as far as objective identification of parameters. By placing the XS where the authors think important hydraulic and morphological things are happening, then necessarily the wavelet analysis and any other method is also forced to bias results toward the same outcome of where significant things are happening. On the other hand, when I put an XS every 3% of bankfull width along the series, then there is no chance anything will be missed and the algorithm can decide for itself what the frequencies, amplitudes, and phases (and other parameters) are for that reach. Equal spacing of XS is the best approach for unbiased results. I think there are some things that can still be analyzed with a small number of mindfully selected XS positions, but I would never take this approach. I do understand the lack of availability of lidar and other remote sensing data to facilitate high-resolution mapping though, but then one has to be thoughtful about what one can reasonably achieve. I think the way forward would be for the authors to explain their viewpoint on why they have a sufficient number of XS for the goals of their study in comparison to the highest density used by the references cited above.*

Response:

We fully agree that the larger the number of cross-sections, the more robust identified correlations will be. Unfortunately, we had to use the dataset as it is as we have currently no means of doing additional field work to enrich it. But we do not think that the "biased" placement impairs the overall methodology. Of course we would be pleased to have the opportunity to test this approach on a much denser dataset in the future.

*p. 8, section 4, line 1. "Hydrological" should be "hydraulic". I believe. These are not interchangeable. Hydrologic would be rainfall-runoff and water balance related, could be purely discharge but discharge alone does not identify riffles vs pools. Hydraulic means on the basis of the depths, velocities, and other flow kinematics.*

Response:

Indeed the correct term is clearly hydraulic in this context.

*p. 8, lines 8-16. I am confused by the writing. On line 8 it says hydraulic data were "surveyed" at 3 discharges. Please clarify that the data were measured in the field and then it is necessary to also describe how the data were measured. There are many different methods possible and one cannot undertake analyses of data without stating how it was collected. Moving on from there, if the data was actually measured, then I have absolutely no idea why the authors mention a method involving 1D hydraulic modeling of the sites. Given field observed cross-sections and hydraulic data, once could use a pure XS analyzer like the old, free software WinXSPro and many other GUIs to extract geometric variables like hydraulic radius with no numerical modeling. If the derivative variables like Rh and Fr are not based on field data, but instead are coming from a 1D hydraulic model, then it opens up a whole can of worms regarding the accuracy of the model outputs, which then necessitates an explanation of model calibration and validation performance. All of this is written unclearly and needs to be revised to explain to readers what is going on. This has profound consequences for evaluating the study.*

Response:

Data are measured in the field; we will add a description of how it was collected from Navratil et al. (2006): "Cross-sections and water surface profile measurements were surveyed in 2002 – 2004 covering the main channel and floodplain and using an electronic, digital, total-station theodolite. Water surface profiles were measured at different flow discharges."

| Reach | Number of cross-sections | Flow discharge surveyed (m3/s) | Gradient |
|---|---|---|---|
| 1 | 14 | 0.22 and 1.26 | 0.0125 |
| 2 | 32 | 1.85 and 2.41 | 0.0044 |
| 3 | 21 | 0.18, 1.13, 1.72, and 1.99 | 0.0018 |
| 4 | 26 | 0.19, 0.33, 0.8, and 11.5 | 0.0024 |
| 5 | 25 | 0.15 | 0.0060 |
| 6 | 36 | 0.21 | 0.0047 |

The numerical model used in this study aims to calibrate the Manning coefficient in order to fit the surveyed water surface profiles. In our revised paper, we will solely rely on measurements at the lowest surveyed discharge and use the model to provide estimates of Manning coefficient n, we will use these values in order to compute the third degree of freedom, bed shear stress $\tau_b(x)$, along the reach.

*Section 4. In the previous section it was stated that hydraulic variables were "surveyed" at 2 low discharges and 1 near bankful discharge. As the relative magnitudes of the variables between rifles and pools are stage dependent, it matters which flow was used to for the analysis in this section. The authors should state that. If somehow all three discharges were used, clarify how. In fact, at-a-station hydraulic geometry is an important tool for identifying riffles and pools more holistically considering the totality of the bankfull channel, so it is too bad few people take note of that and apply it to this purpose.*

Response:

We mentioned in the page 8 at line 3 that we worked with the minimum discharge $Q_{min}$.

*p. 8, section 4, lines 4-5. Most people use PCA for challenging multivariate problems with complex interrelationships that are unknown and thus this is the first way to get a sense of how variables interrelate. That does not characterize the situation for riffle-pool geometry and open channel hydraulics. A wiser strategy here could be to use Buckingham Pi theorem dimensional analysis to create the variables of interest. Also, one can easily reason out that really the variables that matter are those that control or respond directly to morphodynamic processes, such as flow convergence routing or meander migration. That can then guide wise variable selection that is process based. Returning to this list of variables, several of these variables are highly correlated or define each other, so it does not make sense to throw them all into one multivariate analysis as if it is a mystery. For example, bed elevation, max depth, and hydraulic radius are all highly correlated and redundant. Meanwhile, A and P define Rh, so those 3 are also highly correlated. Similarly, Fr is defined by y and u, so the same situation arises. This "throw everything into the soup" strategy of multivariate analysis is not wise and possibly not technically sound, but the authors can review the PCA assumptions and limitations to evaluate that- not worth my time to re-study up on PCA. Even if it is technically ok, it still doesn't make any sense as a strategy as if we do not already know how these variables relate to each other- we do know exactly how they relate.*

*P.9. I am just not understanding why anything in Figure 4 and the associated results text is actually new results or anything other than trivial findings. By definition of variables, A, Rh, and y are positively correlated, while Fr is going to be negatively correlated to y and positively correlated to u. Also, Z has to be negatively correlated to A, Rh, and y. This is all be definition. PCA is not required to know this. Further, I do not agree that the PCA is adding any fundamentally new or useful information for riffle-pool delineation compared to wisely selecting the few independent variables underlying the physics-based analytical relations, especially bed elevation, width, and possibly slope, as together these three control relative velocity between riffle and pool units for a fixed discharge. If the channels are meandering, then thalweg planform curvature would be important, too, as it is well known in the physics to control meander migration. In fact, it is unclear and technically unsound to exclude metrics of channel width from this analysis, as width is the underlying independent variable influencing all the other variables in the list except for detrended bed elevation and depth (which of course are the same thing just inverted and with different vertical datums). The authors need to set up this methodology better to justify why it is necessary and better than what I am proposing as an easier, more process-based approach or else I do not see how this PCA analysis is meritorious.*

*p.9, lines 3-4. The claim that each descriptor adds additional information about the bedforms is easy to show as not true. Rh is defined by A and P, so how is Rh fundamentally new and additional as opposed to using a combination of A and P, unless one defines the mathematical operation of division as adding new content, which it does not. This continues the theme of my last few comments. The authors are applying blind statistical methods to what is a pure analytical problem with 100% defined and known elements. There is no additional information beyond the independent variables and the math operators to combine them into A, Rh, and Fr.*

Response:

In the initial version of the paper, we were comparing the wavelet-based method with two benchmark methods: the BDT (O'Neill and Abrahams, 1984), and an index method that consists in affecting a different numerical value for each class of a given variable/degree of freedom, and then sum these individual index functions into a composite one.

The major concerns are not about the index method in itself, but on the choice of the variables/degrees of freedom. Initially we used the first three axes of a Principal Component Analysis as the degrees of freedom, a choice which has very poor physical meaning. We will entirely cut out this method and focus only on wavelets by using three classical variables: velocity, hydraulic radius (or the closely related cross-sectional averaged depth), and bed shear stress in addition to the planform variable: planform curvature angle.

*p. 8, section 4, line 8. The topic of detrending is a huge issue that requires a bit of unpacking in the writing here, because the outcome of riffle-pool delineation can be largely depending on this very choice based on my own sensitivity analysis of this situation using different detrending methods. Earlier in the manuscripts the authors wisely commented about all the different way different authors measure and analyze pool spacing data (e.g., p.6, line 22). Well, the same challenge arises with detrending. There is no universally right or wrong way given the diversity of purposes for detrending, but each option has consequences for the scientific outcome for a specific purpose, especially for identifying the magnitude and length of residual highs and lows in a bed profile. Without going into all the options, what I request is that the authors state what type of detrending they did. If linear, then was it one line per site (presumably no reach breaks within a site, but there could be) and was care taken to insure that the line began and ended at the same relative elevation to avoid biasing the slope, which is a significant problem.*

Response:

The procedure that we followed to construct the detrended bed elevation relies on the bed elevation and all water surface levels to avoid biasing the slope. For example, given a reach with N surveyed cross-sections in two discharge stages $Q_1$ and $Q_2$, we define the bed elevation as $z_{talweg}(x)$ and the two water surface levels $z_{ws}(x, Q_1)$ and $z_{ws}(x, Q_2)$. For the detrended bed elevation according to that, it is $Z = z_{talweg} - z_{trend}$, where $z_{trend} = S \times x + b_t$, S and $b_t$ are solutions of the system below.

[Figure]

As mentioned in the figure, the bed elevation is in blue, water surface for $Q_1$ is in red and for $Q_2$ in green, the stippled lines are the trends.

*p.9, lines 10-12. This single long sentence attempting to explain a sequence of mathematical steps applied to some data is opaque to me as a reader, as is plot (a) in Figure 5. This should be written out more thoroughly and clearly in steps. For example, presumably the smoothed data is each XS spatial series, but then what constitutes the "sampling" that is "homogenized"? I neither understand the samples nor what homogenization is and why it is needed. Is homogenization the same or different from normalization in this study? If so, why call it two different things that creates reader confusion, but if not then what is it? Sometimes normalization means the strict application of the function that makes the data fit the normal probability distribution while more often it just means to divide variable by another.*

Response:

This data goes through some processes; first, detrending the variables (as bed elevation), then sampling all variables, this process is a linear interpolation with spacing of 1m or smaller. Second, normalizing and centering them which are just the variable minus its mean divided by the standard deviation. The formulation of smoothing was wrong; this treatment that we performed is a general removal of very low

frequency components (wavelength larger than 7 times the mean bankfull width) before applying thresholds. Since we will not use bed elevation anymore, this processing is no longer relevant.

*P. 10, first line. Why is this line bold?*

Response:

It's what's concluded from the previous paragraph which defined the index. As said before, this part will be dismissed.

*P. 10, equation (5). This equation is an all-or-nothing type approach where every location is either classified as riffle or pool for an individual descriptor. This is in contrast to the aforementioned BDT approach that uses a standard deviation tolerance. Also, the method of Pasternack et al. (ESPL 2018a,b) uses a standard deviation tolerance. It would be useful to explain why no tolerance was applied.*

Response:

This section will be completely removed so this discussion is not relevant anymore.

*P.10, line 10. From what I gather considering the equations and the potential values of I, the concept here is that for something to be defined as a riffle or pool versus an intermediate MU type, all three descriptor variables must agree and yield the same heavyside function value of 0 (pool) or 1 (riffle). Conceptually, the authors are substituting a cross-check among 3 variables as the countermeasure to cope with uncertainty in place of tolerance within each variable as the countermeasure for uncertainty. I think putting the concept of the method in words like this would help readers understand the strategy and purpose of the math and procedure that is described. However, looking beyond that, one can ask if this actually works? In other words, is there a resiliency against uncertainty gained by using multiple variables and the specific ones chosen?*

*The authors should address why they think this is so, because this is the kernel of new idea they are proposing but have not actually written out. I have to agree that using more than 1 INDEPENDENT variables would help serve as a check against uncertainty, so that is good idea, but (a) the variables chosen are not independent (both Fr and Rh depend on detrended bed elevation, which is a surrogate for the inverse depth and depth goes into both Fr and Rh) and (b) one can choose to use both a tolerance per BDT and multiple variables per this study. That would yield the best outcome. In Pasternack et al. (ESPL, 2018a,b), we do use both strategies, but for our choice of variables we limit our analysis to only detrended bed elevation and width, as these are the process-based controls on flow convergence routing, they are independent, and they underlie the derivative variables like Fr and Rh. However, we do not use slope, which independently controls velocity and Fr, and we make that choose for a specific process-based reason, but we do exclude it. We also do not look at thalweg planform curvature in those articles, though we have internally thus far. One could reasonably choose to include both slope and thalweg planform curvature. One could also choose to include grain size metrics, as I'm sure prof. Jose Rodriguez would be very insistent on given the importance of that variable to determining relative erosion and deposition on riffle sand pools. Unfortunately, it is incredibly difficult to obtain high-resolution spatial series of substrate grain size as of yet. In any case, I see both positive and negative to what is being done.*

*At a minimum, the authors can explain the general idea in words as I have done, but then also some defense is needed if the authors stand by the decision of variables chosen, because I see the choice as technically unsound given that they are defining each other as explained.*

Response:

As I said before, we will cut out the index method.

*P.11, line 2. I see that p.6 line 10 defined lambda-star as "dimensionless pool spacing", yet here that variable has dimensions of m? Something is wrong.*

Response:

It's a typo! There is no "m".

*p.12, lines 1-2. While most people only apply Fourier analysis to stationary series, the method is not in fact limited as thus, because it can be applied using the "evolutive" methodology to capture non stationary dynamics very similar to what one gets from wavelets. One can reasonably argue that wavelets are superior for non stationary data and because one can apply different wave forms, but to say that Fourier analysis cannot do non stationary analysis is wrong. Many applications of evolutive Fourier analysis exist, but for hydrological data see for example, Pasternack, G. B. and Hinnov, L. A. 2003. Hydro meteorological controls on water level in a vegetated Chesapeake Bay tidal freshwater delta. Estuarine, Coastal, and Shelf Science 58:2:373-393.*

*Section 5.1. I think there is too much redundancy between what was written about wavelets in section 1 (p. 3, lines 20-26) and this section. The introduction can more simply introduce the idea of it and state the scientific questions and hypotheses associated with using it, but then leave the literature review here, so there is only one literature review. My earlier comments about the literature of applying wavelets to geo/hydro data also apply to this section.*

Response:

We will change these paragraphs to:

"Classical mathematical methods, such as Fourier analysis, extract the wavelengths in the frequency domain only for stationary signals but also for nonstationary signals using an "evolutive" methodology based on spectral estimators (Thomson., 1982; Pasternack and Hinnov., 2003). Wavelet transforms can do the same for nonstationary signals and find the localized wavelength but with different waveforms. Analyzing a signal basically consists of looking for similarity between the signal and well-known mathematical functions. In this paper, we use the continuous wavelet transform with the Morlet wavelet (Gabor, 1946) (Fig. 3) applied to spatial series instead of time series, so periods and frequencies in time series are replaced by wavelengths and wavenumbers in spatial series.

The wavelet transform is done by convolving the mother wavelet (the waveform) with the signal data, which begin first with the product of the wavelet and a portion of the signal. That product is then integrated to define a mathematical measure of similarity of that portion of the signal to the reference

wavelet. This process is repeated as the mother wavelet is moved along the signal and also dilated or contracted to different spatial scales. Thus the transform is done in space and scale (or frequency) simultaneously (McKean et al., 2009)."

*p. 15, line 15. Why choose the Orgeval reach, when it is not the longest or having the most XSs? I already deleted my table where I computed the XS density, so does this reach have the highest XS density? Otherwise, why? of course, why not analyze and compare all 6 reaches, as this is a scientific journal article and there could be interesting results in comparing the different reaches? The method itself of applying wavelet analysis to a spatial series is not so novel as to justify limiting to only 1 reach as a single case study.*

*Table 4. I do not understand. Previously it was stated twice that only 1 reach was assessed but now here are data comparing all six reaches. I think the writing of the manuscript should be improved to explain what is going on better. If all six reaches were in fact tested with wavelet analysis, then some comparison between reaches would be interesting for section 5.*

Response:

As I said in the paper this is just a reach example. In the revised paper we will present all the 6 reaches.

*Figure 8. This figure shows a fundamental problem with the wavelet methodology as the preferred tool for mapping riffle and pools as well as quantifying their spacings. Specifically, it cannot return results for some distance at the start and end of the spatial series. In the case shown, there is only results for the range of ~ 81-241m out of 318 m. That leaves a whopping 50% of the reach unassessed. Wow. That's a lot of lost information. Of course, the longer the series and the more frequent the XS sampling, the less loss, but there will always be a loss. This makes the method less valuable than alternatives that retain the information.*

*Section 6.2 This section now states that the comparison is limited to only 81<x<241 m. That's problematic because it's not a fair test of the actual utility of the wavelet method leaving half the reach unevaluated. This should be stated clearly. The comparison is still useful but it does have this huge caveat. A method that leaves half the reach unevaluated can never be better than one that assess the whole reach, if the goal is to characterize the whole reach.*

*Section 6.1. Authors must clarify if the score technique is applied to the entire reach length or only the length for which results overlap. I think one must count the whole reach as it is a deficiency of the wavelet method that it leaves 50% of reach 6 unevaluated. Whatever the authors are doing, they should clarify that.*

Response:

The origin of this problem is the Cone of Influence; it is the region of the wavelet spectrum in which edge effects become important.

Of course we can say the same think for reach length and number of morphological units as for the number of cross-sections: the larger it is, the more robust the results will be, and the smaller the relative portion of "unassessed length" will be. Edge effects due in the cone of influence are clearly a limitation of the wavelet analysis, since it is an analysis in space and scale $(x,k)$ simultaneously. The method has the drawbacks of its advantages, it still remains a powerful tool for non-stationary analysis. Some authors choose to pad the data series with zeros in order to get results on the whole available length. We chose not to use such a padding, which may introduce bias. We prefer a shorter series of local wavenumber $K(x)$, than a longer, potentially biased one.

*p. 17, results header. Some authors like to blend methods and results in paired couplets working through a manuscript, and that is most appropriate when one couplet build son the results of another, but then one would not call section 6 here a results section, as many results have already been presented. If I was the associated editor for this manuscript, I would require the authors to separate the methods content from the results content and go with the traditional ordering of the scientific method, because there is no reason not to. One can state the methods from sections 4 and 5 in one unified methods section and then state the results in a unified results section. As the two sections do not build on each other, then one does not need to use the couplet approach. Then, one can have methods and results subsections for the inter comparison analyses. Finally, discussion should stand alone after all results are presented.*

*Section structure. I think there are problems with the way the manuscript's sections are structured. In general I can follow what the paper is trying to do, but the structure would be better following a traditional scientific method with all methods first then all results second, ands then all actual discussion last. By mixing them all up it is somewhat confusing and more importantly, impossible to tell what methods have answered what important scientific questions. For example, from the structure it is difficult to tell if this study is only a methodological comparison or also a scientific contribution presenting new results about pool spacings that can be compared with the results of other studies. It would be a shame to do all this work and have no contribution to the question of pool spacing in different river types. But getting back to my main concern here, the discussion, if present at all, iOS hidden in bits throughout the manuscript and would work better if isolated and thoroughly presented.*

Response:

That's true, for that we will reorganize as presented before.

*Discussion section 6.3. These paragraph primarily consistent of more results not previously present, but there is a bit of discussion, too. Specifically, all the text in this section from page 19 line 18 to page 21 line 20 are purely results. In fact, p. 21 line 10 even says, "these results: : :" so the authors view these as results too. Really, there is no suitable discussion putting the results of this study into the larger context of methods and results about riffle-pool ID-ing and quantifying their spacings. There should be such a discussion.*

Response:

These results will be transferred to the discussion section. The part where we said "these results" is just an error of wording. In the discussing part we will discuss also results of cross-section spacings and their influence on the results according to the Pasternack et al. (2018b) and Legleiter (2014b).

[revised manuscript text omitted]


[revised manuscript text omitted]


chosen channel width $w$ (e.g., Richards (1976a, b); Dury (1983)) instead of)), otherwise others have used bankfull channel width $w_{bf}$ (e.g., Leopold et al. (1964)). These differences raise questions about the efficiencycertainty of these ratios (dimensionless pool spacings $\lambda^*$) and their dependence on geometric or hydraulic parameters. Moreover, the majority of researchers use the average channel width instead of the bankfull width because both give a similar pool-riffle spacing interval. Here, we are working with $w_{bf}$wbf and with a newtwo wavelength calculation methodmethods (Wavelet & index) that we present in section 3identify 
[revised manuscript text omitted]

 are calculated for the minimum discharge used in this study $Q_{min}$.

[revised manuscript text omitted]

$$\phi(x, s) = Im(\ln W[f(x)](x, s)) \arctan\left(\frac{Im(W[f](x,s))}{Re(W[f](x,s))}\right) \tag{811}$$

[Figure]

Figure 7. Wavelet Morlet mother function, the plot gives the real part and the imaginary part of the wavelets in the space domain (distance).

To respect the nomenclature in the spatial definition and facilitate the extraction, we choose the angular wavenumber (in rad.m$^{-1}$) $k = \frac{2\pi}{\lambda}$ instead of the scale factor. We associate a wavelength $\lambda = 2\pi\alpha s$ with the scale parameter s, where $\alpha$ is the Fourier factor associated with the wavelet, and with $k_\psi = \frac{1}{\alpha}$ the peak wavenumber of the mother wavelet, and

$$\alpha = \frac{2}{\beta + \sqrt{2 + \beta^2}} \tag{912}$$

$$s = \frac{1}{\alpha k} \frac{k_\psi}{k}$$

(10)

1. Thus, the wavelet transform of the function f(x) is defined in the space-wavenumber as:

$$W[f]: \mathbb{R} \times \mathbb{R}_+^* \to \mathbb{C}$$
$$(x,s) \to \sqrt{\alpha k} \int_{-\infty}^{+\infty} f(\eta)\psi^*\big(\alpha k(\eta - x)\big) d\eta$$

(11)

2. Except for the channel angle, all input variables are always positive and may substantially vary in magnitude

3. so we perform

4.  the wavelet transform on the Neperian logarithm of these variables. The whole analysis is performed

5. in *Scilab*, using an adaptation of the toolbox by Torrence and Compo (1998)

6. [*atoc.colorado.edu/research/wavelets/*].

7. The complex

8.  wavelet transform can be

9. classically visualized using a scalogram, i.e., a colored map of the modulus $R(x,k)$ in the $(x,k)$ plane (Fig. 5

10. bottom). The wavelet analysis neglects parts of the signal at both extremities of the series: this is the *cone*

11. *of influence* (Torrence and Compo., 1998), the region of the wavelet spectrum in which edge effects become

12. important. However, as explained previously, the complex transform also yields a phase Phi(x,k) in rad (Eq.

13. (8)) which can also be plotted in the same plane (Fig. 5 top). In our study, we will search for special

14.  space/wavenumber curves mainly using the phase information,

15. i.e. search for *phase* ridges as opposed to *amplitude*  ridges (Lilly and

16. Olhede, 2010).

17. In section 3.2, we give a rigorous definition of Wavelet Ridge

18.  points and curves in a

19. univariate case (i.e., a single spatial series). Then, in section 3.3, we generalize the definition to the

20. multivariate i.e. when the series consists in several correlated variables .

21. ## 3 – 2 – Univariate case:

22. In the univariate case, we choose a single variable $f$ (velocity, hydraulic radius, bed shear stress, or local

23. channel direction angle).

24.

25.

26.

27.

28.

29. For the wavelet $\psi(\eta)$, the ridge point of $W[f](x,s)$ is a space/wavenumber pair $(x, k)$ satisfying the

30.  *phase ridge point conditions* (Lilly and Olhede, 2010):

$$\frac{\partial}{\partial k} Re(\ln W[f](x,k)) = 0$$

(15)

$$\frac{\partial}{\partial x} Im(\ln W[f](x,k)) - k = \alpha k k_\psi = 0$$

(12)

31.

1  or, according to the definition of the phase (Eq. 8) :

2
3
4

5
6
7

$$\left.\frac{\partial \phi}{\partial x}\right|_{(x,k)} - k = 0 \qquad (13\cancel{17})$$

9  This condition states that  the rate of change of transform phase
10  at scale $k$ exactly matches $k$ at location $x$; from this condition,
11  K(x) takes the instantaneous frequency of the signal can be derived (Lilly and Olhede,
12  2008 ; Lilly and Olhede, 2010). The sets of points satisfying the condition form
13

14
15
16   a parametric curve (ridge curve) noted $(x, K(x))$
17   defined by:

$$\left.\frac{\partial \phi}{\partial x}\right|_{(x,K(x))} - K(x) = 0 \qquad (14\cancel{18})$$

18   This property is illustrated in Fig. 5,
19  where a ridge curve is superposed both on the scalogram and on the phase map.
20

21  There may be several curves that verify the Eq. 14; in practice we choose curves that
22  cross continuously  the domain of the wavelet transform (from one cone of
23  influence to another) and belong to the region where a maximum power of the wavelet is. This curve K(x)
24   It also represents the local wavenumber, which is defined on a support $\ell <$ L named assessed length, with
25  L the total reach length.

26  The phase function $\Phi$ is then obtained by evaluating the function $\phi(x, k)$ along the curve (x, K(x)), in thick
27  black in Fig. B1 (A) in Appendix B 1.

$$\Phi(x) = \phi(x, K(x)) \qquad (15\cancel{19})$$

28  In the end, we can extract the wavelength function of pool-riffle sequences, which corresponds to a pseudo-
29  period function of the signal $f$, and which is:

$$\lambda(x) = \frac{2\pi}{K(x)} \qquad (16\cancel{20})$$


[Figure]

Figure 5: First plot: the phase function from which we get the function K(x); Second plot: the power of the wavelet with the region where there is maximum variability depicted by the black curve K(x) (ridge curve). These two figures are represented in a wavenumber/distance space for the Olivet River and the wavelet transform is performed on the logarithm of the velocity. The part of the figure with low opacity shows the cone of influence which is neglected in this study (edge effects are more important for short wavelengths than for long wavelengths).

Also, the shape's amplitude $A_m$, with which pools and riffles vary, is corrected by a coefficient $\sqrt{\frac{1}{\alpha K(x)}}$. This correction comes from the inversion of the direct transformation equation (Eq. 10) which holds the coefficient $\sqrt{\alpha K(x)}.\sqrt{\frac{k_\psi}{K(x)}}$

$$A_m(x) = |W[f](x, K(x))| \sqrt{\frac{1}{\alpha K(x)}} \sqrt{\frac{k_\psi}{K(x)}} = R(x, K(x)) \sqrt{\frac{1}{\alpha K(x)}} \sqrt{\frac{k_\psi}{K(x)}} \qquad (17\cancel{21})$$

The signal is locally similar to a sinusoid $f_{mod}$ of wavenumber K in rad.m$^{-1}$ which model the variability $f$. We can define the pseudo-periodic variable as presented in the Fig. 6 with:

5   3   Pool-riffle identification:

In this part, we apply the ridge wavelet analysis, explained in the previous subsection, to the spatial function z (bed elevation) and we limit the study only with univariate analysis of the pool riffle sequences of the

1
2
3

4

$$f_{mod}(x) = A_m(x) \cos(\Phi(x)) = A_m(x) \cos(\phi(x, K(x))) \; \cancel{I_W = A_m \cos(\Phi(x))}$$

(18)

5 In the example below (Fig. 6), the modeled velocity function follows the variability of the observed velocity,
6 it is a pseudo-periodic, continuous function that approximates the first-order variability of this hydraulic
7 parameter across pool-riffle sequences. The statistics of the K(x) function can be translated into statistics of
8 longitudinal spacings of alternating bedforms, e.g. mean spacing $\lambda^*_{mean}$, median spacing $\lambda^*_{median}$ or
9 spacing standard deviation $\sigma(\lambda^*)$. In Fig. 6 we would find $\lambda^*_{mean} \approx 8.7$, $\lambda^*_{median} \approx 9.12$, and $\sigma(\lambda^*) \approx$
10 0.79 if we were to analyze velocity only ; The pseudo-periodicity of $v_{mod}$ yields to the identification of 6
11 pools (white) and 7 riffles (gray).

[Figure]

13 **Figure 6**: variation of the modeled function $f_{mod}$ which represent the pseudo-periodic variable (e.g., the velocity of
14 the Olivet River) compared to the observed one. This pseudo-periodicity yields to the identification of pools (white)
15 and riffles (gray) in the plot below. The not studied part is due to the cone of influence of the wavelet method.

17 In the next section, we will extend the definition of phase ridge points and ridges to the case where several
18 variables are sampled along the reach, all of them potentially correlated and embedding information about
19 the pseudo-periodicity of channel hydraulic behavior.

20 **3 – 3 – Multivariate case:**

1    The multivariate case is the extension of the univariate to a set of N real-valued signals, we use the
2    coevolution of more than one variable to extract the wavelength of the reach and therefore identify the pool-
3    riffle sequences. We start by computing the wavelet transform for each variable $i = 1..N$ and extract their
4    phase functions $\phi_i(x, k)$. According to the previous section, univariate ridges curves $K_i(x)$ would be defined
5    by:

6    In fact, we present a wavelet power plot (Fig. 8 (A)) applied to bed elevation z in the wavenumber/distance
7    space to show the higher energy region and the extracting ridge curve that contains all the information about
8    the reach (wavelength, phase, and amplitude). The function $I_W$ translates the ridge curve variability in the
9    spatial domain, it is a frequency- and amplitude- modulated sinusoid, and as a result, it gives the bedform
10   variability represented in Fig. 8 (B). This plot shows great conformity between the bed elevation and the
11   signal identified, and the advantage of this method is that, at different scales, it not only gives the wavelength
12   of the pool-riffle repetitions, but it gives the amplitude of each bedform shape associated with the
13   wavelength. Moreover, wavelet ridge analysis can be extended to the multivariate case (Lilly and Olhede,
14   2009).

[Figure]

16   **Figure 8**. Steps to identify pool-riffle sequences for the Orgeval reach using bed elevation z with the wavelet method.
17   (A) Power of the wavelet, which is represented in a wavenumber/distance space, with the region where there is
18   maximum variability depicted by the white curve (ridge curve); (B) variation of $I_W$ function compared to the bed
19   elevation z. (C) Identification of pools (black), riffles (white), and intermediate morphologies (gray).

20   If we analyze the results of this method on the Orgeval reach (6), we find that the wavelet method eliminates
21   part of the bed elevation details and takes the part that contains the high power (Fig. 8 (A) and (B)) on which
22   it identifies the bedforms graphically. Figure 8 (C) shows us the pool-riffle sequences in spectral

$$\left.\frac{\partial \phi_i}{\partial x}\right|_{(x,K_i(x))} - K_i(x) = 0 \tag{19}$$

|  | | | |  |  |
|---|---|---|---|---|---|
|  |  |  |  |  | |
|  |  |  |  | | |
|  |  |  |  | | |

But then the local wavenumber would be specific to a given variable, which is not what we would like. We would rather like to find a common wavenumber for all variables at location $x$, i.e. such that:

~~In this section, we present the results, the comparison between methods and the discussion. First, we introduce the comparison method based on a score technique that compares the two methods index and wavelet. Second, we present the application of this comparison on the index and wavelet methods, and we introduce the BDT results. Finally, these results are discussed according to data presented in section 3.~~

~~Let us assume for example that the wavelet method identifies $n$ pools and $m$ riffles, and the index method identifies $n'$ pools and $m'$ riffles. Consequently, for the wavelet method, the pool intervals of this type of reach are $\widetilde{P_W} = \bigcup_{i=1}^{n} \tilde{p}_i^W$ and the riffle intervals are $\widetilde{R_W} = \bigcup_{i=1}^{m} \tilde{r}_i^W$. On the other hand, the pools and riffles identified by the index method are successively $\widetilde{P_I} = \bigcup_{j=1}^{n'} \tilde{p}_j^I$ and $\widetilde{R_I} = \bigcup_{j=1}^{m'} \tilde{r}_j^I$ where $\tilde{p}_i^W, \tilde{r}_i^W, \tilde{p}_j^I$, and $\tilde{r}_j^I$ are successively pool and riffle interval abscissa number i identified by the wavelet method, and pool and riffle interval abscissa number j identified by the index method. Thus, the score is defined as:~~

$$\left.\frac{\partial \phi_i}{\partial x}\right|_{(x,K(x))} - K(x) \approx 0 \qquad \forall i \, \cancel{score} = \frac{\ell(\widetilde{P_I} \cap \widetilde{P_W}) + \ell(\widetilde{R_I} \cap \widetilde{R_W})}{\ell} \tag{2023}$$

1  The identification of a "master" ridge point/curve is now a minimization problem. We will define it as a

2  local minimum of the squared norm of the vector $\left(\frac{\partial \phi_1}{\partial x}\Big|_{(x,k)} - k, \frac{\partial \phi_2}{\partial x}\Big|_{(x,k)} - k, \dots, \frac{\partial \phi_N}{\partial x}\Big|_{(x,k)} - k\right)$:

3

4

5

11
12
13
14

15
16
17

$$E(x,k) = \sum_{i=1}^{N} \left(\frac{\partial \phi_i}{\partial x}\Big|_{(x,k)} - k\right)^2 \widetilde{P_t} \cap \widetilde{P_w} \qquad (21\underline{24}$$
$$ \qquad (25)$$
$$$$

[revised manuscript text omitted]

13  In the following section, we will compare the wavelet method with a benchmark method using talweg
14  elevation.

15  **4 – 2 – Comparison with benchmark method:**

In this section, we compare our method's results with a selected benchmark method from the literature. It is called the Bed form Differencing Technique (BDT) introduced by O'Neill and Abrahams (1984). We choose this method instead of threshold methods because the latter require thresholds (expert judgment) collected from the field, which is not possible in our case.

The technique of O'Neill and Abrahams (1984) (BDT)Table 4. The score technique results

To validate and discuss these two methods, the technique of O'Neill and Abrahams (1984) (BDT) is chosen. This technique uses a tolerance value (T), which defines the minimum absolute value needed to identify a pool or a riffle (Krueger and Frothingham, 2007). It is calculated using the standard deviation ($S_D$) of the series of bed elevation differences from upstream to downstream for each reach and corrected by a coefficient chosen according to the reach. For this, we test several tolerance values, and for the Graulade (1), Ozanne (4), Avennelles (5) and Orgeval (6) reaches we find the same results. We choose to check onetwo tolerance valuevalues for each reach with T = $S_D$. This method gives pools and riffles positions by assigning a crest as a riffle and a bottom as a pool and therefore the computation of the wavelengths becomes a little difficult. So, we chose to calculate a series of pool-pool and riffle-riffle spacings, their medians, and standard deviations and then calculate their averagesT = 2$S_D$.

This was applied to all rivers and the results are depicted in the Fig. 10. Statistics of the BDT are shown in Table 5 which displays a comparison between these two types of morphological units' identification and mostly the identification of an average wavelength of the reach.

Fig. 10 shows the BDT results on all reaches, this method relies only on topography to determine the positions of pools and riffles, moreover it also uses a threshold T (tolerance) but the technique does not need a calibration reach or field investigation to know how to set this threshold. In this figure, Round points are pools or riffles and from these points we can calculate the wavelengths and longitudinal spacing of each reach as we stated before.

The work of the wavelet analysis is done on the assessed length $\ell$. However, the BDT method works on the total length of the reaches. This was done to determine how effective the wavelength extracted by the wavelet analysis can represent the entire reach even if an entire part is left unassessed.

For the wavelet method (Fig. 98), the wavelength extraction is among its objectives, while for the index and BDT, the computation of the wavelength is done a sort of multiple calculations. To compare these two methods, we will use only the longitudinal spacing ($\lambda^*$) as a criterion.



[revised manuscript text omitted]

questions about the naturality of the rivers. In our case, the rivers are subject to artificial modifications (e.g., bridges, weirs) and rehabilitations, which will have a significant impact on the hydro-morphological parameters (width, depth, meandering, etc.). This can have a very important impact on the identification of pseudo-periods.

The wavelet ridge analysis is powerful in identifying pseudo-periods, amplitude and phase while respecting the correlations between parameters. We can thus identify alternating morphological units in a more objective way in terms of frequency/wavenumber. From this extracted common wavelength using the flow parameters, it is possible to represent the topography continuously.

On the other hand, it represents drawbacks compared to other methods. First, the cone of influence that ignores a large part of the river and sometimes biases the results (in the case of the Graulade (1) and Semme (2) reaches) in the case of small total lengths. It is the region of the wavelet spectrum in which edge effects become important. We can say the same thing for reach length and number of morphological units as for the number of cross-sections: the larger it is, the more robust the results will be, and the smaller the relative portion of "unassessed length" will be. Still, the method remains a powerful tool for non-stationary analysis. Another problem is the amplitude which is sometimes overestimated in some regions of the topography. We visualized this in several cases in our study, since we used the Neperian logarithm to avoid negative values and therefore the inverse function (exponential) will give slightly larger values. However, this does not bias the identified wavelength of the reach.

**6 – Conclusions:**

In this study, we present an automatic procedure based on Wavelet Ridge extraction to identify some characteristics of alternating morphological units (MU), such as their longitudinal spacing and amplitude. The method does not rely on any a priori thresholds to identify MU sequences. It was applied to six rivers with a maximum length of 500 meters. We chose to work with classical hydro-morphological variables (velocity, hydraulic radius, bed shear stress) in addition to the local channel direction angle that evaluates the impact of river sinuosity in the determination of the wavelength.

On the overall, identified wavelength are consistent with values of the literature (mean in 3-7 $w_{bf}$). The use of a multivariate approach yields more robust results than the univariate approaches, by ensuring a consistent covariance of flow variables in the pseudo-periodic behavior.

Given the short length of several reaches, the relatively small number of cross-sections for each reach, and the possible impacts of artificial modifications, this paper is mainly a proof-of-concept of the wavelet approach. It does not preclude the long-term possibility of extending the work to other rivers with other types of MUs, other longer reaches with a large number of cross-sections.

The comparison shows higher amplitudes and standard deviations for the BDT than the wavelet method, which yields a higher signal-to-noise ratio $SNR = \frac{\mu(A_m)}{\sigma(A_m)}$. These higher values mean good wavelet performance to extract more useful information than the noise.

| Reaches | 1: Graulade | | 2: Semme | | 3: Olivet | | 4: Ozanne | | 5: Avennelles | | 6: Orgeval | | Mean | |
|---|---|---|---|---|---|---|---|---|---|---|---|---|---|---|
| Variables | P | R | P | R | P | R | P | R | P | R | P | R | P | R |
| $Fr_{index}$ | 0.30 | 0.37 | 0.09 | 0.12 | 0.01 | 0.04 | 0.08 | 0.24 | 0.10 | 0.36 | 0.06 | 0.22 | 0.11 | 0.23 |
| $Fr_{wav}$ | 0.30 | 0.36 | 0.10 | 0.11 | 0.02 | 0.03 | 0.10 | 0.23 | 0.12 | 0.27 | 0.10 | 0.19 | 0.12 | 0.20 |

| | | | | | | | | | | | | | |
|---|---|---|---|---|---|---|---|---|---|---|---|---|---|
| Rh$_{index}$ | 0.18 | 0.15 | 0.35 | 0.26 | 0.54 | 0.37 | 0.29 | 0.14 | 0.23 | 0.11 | 0.42 | 0.19 | 0.33 | 0.19 |
| Rh$_{wav}$ | 0.18 | 0.15 | 0.32 | 0.30 | 0.50 | 0.44 | 0.24 | 0.15 | 0.20 | 0.15 | 0.33 | 0.23 | 0.29 | 0.23 |
| u/y$_{index}$ | 1.29 | 1.87 | 0.29 | 0.44 | 0.03 | 0.15 | 0.25 | 1.17 | 0.40 | 2.87 | 0.23 | 1.28 | 0.42 | 1.31 |
| u/y$_{wav}$ | 1.28 | 1.92 | 0.34 | 0.44 | 0.06 | 0.09 | 0.36 | 1.09 | 0.68 | 2.04 | 0.57 | 1.06 | 0.55 | 1.12 |

**Table 6.** Hydraulic variables results for all reaches for the Froude Number Fr, the hydraulic radius Rh, and the ratio velocity/ depth u/y, these variables are averaged on the number of pools and riffles for each reach.

To conclude, pools and riffles have a subjective definition, and this study is a step towards a more quantitative approach. On the one hand, the wavelet method identifies bedforms and extracts their wavelength by focusing on the most revealing part of the reach, which contains all the reach's information, but what differentiates this method from the others is its capability to extract the pool riffle amplitudes. It could be extended in a multivariate case with a quantitative approach by crossing several variables. On the other hand, the index method suggests identifying all the major and minor bed profile undulations, but it cannot extrapolate a wavelength to all the reaches based on only a small part of this reach.

**Appendix A: List of symbols**

$A_{i,mod}$:     Signal amplitude of the shape of the modeled variable number i

$A(x)$:     Cross-section areaWetted surface (m²)

$A_m$:     Signal amplitude of the shape

$\cos(\theta)$:     Cosine of local channel direction angle

$\cos(\theta)_{mod}$:     Modeled cosine of local channel direction angle

$f$:     Space series function (m)

$f_i$: Fr:     Measured space series functionFroude number i

$f_{i,mod}$:     Modeled space series function number i with multivariate wavelet analysis

$f_{mod}$:     Modeled variable with the univariate wavelet analysis

g:     Acceleration due to gravity and its value is 9.81 m.s$^{-2}$

J$_l$:     Energy slope (m.m$^{-1}$)Index function

$I_w$:     Identified signal that represents alternating pools and riffles

$k(x)$:     Wavenumber (rad.m$^{-1}$)

$K(x)$:     Local wavenumber that corresponds to the maximum variance of the signal (rad.m$^{-1}$)

Ks:     Strickler coefficient

$k_\psi$:     Peak wavenumber, which is the spatial frequency domain that has a maximum amplitude (rad.m$^{-1}$)

$\ell$:     K(x) support (m)

L:     Total reach length (m)

$\ell[a;b]$:     Length of [a; b] (m)

N:     Number of total chosen variablesselected descriptors

n and m:     Manning's roughness coefficientNumber of pools and riffles identified, respectively, by the wavelet method

n' and m':     Number of pools and riffles identified, respectively, by the index method

p:     Normalized descriptor

$p_x$:     Descriptor's value

$\tilde{p}_i^W$:     Interval of the abscissa of the ith pool identified by the wavelet method

$\tilde{p}_j^I$:     Interval of the abscissa of the jth pool identified by the index method

$p_m$:     Descriptor's mean

$P(x)$:     Wetted perimeter (m)

$\tilde{P}_I$:     Pool interval identified by the index method

$\tilde{P}_W$:     Pool interval identified by the wavelet method

$Q_{min}$: Minimum discharge modeled (m3.s$^{-1}$)

$R(x,s)$ or $R(x,k)$ or $R$ — Absolute value or modulus of the wavelet transform at a position x and with a scale s or wavenumber k

$R_h$ — Hydraulic radius (m)

$R_{h,mod}$  — Modeled hydraulic radius (m)

$s$: Dilation or scale factor

$S$: Reach slope (m.m$^{-1}$)

$S_D$: Standard deviation of the bed elevation diffrence series (m)

$T$: Tolerance value, which is the minimum absolute value of the cumulative elevation change required for the identification of pools and riffles (m)

$u(x)$ or $v(x)$: Velocity (m.s$^{-1}$)

$v_{mod}$  Modeled  velocity (m.s$^{-1}$)

$v_{obs}$ Mesured velocity (m.s$^{-1}$)

$w(x)$: Reach width in the x abscissa (m)

$w_{bf}$: Reach bankfull width (m)

$w_m$ Mean width (m)

$x$: Translation factor in the wavelet transform or the abscissa position (m)

$y = y_{max}$: Water depth measured from the the talweg elevation y = z$_{ws}$ -z (m)

$y_m$: Mean depth (m)

$z$ or $z_{t,Obs}$: Measured bed  elevation or talweg elevation (m)

$z_{t,Uni}$: Modeled bed elevation using the univariate wavelet analysis (m)

$z_{t,Multi}$: Modeled bed elevation using the Multivariate wavelet analysis (m)

$z_{ws}$: Water surface elevation measured from the 0NGF (m)

$\alpha$: Fourier factor associated with the wavelet (m.rad$^{-1}$)

$\beta$: Dimensionless frequency taken to be 6 recommended by Torrence and Compo (1998)

$\lambda$: Reach wavelength (m)

$\lambda^*$: Typical pool (riffle) spacing or dimensionless reach wavelength or longitudinal spacing

$\rho$  Water density (997kg/m$^3$)

$\sigma(w)$: Standard deviation of the width along the reach (m)

$\sigma()$  Standard  deviation

$\tau_b(x)$: Bed shear stress in the x abscissa (Pa)

$\tau_{b,mod}(x)$: Modeled bed shear stress in the x abscissa (Pa)

$\theta(x)$: local channel direction angle which is the local angular deviation of the channel direction from a lower-frequency curve (degrees)

$\Phi$: Corresponding phase at the position x and the wavenumber K with $\Phi(x) = \phi(x, K(x))$ (rad)

$\Phi_i$: Phase at the position x and the wavenumber K for the variable number i

$\phi(x,s)$ or $\phi(x,k)$ or $\phi$: Phase or argument at a position x and with a scale s or wavenumber k (rad)

$\phi_i$: Phase of the variable number i

$\psi$: Mother wavelet function

$\psi_{s,x}$: Daughter wavelet function

**Appendix B: Mathematical calculus for the wavelet transform**

**1 – The univariate case**

The conjugate form of the mother wavelet is:

$$\psi^*(\eta) = \pi^{-\frac{1}{4}} e^{-i\beta\eta - \frac{\eta^2}{2}} \tag{B1}$$

whose derivative in relation to the mute variable $\eta$ is:

$$\psi^{*\prime}(\eta) = -\pi^{-\frac{1}{4}}(i\beta + \eta)e^{-i\beta\eta - \frac{\eta^2}{2}}$$
$$= -(i\beta + \eta)\psi^*(\eta) \tag{B2}$$

In  section 3 - 1, $\eta$ is a mute integration variable and $x$ appears only in the argument $\alpha k(\xi - x)$ of the function $\psi^*$. By applying the derivation formula of a composite function, the derivative of the wavelet transform is expressed by:

$$\frac{\partial}{\partial x}W[f(x)](x,k) = \sqrt{\alpha k}\int_{-\infty}^{+\infty} f(\eta)\frac{\partial}{\partial x}\left[\psi^*(\alpha k(\eta - x))\right]d\eta$$

$$= \sqrt{\alpha k}\int_{-\infty}^{+\infty} f(\eta)(-\alpha k)\psi^{*\prime}(\alpha k(\eta - x))d\eta$$

$$= (\alpha k)\sqrt{\alpha k}\int_{-\infty}^{+\infty} f(\eta)(i\beta + \alpha k(\eta - x))\psi^*(\alpha k(\eta - x))d\eta \tag{B3}$$

$$= (\alpha k)\sqrt{\alpha k}\int_{-\infty}^{+\infty} [(i\beta - \alpha kx)f(\eta) + \alpha k\eta f(\eta)]\psi^*(\alpha k(\eta - x))d\eta$$

$$= (\alpha k)(i\beta - \alpha kx)W[f(x)](x,k) + (\alpha k)^2 W[xf(x)](x,k)$$

On the other hand, we have:

$$\frac{\partial}{\partial x}W[f(x)](x,k) = \frac{\partial}{\partial x}\left(R(x,k)e^{i\phi(x,k)}\right)$$
$$= \left[\frac{1}{R(x,k)}\frac{\partial R(x,k)}{\partial x} + i\frac{\partial \phi(x,k)}{\partial x}\right]R(x,k)e^{i\phi(x,k)} \tag{B4}$$

$$\frac{\partial}{\partial x}\text{Re}\left(\ln W[f(x)](x,k)\right) = \frac{1}{R(x,k)}\frac{\partial R(x,k)}{\partial x} = \text{Re}\left(\frac{1}{W[f(x)](x,k)}\frac{\partial}{\partial x}W[f(x)](x,k)\right)$$

$$\frac{\partial}{\partial x}\text{Im}\left(\ln W[f(x)](x,k)\right) = \frac{\partial \phi(x,k)}{\partial x} = \text{Im}\left(\frac{1}{W[f(x)](x,k)}\frac{\partial}{\partial x}W[f(x)](x,k)\right) \tag{B5}$$

Finally:

$$\frac{\partial \phi(x,k)}{\partial x} = \mathrm{Im}\left((\alpha k)(i\beta - \alpha k x) + (\alpha k)^2 \frac{W[xf(x)](x,k)}{W[f(x)](x,k)}\right)$$

$$\frac{\partial \phi(x,k)}{\partial x} = (\alpha k)\beta + (\alpha k)^2 \mathrm{Im}\left(\frac{W[xf(x)](x,k)}{W[f(x)](x,k)}\right)$$

(B6)

1    The previous expression numerically avoids the derivative of the function $\phi(x,k)$, which varies quickly for

2    large wavenumbers.

[Figure]

[Figure]

1 .

**Figure B1**. Steps of determining the local wavenumber K(x) using the wavelet univariate ridge analysis of the the velocity of the Olivet (3) reach represented in the four panels. (A) The phase function $\phi(x, k)$; (B) the power's cone of influence of the wavelet to characterize the region where there is a maximum variability of the velocity in Neperian logarithm; (C) the function $\frac{\partial\phi(x,k)}{\partial x}$; (D) the function $k$.

**2 – The multivariate case**

In the multivariate case, we should resolve the Eq. 20 which contain three derivatives to compute. The first one is already done in the univariate case which is:

$$\frac{\partial\phi_i(x,k)}{\partial x} = (\alpha k)\beta + (\alpha k)^2 \text{Im}\left(\frac{W[xf_i(x)]}{W[f_i(x)]}\right) \tag{B7}$$

The second one is the computation of $\frac{\partial^2\phi_i(x,k)}{\partial k\partial x}$:

$$\frac{\partial^2\phi_i(x,k)}{\partial k\partial x} = \alpha\beta + 2\alpha^2 k \text{Im}\left(\frac{W[xf_i(x)]}{W[f_i(x)]}\right) + (\alpha k)^2 \text{Im}\left(\frac{\partial}{\partial k}\left(\frac{W[xf_i(x)]}{W[f_i(x)]}\right)\right) \tag{B8}$$

For that we should develop $\frac{\partial}{\partial k}\left(\frac{W[xf_i(x)]}{W[f_i(x)]}\right)$:

$$\frac{\partial}{\partial k}\left(\frac{W[xf_i(x)]}{W[f_i(x)]}\right) = \frac{1}{W[f_i(x)]}\frac{\partial W[xf_i(x)]}{\partial k} - \frac{W[xf_i(x)]}{(W[f_i(x)])^2}\frac{\partial W[f_i(x)]}{\partial k}$$

We calculate each derivative:

$$\frac{\partial W[f_i(x)]}{\partial k} = \left(\frac{1}{\sqrt{k}}\frac{\partial\sqrt{k}}{\partial k}\right)W[f_i(x)] + \sqrt{\alpha k}\int_{-\infty}^{+\infty} f(\eta)\frac{\partial}{\partial k}\left[\psi^*\big(\alpha k(\eta-x)\big)\right]d\eta$$

1  $= \left(\dfrac{1}{2k}\right) W[f_i(x)] + \sqrt{\alpha k} \displaystyle\int_{-\infty}^{+\infty} f(\eta)\alpha(\eta - x)\psi^{*\prime}\big(\alpha k(\eta - x)\big)d\eta$

2  $= \left(\dfrac{1}{2k}\right) W[f_i(x)] + \sqrt{\alpha k} \displaystyle\int_{-\infty}^{+\infty} f(\eta)\alpha(\eta - x)\big(i\beta + \alpha k(\eta - x)\big)\psi^{*}\big(\alpha k(\eta - x)\big)d\eta$

3  $= \left(\dfrac{1}{2k}\right) W[f_i(x)] + \sqrt{\alpha k} \displaystyle\int_{-\infty}^{+\infty} [(i\beta - \alpha^2 k x^2) + (-i\beta\alpha + 2\alpha^2 k x)\eta - (\alpha^2 k)\eta^2]f(\eta)\psi^{*}\big(\alpha k(\eta - x)\big)d\eta$

4  $= \left(\dfrac{1}{2k} - \alpha^2 k x^2 + i\beta\alpha x\right) W[f_i(x)] + (2\alpha^2 k x - i\beta\alpha)W[x f_i(x)] - (\alpha^2 k)W[x^2 f_i(x)]$

5  We find a general formulation with p=0…N:

6  $\dfrac{\partial W[x^p f_i(x)]}{\partial k} = \left(\dfrac{1}{2k} - \alpha^2 k x^2 + i\beta\alpha x\right) W[x^p f_i(x)] + (2\alpha^2 k x - i\beta\alpha)W[x^{p+1} f_i(x)]$

7  $\qquad\qquad\qquad - (\alpha^2 k)W[x^{p+2} f_i(x)]$

8  The third one is the computation of $\dfrac{\partial^3 \phi_i(x,k)}{\partial k^2 \partial x}$:

$\dfrac{\partial^3 \phi_i(x,k)}{\partial k^2 \partial x} = 2\alpha^2 \mathrm{Im}\left(\dfrac{W[x f_i(x)]}{W[f_i(x)]}\right) + 4\alpha^2 k\, \mathrm{Im}\left(\dfrac{\partial}{\partial k}\left(\dfrac{W[x f_i(x)]}{W[f_i(x)]}\right)\right)$

$\qquad\qquad + (\alpha k)^2 \mathrm{Im}\left(\dfrac{\partial^2}{\partial k^2}\left(\dfrac{W[x f_i(x)]}{W[f_i(x)]}\right)\right)$

(B9)

10  $\dfrac{\partial^3 \phi_i(x,k)}{\partial k^2 \partial x} = 2\alpha^2 \mathrm{Im}\left(\dfrac{W[x f_i(x)]}{W[f_i(x)]}\right) + 4\alpha^2 k\, \mathrm{Im}\left(\dfrac{\partial}{\partial k}\left(\dfrac{W[x f_i(x)]}{W[f_i(x)]}\right)\right) + (\alpha k)^2 \mathrm{Im}\left(\dfrac{\partial^2}{\partial k^2}\left(\dfrac{W[x f_i(x)]}{W[f_i(x)]}\right)\right)$

11  With :

12  $\dfrac{\partial^2}{\partial k^2}\left(\dfrac{W[x f_i(x)]}{W[f_i(x)]}\right) = \dfrac{1}{W[f_i(x)]}\dfrac{\partial^2 W[x f_i(x)]}{\partial k^2} - \dfrac{W[x f_i(x)]}{(W[f_i(x)])^2}\dfrac{\partial^2 W[f_i(x)]}{\partial k^2} - \dfrac{2}{(W[f_i(x)])^2}\dfrac{\partial W[f_i(x)]}{\partial k}\dfrac{\partial W[x f_i(x)]}{\partial k}$

[revised manuscript text omitted]

**Supplementary materials:**

[Figure]

**Figure S1**: Comparison between the univariate and the multivariate results for the six reaches (from 1 to 6) and using the four variables (velocity, hydraulic radius, bed shear stress, and cosine of local channel direction angle).

---

## Author Response (AR2)

**Editor's comments**

Dear Editor,

The authors would like to greatly thank Theresa Blume for reviewing our revised version. To meet your request, we have revised the manuscript according to the reviewer's comments and provide our responses point by point as follows.

Best regards,

On behalf of all co-authors,

Mounir Mahdade

**Anonymous Referee #1**

Dear Referee,

We thank you for your insightful comments and your thorough consideration and critical review that helped us to improve our manuscript.

Kind regards,

The authors

| *Comment of the reviewer* | Response of the authors |
|---|---|
| *Page 2, line 13: "any alternate morphological units": the definition of morphological units is provided later in the text (line 18), the authors might consider improving the readability of their manuscript by providing the definition in Line 13.* | We changed it to "to analyze any morphology characterized by alternating topographic forms (morphological units, MUs)". |
| *Page 2, paragraph 1.1: I understand the choice of using sub-paragraphs to improve the organization of the introduction, however, I found this part of the manuscript a bit abrupt. I suggest adding a sentence to clarify the consequentiality between different paragraphs.* | We added to the end of the paragraph: "However, it is necessary to find a method that can extract information concerning these morphologies (position, length, etc.). For this reason, it is interesting to list the works that quantitatively assess this morphological variability." |
| *Page 3, lines 19-21: could please the authors reword this sentence? Some of the terms are colloquial.* | I think maybe you want to write page 2 instead of page 3. We deleted "However, this may seem artificial and arbitrary". |
| *Page 4, lines 6-8: these lines repeat lines 14-15 in page 1.* | We deleted the repeat lines. |
| *Page 5, line 4: is "efficiency" the most appropriate term here?* | We changed it with "selection" |
| *Page 5, lines 6-7: the authors might consider adding a sentence or two the explain why a new method was required. I believe that such an* | We added a sentence to clarify: "Here, we are working with $Wbf$ and with a new automatic wavelength calculation method that uses the |

| | |
|---|---|
| *explicit explanation would clarify the scientific relevance of this work.* | whole covariance structure of a set of hydraulically-independent variables, without the need of ad hoc thresholding of these variables" |
| *Page 6, line 19: why did the authors use inverted commas for the word continuous?* | We wanted to emphasize this property. In the final version we deleted inverted commas. |
| *Page 6, line 20: "The" should not be capitalised.* | Corrected |
| *Page 6, lines 21-24: albeit the authors' strategy is reasonable, I believe that adding a few sentences to explain the reasoning underpinning the choice of the variables would increase the scientific soundness of the paper.* | We changed these sentences to: "This work will be done on four classical variables (velocity, hydraulic radius, bottom shear stress and local channel direction angle) because they respond directly to morphodynamic processes (flow convergence routing or meander migration) and they are independent hydraulic degrees of freedom." |
| *Page 6, lines 28-29: I suggest explaining here the motivations that led to the choice of the BDT method as a reference (I am aware that this choice is explained later in the test, however, I feel that adding an explanation here would improve the readability of the manuscript).* | We changed this sentence to: "… with the bedform differencing technique (BDT) developed by O'Neill and Abrahams (1984) to determine if they yield the same results in terms of spacing. We choose this method instead of threshold methods because the latter require ad hoc thresholding / parameter range definition from independent calibration data, which was not possible in our case." |
| *Page 8, line 5: I recommend replacing "it seems" with a sound explanation.* | Explanation added : "it is a more robust way of computing $\tau_b$ here than through the finite differentiation of the total head function $\frac{v(x)^2}{2g} + z(x)$ between adjacent cross-sections to get the energy slope J, given the typical number and spacing of surveyed cross-sections for each reach in the dataset." |
| *Page 9, line 8: I think that the adjective "interesting" is too vague. The authors might consider adding one or two sentences to explain what they mean with such a qualitative word (the readers can then check the appendix to learn all the details).* | We changed this sentence to: "The choice of the Morlet wavelet is justified by the analytical properties in its derivation and its flexibility due to the exponential form (see Appendix B)." |
| *Page 10, line 14: "amounts" – do the authors mean "requires"?* | Corrected |
| *Page 10, lines 16-20: I appreciate that the authors considerably improved the structure of their first manuscript, however, these lines contains significant references that belong to the literature review.* | We understand this remark, however these references are related to the use of wavelets in the analysis of river morphology from a more general point of view, so we just put them as an introduction to the methodological section for the sake of brevity. |
| *Page 11, line 20: "the extraction" OF…. I suggest improving the readability of this sentence.* | We changed it to: "the extraction of wavelengths" |

| | |
|---|---|
| *Page 12, line 6: the authors might consider adding "that is" before "the region…" to improve the readability of the sentence.* | Corrected |
| *Page 12, line 8: I suggest replacing "special" with something less colloquial. What does "special" mean in this context?* | We deleted it to avoid colloquial terms. |
| *Page 12, line 11: is the reference to equation 8 correct?* | Yes, it is the definition of the phase: $\phi(x, s) = \text{Im}(\ln \mathcal{W}\left[f(x)\right](x, s))$
So we can replace it in the equation: $\frac{\partial}{\partial x}\text{Im}(\ln \mathcal{W}\left[f(x)\right](x, s)) - k = 0$ |
| *Page 12, line 24: please remove "the" before "Eq.14".* | Corrected |
| *Page 15, lines 7-8: I suggest rewording this sentence to match the writing style of a scientific publication.* | We changed it to: "But then the local wavenumber would be specific to a given variable. Otherwise, the multivariate case requires to determine a common wavenumber between all the variables such that:" |
| *Page 15, lines 18-20: could please the authors clarify this sentence?* | We changed it to: "The procedure is applied to a set of variables $[v, R_h, \tau_b, \theta]$ and the goal is to seek for the commun wavenumber between all these variables. In the Fig. 7, we illustrate the result of this procedure applied on the Olivet River for all the four variables. A unique wavenumber is extracted which represents a co-evolution of all these variables.
As a result, the phase shift of every variable is calculated by:" |
| *Page 16, line 10: could please the authors clarify the meaning of "which is corrected afterwards."?* | We mean that this shift is corrected in the following positions. So, we changed it to: "is corrected in the following $x$ positions" |
| *Page18, line 8: could please the authors reword "work onto an…"?* | We changed it to: "The wavelet method extracts the wavelength for an assessed length …" |
| *Page 19, lines 7-22: in my opinion, the readers can infer these pieces of information (by themselves) from the table. Could please the authors provide an in-depth interpretation of these results?* | We added a comment at the end of the section : "However, there is no direct way of validating the estimates from these raw results : a way of doing so would be to build a synthetic, equivalent periodic geometry parameterized by the identified wavelength in order to verify that it yields, for example, a similar reach-average rating curve. This will be the subject of further work." |
| *Page 19, line 24: could please the authors explain the choice of the word "interestingly"? How "interestingly"?* | We changed it to "Consequently". It is the interpretation of what has been said before. |
| *Page 21, line 6: the BDT method has already been introduced. I suggest avoiding this repetition.* | We deleted the repetition. |
| *Page 21, line 7: the BDT method does not require expert judgement, however, is the BDT method* | We added: "… the literature (i.e., BDT). This method shows good results in the identification of |

| | |
|---|---|
| *accurate? Could please the authors provide some reference?* | these bedfroms according to some researches (e.g., Frothingham and Brown, 2002; Krueger and Frothingham, 2007). |
| *Page 22, line 14: could please the authors explain and reword "is done a sort of multiple calculations"?* | We changed this to: "while BDT does not directly calculate the wavelength. It is computed by averaging the pool-to-pool and riffle-to-riffle distances." |
| *Page 22, line 21: could please the authors replace "reasonable" with a quantitative or semi-quantitative assessment? In my opinion, "reasonable" is too vague.* | We replaced it with: "This indicates that a length greater than two cycles (pool-riffle) is always required …" |
| *Page 22, line 25: could please the authors elaborate on the potential reasons underpinning this "big difference"?* | We added some reasons: "This difference is due to the choice of the tolerance value, which is low in our case to the point of not filtering out the high-frequency variability of bed elevation and therefore gives a lower periodicity compared to the wavelets." |
| *Page 24, lines 2-3: what is the difference between "benchmark" and "reference" in this context?* | Benchmark means an example from the literature and a reference method is a true and accurate method. For that we will delete "reference" to avoid the misunderstanding. |
| *Page 25, line 17: the authors might consider adding a reference to the paragraph presenting the "good results" or a quantitative assessment.* | We added a comment on performance : "Ultimately, hydraulic modeling will be the true test of the potential of a pseudo-periodic equivalent geometry (e.g. for simulating a reach-average rating curve)" |
| *Page 25, line 26: I believe that the authors meant to write "it presents".* | Corrected |
| *Page 25, line 28: please rephrase "we can say the same thing".* | We replaced it with: "Similarly". |
| *Page 26, line 7: please reword "on the overall".* | We replaced it with: "As a result". |

[revised manuscript text omitted]

**Appendix B: Mathematical calculus for the wavelet transform**

**1 – The univariate case**

The conjugate form of the mother wavelet is:

$$\psi^*(\eta) = \pi^{-\frac{1}{4}}e^{-i\beta\eta-\frac{\eta^2}{2}} \tag{B1}$$

Its derivative in relation to the mute variable $\eta$ is:

$$\psi^{*'}(\eta) = -\pi^{-\frac{1}{4}}(i\beta+\eta)e^{-i\beta\eta-\frac{\eta^2}{2}} \tag{B2}$$
$$= -(i\beta+\eta)\psi^*(\eta)$$

In section 3 - 1, $\eta$ is a mute integration variable and $x$ appears only in the argument $\alpha k(\xi-x)$ of the
function $\psi^*$. By applying the derivation formula of a composite function, the derivative of the wavelet
transform is expressed by:

$$\frac{\partial}{\partial x}W[f(x)](x,k) = \sqrt{\alpha k}\int_{-\infty}^{+\infty} f(\eta)\frac{\partial}{\partial x}\big[\psi^*(\alpha k(\eta-x))\big]d\eta$$

$$= \sqrt{\alpha k}\int_{-\infty}^{+\infty} f(\eta)(-\alpha k)\psi^{*'}(\alpha k(\eta-x))d\eta$$

$$= (\alpha k)\sqrt{\alpha k}\int_{-\infty}^{+\infty} f(\eta)(i\beta+\alpha k(\eta-x))\psi^*(\alpha k(\eta-x))d\eta \tag{B3}$$

$$= (\alpha k)\sqrt{\alpha k}\int_{-\infty}^{+\infty} [(i\beta-\alpha kx)f(\eta)+\alpha k\eta f(\eta)]\psi^*(\alpha k(\eta-x))d\eta$$

$$= (\alpha k)(i\beta-\alpha kx)W[f(x)](x,k)+(\alpha k)^2W[xf(x)](x,k)$$

On the other hand, we have:

$$\frac{\partial}{\partial x}W[f(x)](x,k) = \frac{\partial}{\partial x}\big(R(x,k)e^{i\phi(x,k)}\big)$$
$$= \left[\frac{1}{R(x,k)}\frac{\partial R(x,k)}{\partial x}+i\frac{\partial\phi(x,k)}{\partial x}\right]R(x,k)e^{i\phi(x,k)} \tag{B4}$$

$$\frac{\partial}{\partial x}\text{Re}\big(\ln W[f(x)](x,k)\big) = \frac{1}{R(x,k)}\frac{\partial R(x,k)}{\partial x} = \text{Re}\left(\frac{1}{W[f(x)](x,k)}\frac{\partial}{\partial x}W[f(x)](x,k)\right)$$

$$\frac{\partial}{\partial x}\text{Im}\big(\ln W[f(x)](x,k)\big) = \frac{\partial\phi(x,k)}{\partial x} = \text{Im}\left(\frac{1}{W[f(x)](x,k)}\frac{\partial}{\partial x}W[f(x)](x,k)\right) \tag{B5}$$

Finally:

$$\frac{\partial\phi(x,k)}{\partial x} = \text{Im}\left((\alpha k)(i\beta-\alpha kx)+(\alpha k)^2\frac{W[xf(x)](x,k)}{W[f(x)](x,k)}\right)$$

$$\frac{\partial\phi(x,k)}{\partial x} = (\alpha k)\beta+(\alpha k)^2\text{Im}\left(\frac{W[xf(x)](x,k)}{W[f(x)](x,k)}\right) \tag{B6}$$

The previous expression numerically avoids the derivative of the function $\phi(x,k)$, which varies quickly for
large wavenumbers.

[Figure]

**Figure B1.** Steps of determining the local wavenumber K(x) using the wavelet univariate ridge analysis of the the velocity of the Olivet (3) reach, represented in the four panels. (A) The phase function $\phi(x,k)$; (B) the power's cone of influence of the wavelet to characterize the region where there is a maximum variability of the velocity in Neperian logarithm; (C) the function $\frac{\partial\phi(x,k)}{\partial x}$; (D) the function $k$.

**2 – The multivariate case**

In the multivariate case, we should resolve the Eq. 20 which contain three derivatives to compute. The first one is already done in the univariate case which is:

$$\frac{\partial \phi_i(x,k)}{\partial x} = (\alpha k)\beta + (\alpha k)^2 \mathrm{Im}\left(\frac{W[xf_i(x)]}{W[f_i(x)]}\right) \tag{B7}$$

The second one is the computation of $\frac{\partial^2 \phi_i(x,k)}{\partial k \partial x}$:

$$\frac{\partial^2 \phi_i(x,k)}{\partial k \partial x} = \alpha\beta + 2\alpha^2 k \mathrm{Im}\left(\frac{W[xf_i(x)]}{W[f_i(x)]}\right) + (\alpha k)^2 \mathrm{Im}\left(\frac{\partial}{\partial k}\left(\frac{W[xf_i(x)]}{W[f_i(x)]}\right)\right) \tag{B8}$$

For that we should develop $\frac{\partial}{\partial k}\left(\frac{W[xf_i(x)]}{W[f_i(x)]}\right)$:

$$\frac{\partial}{\partial k}\left(\frac{W[xf_i(x)]}{W[f_i(x)]}\right) = \frac{1}{W[f_i(x)]}\frac{\partial W[xf_i(x)]}{\partial k} - \frac{W[xf_i(x)]}{(W[f_i(x)])^2}\frac{\partial W[f_i(x)]}{\partial k}$$

We calculate each derivative:

$$\frac{\partial W[f_i(x)]}{\partial k} = \left(\frac{1}{\sqrt{k}}\frac{\partial \sqrt{k}}{\partial k}\right)W[f_i(x)] + \sqrt{\alpha k}\int_{-\infty}^{+\infty} f(\eta)\frac{\partial}{\partial k}[\psi^*(\alpha k(\eta - x))]d\eta$$

$$= \left(\frac{1}{2k}\right)W[f_i(x)] + \sqrt{\alpha k}\int_{-\infty}^{+\infty} f(\eta)\alpha(\eta - x)\psi^{*\prime}(\alpha k(\eta - x))d\eta$$

$$= \left(\frac{1}{2k}\right)W[f_i(x)] + \sqrt{\alpha k}\int_{-\infty}^{+\infty} f(\eta)\alpha(\eta - x)(i\beta + \alpha k(\eta - x))\psi^*(\alpha k(\eta - x))d\eta$$

$$= \left(\frac{1}{2k}\right)W[f_i(x)] + \sqrt{\alpha k}\int_{-\infty}^{+\infty} [(i\beta - \alpha^2 kx^2) + (-i\beta\alpha + 2\alpha^2 kx)\eta - (\alpha^2 k)\eta^2]f(\eta)\psi^*(\alpha k(\eta - x))d\eta$$

$$= \left(\frac{1}{2k} - \alpha^2 kx^2 + i\beta\alpha x\right)W[f_i(x)] + (2\alpha^2 kx - i\beta\alpha)W[xf_i(x)] - (\alpha^2 k)W[x^2 f_i(x)]$$

We find a general formulation with p=0…N:

$$\frac{\partial W[x^p f_i(x)]}{\partial k} = \left(\frac{1}{2k} - \alpha^2 kx^2 + i\beta\alpha x\right)W[x^p f_i(x)] + (2\alpha^2 kx - i\beta\alpha)W[x^{p+1} f_i(x)]$$
$$- (\alpha^2 k)W[x^{p+2} f_i(x)]$$

The third one is the computation of $\frac{\partial^3 \phi_i(x,k)}{\partial k^2 \partial x}$:

$$\frac{\partial^3 \phi_i(x,k)}{\partial k^2 \partial x} = 2\alpha^2 \text{Im}\left(\frac{W[xf_i(x)]}{W[f_i(x)]}\right) + 4\alpha^2 k\text{Im}\left(\frac{\partial}{\partial k}\left(\frac{W[xf_i(x)]}{W[f_i(x)]}\right)\right)$$
$$+ (\alpha k)^2 \text{Im}\left(\frac{\partial^2}{\partial k^2}\left(\frac{W[xf_i(x)]}{W[f_i(x)]}\right)\right)$$

(B9)

$$\frac{\partial^3 \phi_i(x,k)}{\partial k^2 \partial x} = 2\alpha^2 \text{Im}\left(\frac{W[xf_i(x)]}{W[f_i(x)]}\right) + 4\alpha^2 k\text{Im}\left(\frac{\partial}{\partial k}\left(\frac{W[xf_i(x)]}{W[f_i(x)]}\right)\right) + (\alpha k)^2 \text{Im}\left(\frac{\partial^2}{\partial k^2}\left(\frac{W[xf_i(x)]}{W[f_i(x)]}\right)\right)$$

With :

$$\frac{\partial^2}{\partial k^2}\left(\frac{W[xf_i(x)]}{W[f_i(x)]}\right) = \frac{1}{W[f_i(x)]}\frac{\partial^2 W[xf_i(x)]}{\partial k^2} - \frac{W[xf_i(x)]}{(W[f_i(x)])^2}\frac{\partial^2 W[f_i(x)]}{\partial k^2} - \frac{2}{(W[f_i(x)])^2}\frac{\partial W[f_i(x)]}{\partial k}\frac{\partial W[xf_i(x)]}{\partial k}$$

[revised manuscript text omitted]

**Supplementary materials:**

[Figure]

**Figure S1**: Comparison between the univariate and the multivariate results for the six reaches (from 1 to 6) and using the four variables (velocity, hydraulic radius, bed shear stress, and cosine of local channel direction angle)

---

## Author Response (AR3)

**Reply to Editor's comments:**

1) Prof. Pasternack asked you to provide the script the you used for your analyses and I agree with him that this would be helpful. Please provide this script in the supplementary material or appendix and refer to it in the methods section.

Response:

Concerning the script, we understand the request of Pr. Pasternack, but currently, the code used is not written in a user-friendly manner and not really fit for diffusion. It is based on a Scilab migration of the Torrence & Compo Matlab toolbox + several additional functions such as the ones performing the derivatives of the wavelet in x or k, or computing the wavelet ridges (the latter are basically level curves of specific functions, and can be readily obtained in Scilab through the contour2di function for example). On top of this comes a lot of data reading routines (bathymetry of reaches, hydraulic variables, etc.). We think that providing all the necessary mathematical expressions for wavelet analysis in the Appendix avoids the burden of passing the full messy code to the reader. But, we will mention that it is available upon request in *code availability*. We hope you will understand this position.

2) There are still quite a few grammatical errors, please do another round of proof-reading

3) Title: in a river channel or in river channels - please correct p.1,l.4: large-scale proxies p.1,l.11: a set of multiple variables - this seems to be an unnecessary doubling, it could be either "a set of variables" or just "multiple variables".

p.2,l.19: state of the art p.15,l.7: "on the other hand" instead of "otherwise"? Otherwise is a bit confusing here.

p.16,l.8: "the correlations between these variables which are well respected" - what do you mean be "well respected"?

p.21,l.14 "according to some studies", not "some researches"

p.25,l.7: delete "some"

p.26,l.4 "possibility of extending the work to other rivers with other types of MUs, other longer reaches with a large number..." - I think there is something missing after the comma, possibly an "or".

Response:

All theses grammatical errors are corrected, thank you.

p.1,l.21: "this is their essential input" - what does "their" refer to? Please clarify

Response:

It refers to hydraulic modeling; we changed it with: "this is the essential input of models …".

p.2,l.16-18:

"As it is necessary to find a method that can extract information concerning morphologies (position, length, etc.). For this reason, it is interesting to list the works that quantitatively assess this morphological variability." - I suggest to delete this sentence here as it is more confusing than helpful. If I understand correctly the reviewer was more referring to the fact that the 2nd paragraph under 1 is already starting to describe state of the art methods to determine morphological variability as well as the study objectives and so the sudden break given by the heading of 1.1 comes a bit surprising as the text provides more of a continued discussion of the same topic before and after this break. You also start to provide the study objectives under 1, and then repeat this under 1.2. To improve this you could check if some of the information provided under 1 could also be distributed to both 1.1 and 1.2 so that there is a clearer differentiation between the three sections, keeping section 1 rather short.

Response:

In section 1, we introduced the relation between the hydraulic modeling and the variability of river geometry, which is related to the morphological units, and of course, we should state some literature there. In section 1.1, we focused only on extracting MUs features. And in section 1.2, the objectives of the study. But, of course, your suggestions are very interesting to make this paragraph clear. For that, we modified the section 1 by:

[revised manuscript text omitted]

Its derivative depending on the mute variable $\eta$ is:

$$\psi^{*\prime}(\eta) = -\pi^{-\frac{1}{4}}(i\beta + \eta)e^{-i\beta\eta - \frac{\eta^2}{2}}$$
$$= -(i\beta + \eta)\psi^*(\eta) \tag{B2}$$

In section 3 - 1, $\eta$ is a mute integration variable, and $x$ appears only in the argument $\alpha k(\xi - x)$ of the function $\psi^*$. By applying the derivation formula of a composite function, the derivative of the wavelet transform is expressed by:

$$\frac{\partial}{\partial x}W[f(x)](x,k) = \sqrt{\alpha k}\int_{-\infty}^{+\infty} f(\eta)\frac{\partial}{\partial x}\big[\psi^*(\alpha k(\eta - x))\big]d\eta$$
$$= \sqrt{\alpha k}\int_{-\infty}^{+\infty} f(\eta)(-\alpha k)\psi^{*\prime}(\alpha k(\eta - x))d\eta$$
$$= (\alpha k)\sqrt{\alpha k}\int_{-\infty}^{+\infty} f(\eta)(i\beta + \alpha k(\eta - x))\psi^*(\alpha k(\eta - x))d\eta \tag{B3}$$
$$= (\alpha k)\sqrt{\alpha k}\int_{-\infty}^{+\infty}[(i\beta - \alpha kx)f(\eta) + \alpha k\eta f(\eta)]\psi^*(\alpha k(\eta - x))d\eta$$
$$= (\alpha k)(i\beta - \alpha kx)W[f(x)](x,k) + (\alpha k)^2 W[xf(x)](x,k)$$

On the other hand, we have:

$$\frac{\partial}{\partial x}W[f(x)](x,k) = \frac{\partial}{\partial x}\big(R(x,k)e^{i\phi(x,k)}\big)$$
$$= \left[\frac{1}{R(x,k)}\frac{\partial R(x,k)}{\partial x} + i\frac{\partial\phi(x,k)}{\partial x}\right]R(x,k)e^{i\phi(x,k)} \tag{B4}$$

$$\frac{\partial}{\partial x}\text{Re}\big(\ln W[f(x)](x,k)\big) = \frac{1}{R(x,k)}\frac{\partial R(x,k)}{\partial x} = \text{Re}\left(\frac{1}{W[f(x)](x,k)}\frac{\partial}{\partial x}W[f(x)](x,k)\right)$$
$$\frac{\partial}{\partial x}\text{Im}\big(\ln W[f(x)](x,k)\big) = \frac{\partial\phi(x,k)}{\partial x} = \text{Im}\left(\frac{1}{W[f(x)](x,k)}\frac{\partial}{\partial x}W[f(x)](x,k)\right) \tag{B5}$$

Finally:

$$\frac{\partial\phi(x,k)}{\partial x} = \text{Im}\left((\alpha k)(i\beta - \alpha kx) + (\alpha k)^2\frac{W[xf(x)](x,k)}{W[f(x)](x,k)}\right) \tag{B6}$$

$$\frac{\partial\phi(x,k)}{\partial x} = (\alpha k)\beta + (\alpha k)^2 \text{Im}\left(\frac{W[xf(x)](x,k)}{W[f(x)](x,k)}\right)$$

The previous expression numerically avoids the derivative of the function $\phi(x,k)$, which varies quickly for
large wavenumbers.

[Figure]

**Figure B1**. Steps of determining the local wavenumber K(x) using the wavelet univariate ridge analysis of  the
velocity of the Olivet (3) reach, represented in the four panels. (A) The phase function $\phi(x,k)$; (B) the  power's
cone of influence of the wavelet to characterize the region where there is a maximum variability of the velocity in
Neperian logarithm; (C) the function $\frac{\partial\phi(x,k)}{\partial x}$; (D) the function $k$.

**2 – The multivariate case**

In the multivariate case, we should resolve the Eq. 20, which contains three derivatives to compute. The
first one is already done in the univariate case :

$$\frac{\partial\phi_i(x,k)}{\partial x} = (\alpha k)\beta + (\alpha k)^2 \text{Im}\left(\frac{W[xf_i(x)]}{W[f_i(x)]}\right) \tag{B7}$$

The second one is the computation of $\frac{\partial^2\phi_i(x,k)}{\partial k\partial x}$:

$$\frac{\partial^2 \phi_i(x,k)}{\partial k \partial x} = \alpha\beta + 2\alpha^2 k \text{Im}\left(\frac{W[xf_i(x)]}{W[f_i(x)]}\right) + (\alpha k)^2 \text{Im}\left(\frac{\partial}{\partial k}\left(\frac{W[xf_i(x)]}{W[f_i(x)]}\right)\right) \tag{B8}$$

For that, we should develop $\frac{\partial}{\partial k}\left(\frac{W[xf_i(x)]}{W[f_i(x)]}\right)$:

$\frac{\partial}{\partial k}\left(\frac{W[xf_i(x)]}{W[f_i(x)]}\right) = \frac{1}{W[f_i(x)]}\frac{\partial W[xf_i(x)]}{\partial k} - \frac{W[xf_i(x)]}{(W[f_i(x)])^2}\frac{\partial W[f_i(x)]}{\partial k}$

We calculate each derivative:

$\frac{\partial W[f_i(x)]}{\partial k} = \left(\frac{1}{\sqrt{k}}\frac{\partial\sqrt{k}}{\partial k}\right)W[f_i(x)] + \sqrt{\alpha k}\int\limits_{-\infty}^{+\infty} f(\eta)\frac{\partial}{\partial k}[\psi^*(\alpha k(\eta - x))]d\eta$

$= \left(\frac{1}{2k}\right)W[f_i(x)] + \sqrt{\alpha k}\int\limits_{-\infty}^{+\infty} f(\eta)\alpha(\eta - x)\psi^{*'}(\alpha k(\eta - x))d\eta$

$= \left(\frac{1}{2k}\right)W[f_i(x)] + \sqrt{\alpha k}\int\limits_{-\infty}^{+\infty} f(\eta)\alpha(\eta - x)(i\beta + \alpha k(\eta - x))\psi^*(\alpha k(\eta - x))d\eta$

$= \left(\frac{1}{2k}\right)W[f_i(x)] + \sqrt{\alpha k}\int\limits_{-\infty}^{+\infty} [(i\beta - \alpha^2 k x^2) + (-i\beta\alpha + 2\alpha^2 k x)\eta - (\alpha^2 k)\eta^2]f(\eta)\psi^*(\alpha k(\eta - x))d\eta$

$= \left(\frac{1}{2k} - \alpha^2 k x^2 + i\beta\alpha x\right)W[f_i(x)] + (2\alpha^2 k x - i\beta\alpha)W[xf_i(x)] - (\alpha^2 k)W[x^2 f_i(x)]$

We find a general formulation with p=0…N:

$\frac{\partial W[x^p f_i(x)]}{\partial k} = \left(\frac{1}{2k} - \alpha^2 k x^2 + i\beta\alpha x\right)W[x^p f_i(x)] + (2\alpha^2 k x - i\beta\alpha)W[x^{p+1} f_i(x)]$

$\qquad\qquad - (\alpha^2 k)W[x^{p+2} f_i(x)]$

The third one is the computation of $\frac{\partial^3 \phi_i(x,k)}{\partial k^2 \partial x}$:

$$\frac{\partial^3 \phi_i(x,k)}{\partial k^2 \partial x} = 2\alpha^2 \text{Im}\left(\frac{W[xf_i(x)]}{W[f_i(x)]}\right) + 4\alpha^2 k \text{Im}\left(\frac{\partial}{\partial k}\left(\frac{W[xf_i(x)]}{W[f_i(x)]}\right)\right)$$
$$+ (\alpha k)^2 \text{Im}\left(\frac{\partial^2}{\partial k^2}\left(\frac{W[xf_i(x)]}{W[f_i(x)]}\right)\right) \tag{B9}$$

$\frac{\partial^3 \phi_i(x,k)}{\partial k^2 \partial x} = 2\alpha^2 \text{Im}\left(\frac{W[xf_i(x)]}{W[f_i(x)]}\right) + 4\alpha^2 k \text{Im}\left(\frac{\partial}{\partial k}\left(\frac{W[xf_i(x)]}{W[f_i(x)]}\right)\right) + (\alpha k)^2 \text{Im}\left(\frac{\partial^2}{\partial k^2}\left(\frac{W[xf_i(x)]}{W[f_i(x)]}\right)\right)$

With :

$\frac{\partial^2}{\partial k^2}\left(\frac{W[xf_i(x)]}{W[f_i(x)]}\right) = \frac{1}{W[f_i(x)]}\frac{\partial^2 W[xf_i(x)]}{\partial k^2} - \frac{W[xf_i(x)]}{(W[f_i(x)])^2}\frac{\partial^2 W[f_i(x)]}{\partial k^2} - \frac{2}{(W[f_i(x)])^2}\frac{\partial W[f_i(x)]}{\partial k}\frac{\partial W[xf_i(x)]}{\partial k}$

[revised manuscript text omitted]

**Supplementary materials:**

[Figure]

**Figure S1**: Comparison between the univariate and the multivariate results for the six reaches (from 1 to 6) and using the four variables (velocity, hydraulic radius, bed shear stress, and cosine of local channel direction angle)